# Unsupervised Similarity Learning for Spectral Clustering

## Abstract

Spectral clustering is a widely adopted method capable of identifying non-convex cluster boundaries. However, traditional spectral clustering requires the definition of a predefined similarity metric for constructing the Laplacian matrix, a requirement that limits flexibility and adaptability. Instead of predefining this metric upfront as a fixed parametric function, we introduce a novel approach that learns the optimal parameters of a similarity function through parameter optimization. This optimizes a similarity function to assign high similarity values to data pairs with shared discriminative features and low values to those without such features. Previous methods, which also adapt similarity measures, often depend on hyperparameters or resort to non-convex optimization strategies, which are unsuitable in unsupervised scenarios due to their dependency on initial conditions and inability to validate using labels. Our proposed method leverages convex optimization to learning the parameters of the similarity metrics without relying on hyperparameters, thus ensuring robust and reliable unsupervised learning suitable for spectral clustering. We demonstrate the effectiveness of our approach across multiple benchmark datasets, confirming its superiority in performance and adaptability.

## 1 Introduction

Spectral clustering (SC) (von Luxburg, 2007) is a very effective method to cluster data with non-convex separation boundaries. In this approach, the individual data points can be considered as nodes of a fully connected graph where the edge weights are the similarity values. The underlying assumption for SC is that for every pair of data in the training set, the similar ones should have a high similarity score and vice versa. Thus, the objective for SC becomes minimizing the graph cut (GC), namely minimizing the sum of edge weight between clusters and maximizing the sum within clusters (see fig. 1).

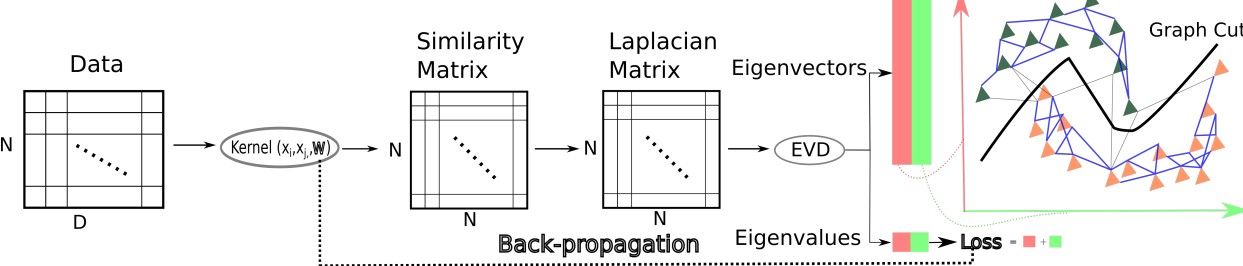

Figure 1: The proposed kernel function (an alternation of the radial basis function) is optimized using gradient descent to minimize the sum of the first $K$ eigenvalues.

The performance of SC is primarily attributed to the manually chosen parameters of the *similarity metric*. However, an incorrectly chosen similarity metric can be detrimental to the clustering performance as it might not emphasize enough the similar pairs relative to the dissimilar ones (see fig. 2c). Hence, instead of hand-picking the parameters of the similarity metric, one can try to learn them instead (see fig. 1). To do so, we use the graph-cut loss to generate a gradient for the parameter of the kernel function. This gradient is then utilized to optimize the parameter of the kernel function (see fig. 1).

Existing similarity learning methods (Kang et al., 2017; Qiao et al., 2018; Zelnik-manor & Perona, 2004; Ng et al., 2001; Karasuyama & Mamitsuka, 2013; Roweis & Saul, 2000; Zhang et al., 2014; Wang & Zhang, 2008; Daitch et al., 2009; Elhamifar & Vidal, 2012; Liu et al., 2010; Zhuang et al., 2011; Zhang et al., 2010; Lu et al., 2013; Liu et al., 2014; Li & Fu, 2013; 2014) feature one or more hyperparameters in the objective function. Unlike supervised learning, validating hyperparameter values through an annotated validation set is not possible in an unsupervised learning setup. Given the need for an extensive exploratory and exploitative search in supervised hyperparameter optimization, it is unlikely that optimal parameters will be selected due to the suboptimal range of values being considerably larger than the optimal range in real-world scenarios. If we apply the same principle to an unsupervised setting, randomly selecting hyperparameters would likely result in suboptimal values. This, in turn, would result in incorrect similarity scores affecting the downstream clustering performance (Fan et al., 2020).

Moreover, existing similarity learning methods (Kang et al., 2017; Qiao et al., 2018; Zelnik-manor & Perona, 2004; Ng et al., 2001; Karasuyama & Mamitsuka, 2013; Roweis & Saul, 2000; Zhang et al., 2014; Wang & Zhang, 2008; Daitch et al., 2009; Elhamifar & Vidal, 2012; Liu et al., 2010; Zhuang et al., 2011; Zhang et al., 2010; Lu et al., 2013; Liu et al., 2014; Li & Fu, 2013; 2014) employ non-convex optimization, which complicates their application in unsupervised scenarios due to their dependence on the initialization and the challenges associated with validating solutions in the absence of labels.

To mitigate these drawbacks, *we propose a novel variation in the radial basis function (RBF) kernel and try to learn its optimal bandwidth through a convex optimization without introducing any hyperparameter in the loss function (see fig. 1).* In this study, the initial number of clusters (i.e., K) is predetermined; however, the unsupervised estimation of the cluster count is possible by identifying density peaks within the data topology, as demonstrated in Rodriguez & Laio (2014).

Building on these insights, the method we introduce offers an unsupervised approach to learning similarities that determine the optimal weights for the edges of a fully connected weighted graph, on which SC is then applied. We assess the clustering performance of our method on various image and text datasets, demonstrating an improvement over alternative approaches.

The contributions of this study are outlined as follows:

- We introduce a novel modification of the RBF kernel that enables the learning of its parameters through convex optimization;

- Our approach has been empirically validated and demonstrated to enhance clustering accuracy across various standard benchmark datasets.

## 2 Method

The proposed method works directly on the raw data (e.g., in fig. 2a) and tries to further emphasize any existing separation within the data points (see fig. 1). Accordingly, the optimal similarity function would lower the smaller pairwise distances so that all the data points would eventually collapse to a single point (see fig. 2c). At the same time, the more pronounced distances are amplified (see fig. 2c). To do so, we choose to exponentiate these pairwise distances in a controlled manner (see eq. (1) and fig. 2b), where the smaller the distance between two data points, the higher their similarity and vice versa. As a result, the bigger distances are exponentiated towards higher values much more than the smaller ones (see fig. 2b). Since the new transformed distances are normalized within a unit vector, increasing the bigger magnitudes would render the smaller ones towards even smaller values and vice versa (see fig. 2c).

The proposed kernel enables a controlled exponential modulation of the distances via its bandwidth $\sigma$ (see eq. (1)). To ensure the convexity of the problem, we redefine the distance function as $d(x,y) = u(x,y) + 1$, where $u(x,y) = \|x - y\|^2$ is used for non-binary data and $u(x,y) = 1 - \cos_{\text{sim}}(x,y)$ for binary data. Note that these distances are non-negative (i.e., $u(x,y) \geq 0$), and under these definitions, the kernel function in eq. (1) demonstrates convex behavior over the bandwidth interval $\sigma \in (0, \frac{\sqrt{6}}{3}\}$ (see Corollary 1). Since SC is predominantly affected by relative rather than absolute distances, an uniform translation of the Euclidean

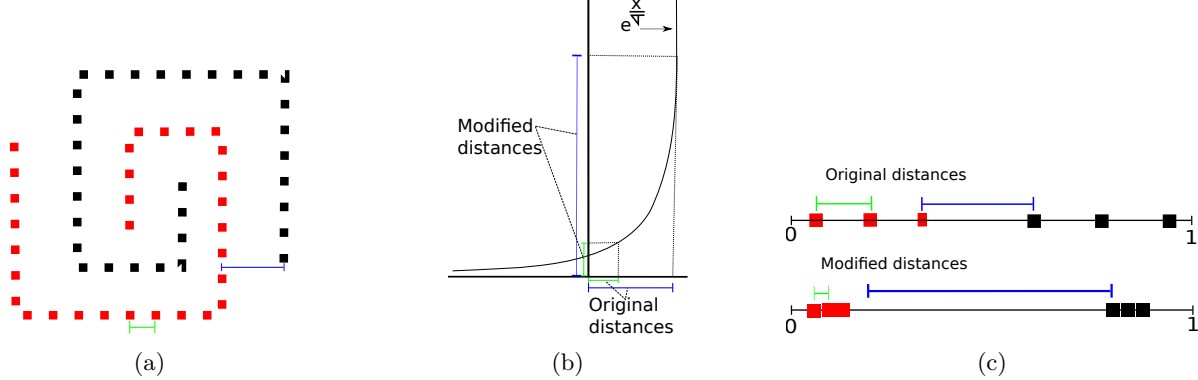

(a)                                          (b)                                          (c)

Figure 2: From the data presented in fig. 2a, the separation between each cluster is enhanced. Each distance is modulated through the exponential function of the proposed kernel, as detailed in fig. 2b. Consequently, larger distances undergo greater amplification compared to smaller distances. After normalization to a unit vector, the magnitude of smaller distances decreases, while that of larger ones increases relative to each other, as shown in fig. 2c.

metric by a unit value does not alter the hierarchy of pairwise distances within a dataset (see Appendix E). Consequently, such a shift does not impact the SC algorithm's results. Moreover, the proposed modification of the RBF (see eq. (1)) preserves the intrinsic kernel characteristics of the RBF (see Lemma 1).

$$K(x,y) = e^{-\frac{d(x,y)}{\sigma^2}} = e^{-\frac{u(x,y)+1}{\sigma^2}} = \sum_{n=0}^{\infty} -\frac{(u(x,y)+1)^n}{(\sigma^2)^n n!}, \text{s.t: } \sigma \text{ is kernel bandwidth.} \quad (1)$$

Notice that a kernel function $K$ maps two inputs $x$ and $y$ to a scalar that represents their inner product in a higher-dimensional space, expressed as $K(x,y) = \langle \phi(x), \phi(y) \rangle$, where $\phi$ is an implicit mapping.

**Lemma 1.** *The proposed function in eq. (1) is a kernel.*

*Proof.* The function described in eq. (1) can be decomposed as follows:

$$K(x,y) = e^{-\frac{d(x,y)}{\sigma^2}} = e^{-\frac{u(x,y)+1}{\sigma^2}} = e^{\frac{-1}{\sigma^2}} e^{\frac{-u(x,y)}{\sigma^2}} = \underbrace{e^{\frac{-1}{\sigma^2}}}_{c>0} \underbrace{e^{\frac{-u(x,y)}{2\sigma^2}}}_{K_1(x,y)} \underbrace{e^{\frac{-u(x,y)}{2\sigma^2}}}_{K_1(x,y)} = cK_1(x,y)K_1(x,y). \quad (2)$$

Given that $K_1(x,y)$ is an RBF kernel and considering the linear and multiplicative properties of kernels as detailed in Propositions 13.1 and 13.2 in Schölkopf & Smola (2002), it follows that $K(x,y)$ is a valid kernel.

$\square$

The superior performance of the proposed kernel is primarily due to its capability to project data into an infinite-dimensional space, as described by the last term in eq. (1) (Shashua, 2009). By properly tuning its bandwidth $\sigma$, one can smoothly truncate the infinite-dimensional expansion (see eq. (1)) and select the dimensions to be incorporated in the comparison. In other words, the bandwidth $\sigma$ adjusts the amount of modulation for the similarity metric. Using the proposed kernel as a generic similarity formulation, the task reduces to optimization of $\sigma$ that yields the most representative distance from a general formulation. To amplify the distance $d(x,y)$ as presented in eq. (1), the parameter $\sigma$ must be set to a value less than one (i.e., $\sigma \leq 1$) where $\sigma = 0$ is not allowed. *Traditionally, in a SC setting, $\sigma$ is a hyper-parameter which is handpicked. Instead, we propose learning $\sigma$ directly from the given data.*

## 2.1 Learning operation

In SC, the objective remains identical to other clustering techniques; similar data are projected together while the dissimilar ones are far apart (von Luxburg, 2007). This objective is equivalent to the GC (see fig. 1) when data are considered as nodes of the connected graph while the edges are the pair similarity values. Thus, to construct an objective function that is amenable to the gradient-based method, one has to consider the data projections (i.e., $f$) and the similarity values ($A_{i,j} \in \mathbb{R}, \forall i, j \in \{1, \cdots, N\}$) simultaneously. Utilizing the general formulation for the graph-cut (GC) (von Luxburg, 2007) (see eq. (3)) as the objective function, it is possible to get an empirical assessment of clustering performance, where $\boldsymbol{L}$ in eq. (3) is the Laplacian matrix and $f$ are the data embeddings.

$$\text{GC}[\boldsymbol{A}, f] = \sum_{i,j,j<i} A_{i,j}(f_i - f_j)^2 = f^\top \boldsymbol{L} f. \tag{3}$$

The formulation in eq. (3) reduces the task to learning the similarity matrix (i.e., $\boldsymbol{A}^{[N,N]}$ also known as kernel matrix) and the representation of the data (i.e., $f$).

As a result, the computation of GC on the graph reduces to matrix multiplication. Since the utilized similarity is parameterized solely by the bandwidth $\sigma$, we optimize the computed similarity matrix by tuning $\sigma$ (see eq. (4)).

$$\arg \min_{\sigma, f} \text{GC}[\boldsymbol{A}(\sigma), f(\boldsymbol{A})] \to \frac{\partial^2 \text{GC}[\boldsymbol{A}(\sigma), f]}{\partial \sigma \partial f} = 0. \tag{4}$$

Instead of finding both $\sigma$ and $f$ simultaneously, an alternation between each parameter is adopted (see Algorithm 1). We utilize a Lagrangian multiplier to restrict the domain of the data representation $f$ to a unit magnitude $f^\top f = 1$ which enables the derivation of $f$ and $\lambda$ as the eigenvectors and eigenvalue of the Laplacian matrix $\boldsymbol{L}$ (see eq. (5) from Strang (2006)). The Laplacian matrix $\boldsymbol{L}$ is defined as $\boldsymbol{L} = \boldsymbol{D} - \boldsymbol{A}$ where $\boldsymbol{D}$ is degree matrix. The degree matrix is diagonal, with each diagonal entry ($i.e., \boldsymbol{D}_{ii}$) representing the sum of the weights of all edges connected to node ($i$), ($i.e., \boldsymbol{D}_{ii} = \sum_{j=1}^{n} \boldsymbol{A}_{ij}$).

$$\frac{\partial\{\text{GC}[\boldsymbol{A}(\sigma), f] + \lambda(f^\top f - 1)\}}{\partial f} = 0 \to \boldsymbol{L} f = \lambda f \tag{5}$$

Since the kernel is symmetric (see eq. (1)), the resulting Laplacian matrix $\boldsymbol{L}$ is also symmetric upon which eigenvalue decomposition (EVD) produces real and non-negative eigenvalues ($\lambda_i \geq 0, \forall i \in 1, \cdots, N$), and orthogonal eigenvectors ($f^\top f = 1$) (Strang, 2006). In the case of using the bottom $K$ eigenvector ($i.e., f_{i,\cdots,K}$), the GC reduces to the sum of the corresponding eigenvalues $\lambda$ as in eq. (6) (see Ky Fan's Theorem (So et al., 2010)):

$$\text{GC}[\boldsymbol{A}(\sigma), f_{i,\cdots,K}(A)] = \text{GC}(\sigma)|_{f_{i,\cdots,K}} = \sum_{i=1}^{K} \lambda_i(\sigma) \tag{6}$$

In contrast to eq. (5), which offers a closed-form solution, finding $\sigma$ from eq. (6) requires the use of gradient descent because it involves a nonlinear operation as dictated by the kernel in eq. (1). Despite the intrinsic non-convex nature of SC (von Luxburg, 2007), we introduce a convex approach for determining the optimal bandwidth $\sigma$ in the interval $\Sigma = (0, \sqrt{6}/3]$ (see Theorem 1).

**Theorem 1.** *The loss function* $loss = \sum_{i=1}^{K} \lambda_i(\sigma)$ *is convex in* $\forall \sigma \in \Sigma = (0, \sqrt{6}/3], \forall f \in R^N$.

*Proof.* The convexity of Equation (6) is easily shown by proving that:

$$\frac{\partial^2 \text{loss}(\sigma)}{\partial \sigma^2} = \sum_{i=1}^{K} \frac{\partial^2 \{\lambda_i(\sigma)\}}{\partial \sigma^2} \geq 0. \tag{7}$$

Given that $\boldsymbol{L}(\sigma)$ is itself symmetric, its first and second differentiation w.r.t $\sigma \in \Sigma$ is also another real symmetric Laplacian matrix (see Lemma 6). Hence, the first and second differentiation of the eigenvalues of a symmetric matrix is $d\lambda = f^\top d\boldsymbol{L} f$ (Petersen & Pedersen, 2008) and $d^2\lambda = f^\top d^2\boldsymbol{L} f$ (see Lemma 3 in Appendix B) respectively. The second derivative reduces to:

$$\frac{\partial^2 \text{loss}(\sigma)}{\partial \sigma^2} = \sum_{i=1}^{K} \frac{\partial^2 \{\lambda_i(\sigma)\}}{\partial \sigma^2} = \sum_{i=1}^{K} f_i^\top \frac{\partial^2 \boldsymbol{L}(\sigma)}{\partial \sigma^2} f_i \geq 0, \forall \sigma \in \Sigma. \tag{8}$$

Since any real symmetric Laplacian matrix is a positive semi-definite (PSD) (von Luxburg, 2007), its Rayleigh quotient is always positive (see Strang (2006)). Hence, even for eigenvector $f$, the result is always non-negative

$$f_i^\top \frac{\partial^2 \boldsymbol{L}(\sigma)}{\partial \sigma^2} f_i \geq 0, \forall i \in \{1, \cdots, K\},$$

resulting in

$$\sum_{i=0}^{k} f_i^\top \frac{\partial^2 \boldsymbol{L}(\sigma)}{\partial \sigma^2} f_i \geq 0, \forall f \in R^N.$$

$\square$

Due to this convexity condition, the parameter $\sigma$ can be iteratively refined via Newton's method (see eq. (9)) to achieve a faster than gradient descent convergent progression toward the optimal solution (Strang, 2006).

$$\sigma_i \leftarrow \sigma_i - \frac{\text{grad}_\sigma}{\text{step}_\sigma} \leftarrow \sigma_i - \frac{\sum_{i=0}^{k} f_i^\top \frac{\partial \boldsymbol{L}(\sigma)}{\partial \sigma} f_i}{\sum_{i=0}^{k} f_i^\top \frac{\partial^2 \boldsymbol{L}(\sigma)}{\partial \sigma^2} f_i} \tag{9}$$

The new algorithm adopts a two-stage operation within each iteration, starting computation of the Laplacian matrix and its first and second derivative w.r.t $\sigma$, followed by the $\sigma$ update (see Algorithm 1.).

Notice that the search for $\sigma$ remains within the convex space if, and only if, $f_{1,\ldots,K}$ remain restricted as the eigenvectors of $\boldsymbol{L}$. Moreover, as both $\sigma$ and the series of eigenvectors $f_{1,\ldots,K}$ are iteratively updated to minimize the loss function, it can be concluded that the loss function is continuously evolving, increasingly approaching a more flattened landscape.

Although reducing the magnitude of $\sigma$ does lead to a lower GC value in SC, setting $\sigma$ arbitrarily close to zero is not advisable. This setting results in all the edge weights of the graph converging to zero. Consequently, each data point becomes isolated into its own cluster, effectively obliterating the underlying data structure and preventing the detection of inherent clusters.

As the denominator involves the square of $\sigma$ (see eq. (1)), both positive and negative values of $\sigma$ of the same magnitude have the same modulation effect in the proposed kernel. Hence, we consider only positive values for $\sigma$, as its negative values do not introduce any extra impact. Furthermore, if the loss at the initialization is positive, then the optimal $\sigma$ is in $\Sigma$ (i.e., $\sigma_{\text{optimal}} \in \Sigma$) (see Lemma 2).

**Lemma 2.** *If the loss $\left(\sigma = \frac{\sqrt{6}}{3}\right) > 0$, then $\sigma_{optimal} < \frac{\sqrt{6}}{3}$.*

*Proof.* Given the initial condition

$$\text{loss}\left(\sigma = \frac{\sqrt{6}}{3}\right) > 0,$$

and noting that the gradient w.r.t $\sigma$ is non-negative over $\mathbb{R}_{\geq 0}$ (i.e., $\frac{\partial \text{loss}(\sigma)}{\partial \sigma} \geq 0, \forall \sigma \in \mathbb{R}_{\geq 0}$ (see Theorem 2 in Appendix C)), it follows that the loss is non-decreasing over $\mathbb{R}_{\geq 0}$. Namely:

$$\text{loss}(\sigma_1) \geq \text{loss}(\sigma_2), \forall \sigma_1, \sigma_2 \in \mathbb{R}_{\geq 0}, \text{ s.t: } \sigma_1 \geq \sigma_2.$$

Therefore,

$$\forall \sigma > \frac{\sqrt{6}}{3}, \quad \text{loss}(\sigma) \geq \text{loss}\left(\sigma = \frac{\sqrt{6}}{3}\right),$$

which indicates that the values of the loss increase for $\sigma$ values greater than $\frac{\sqrt{6}}{3}$. Consequently, the optimal $\sigma$ cannot be exceed $\frac{\sqrt{6}}{3}$. $\qquad\square$

To introduce the scale invariance in the distance matrix ($\Delta^{[N,N]}$ in Algorithm 1) and potential numerical underflowing, its matrix values are normalized so that its minimum value becomes zero, and its maximum value becomes one.

This proposed method works solely on the Laplacian matrix setup. Other versions of normalized Laplacian were tried, i.e., symmetric normalized Laplacian $\left(\boldsymbol{L}_{\text{normal}}^{\text{symmetric}} = \boldsymbol{D}^{-\frac{1}{2}} \boldsymbol{L} \boldsymbol{D}^{-1\frac{1}{2}}\right)$, and normalized Laplacian row-wise stochastic version $\left(\boldsymbol{L}_{\text{normal}}^{\text{stochastic}} = \boldsymbol{D}^{-1} \boldsymbol{L}\right)$ but the gradient computation was not updating $\sigma$ in the anticipated direction. Furthermore, since existing kernels did not perform well, the proposed kernel has been designed to enable better performance and facilitate parameter updates within the convex optimization setup.

**Complexity analysis:** The Euclidean distance matrix is computed once outside the loop at $O(N^2)$ complexity. Since we are looking solely for the first $K$ eigenvectors, the EVD complexity is $O(K \times N^2)$ (Golub & Van Loan). Similarly, the complexity of the first and second derivatives is $O(K \times N^2)$ since the output of the computational graph has $K$ eigenvectors. Over $n_{iter}$ iterations, the total complexity would be $O(n_{iter} \times K \times N^2)$.

## 3 Related work

The similarity learning domain contains various methodologies established on different fundamental assumptions (Qiao et al., 2018), primarily categorized into empirical heuristics and graph learning approaches.

**Empirical heuristic:** The heuristic methods assigns a unique $\sigma$ for each data point by considering local statistics within a selected neighborhood in the raw (Zelnik-manor & Perona, 2004) or embedded space (Ng et al., 2001). Yet, scaling this approach is difficult because it requires computing as many $\sigma$ values as the dataset size, and the choice of neighborhood critically affects performance.

**Graph learning:** Rather than relying on heuristics, some studies adopt a data-driven approach to individually learn a scale parameter (i.e., $\{\sigma_k\}_{k=1}^d$, where $d$ is the number of dimensions), for each dimension of the dataset (Karasuyama & Mamitsuka, 2013). The learning process involves minimizing the cumulative distance between each data point $x_i$ and the weighted mean of its neighbors as in eq. (10):

$$\min_{\{\sigma_k\}_{k=1}^d} \sum_{i=1}^n \left\| x_i - \frac{1}{d_j} \sum_{j=1}^n E_{ij} W_{ij} x_j \right\|, \text{ s.t: } d_j \rightarrow \text{node degree} \qquad (10)$$

In eq. (10), $W_{ij}$ represents the affinity between data points, defined as[2] $W_{ij} = \exp\left(-\frac{1}{2}\sum_{k=1}^d \frac{d(x_{ik}, x_{jk})}{\sigma_k^2}\right)$, and $E_{ij}$ denotes the elements of the connectivity matrix $\boldsymbol{E}$ (filled with binary values to indicate the presence (i.e., $E_{ij} = 1$) or absence (i.e., $E_{ij} = 0$) of an edge between each data pair), which are determined separately

---

[1]The matrix $\mathbf{J}$ has the same dimensions as $\mathbf{C}$ and all of its entries are ones.
[2]$d(x_{ik}, x_{jk})$ is the distance.

---

**Algorithm 1** Similarity Learning

---

1: **procedure** LEARNSIMILARITY(Data $\mathbf{X}^{[N,D]}$, Number of clusters $K$)
2:     Compute the distance matrix $\Delta^{[N,N]}$.
3:     Scale and translate the distance matrix: $\boldsymbol{C}^{[N,N]} \leftarrow \frac{\Delta^2}{\max(\Delta^2)} + \mathbf{J}^1$.
4:     Initialize $\sigma_0 \leftarrow \sqrt{6}/3$.
5:     **for** $i = 1$ to $n_{\text{iter}}$ **do**
6:         Compute the affinity matrix: $\boldsymbol{A} \leftarrow e^{-\boldsymbol{C}/\sigma_i^2}$.
7:         Compute derivatives of $\boldsymbol{A}$ relative to $\sigma_i$:

$$\frac{\partial \boldsymbol{A}}{\partial \sigma_i} \leftarrow \frac{2\boldsymbol{C}}{\sigma_i^3} e^{-\boldsymbol{C}/\sigma_i^2},$$

$$\frac{\partial^2 \boldsymbol{A}}{\partial \sigma_i^2} \leftarrow \frac{2\boldsymbol{C}}{\sigma_i^6} e^{-\boldsymbol{C}/\sigma_i^2}(2\boldsymbol{C} - 3\sigma_i^2 \mathbf{J}).$$

8:         Compute the degree and Laplacian matrices and their derivatives:

$$\boldsymbol{D} \leftarrow \text{diag}(\sum \boldsymbol{A}),$$

$$\frac{\partial \boldsymbol{D}}{\partial \sigma_i} \leftarrow \text{diag}(\sum \frac{\partial \boldsymbol{A}}{\partial \sigma_i}),$$

$$\frac{\partial^2 \boldsymbol{D}}{\partial \sigma_i^2} \leftarrow \text{diag}(\sum \frac{\partial^2 \boldsymbol{A}}{\partial \sigma_i^2}),$$

$$\boldsymbol{L} \leftarrow \boldsymbol{D} - \boldsymbol{A},$$

$$\frac{\partial \boldsymbol{L}}{\partial \sigma_i} \leftarrow \frac{\partial \boldsymbol{D}}{\partial \sigma_i} - \frac{\partial \boldsymbol{A}}{\partial \sigma_i},$$

$$\frac{\partial^2 \boldsymbol{L}}{\partial \sigma_i^2} \leftarrow \frac{\partial^2 \boldsymbol{D}}{\partial \sigma_i^2} - \frac{\partial^2 \boldsymbol{A}}{\partial \sigma_i^2}.$$

9:         Perform EVD on $\boldsymbol{L}$ and update $\sigma$:

$$f_{1,\dots,K} \leftarrow \text{EVD}(\boldsymbol{L}),$$

$$\text{grad}_\sigma \leftarrow \sum_{i=1}^{K} f_i^\top \frac{\partial \boldsymbol{L}}{\partial \sigma} f_i,$$

$$\text{step}_\sigma \leftarrow \sum_{i=1}^{K} f_i^\top \frac{\partial^2 \boldsymbol{L}}{\partial \sigma^2} f_i,$$

$$\sigma_i \leftarrow \sigma_i - \frac{\text{grad}_\sigma}{\text{step}_\sigma}.$$

10:     **return** $\sigma_i, f_{1,\dots,K}$

---

using a heuristic. The values of $\{\sigma_k\}_{k=1}^d$ are optimized through a gradient descent procedure, iteratively refining them to improve the representation of the manifold's local geometry.

Certain techniques induce locality constraints to evaluate similarity values (i.e., $\boldsymbol{W}$) (Roweis & Saul, 2000). This is achieved by minimizing the cumulative difference of individual data relative to neighborhood constraints as in eq. (11).

$$\min_{\boldsymbol{W}} \sum_{i=1}^{n} \left\| x_i - \sum_{j=1}^{n} E_{ij} W_{ij} x_j \right\|, \text{ s.t: } \sum_{j=1}^{n} E_{ij} W_{ij} x_j = 1 \tag{11}$$

Further enhancements to this method include the imposition of an additional constraint on the similarity values (i.e., $W_{ij} \geq 0$) (Wang & Zhang, 2008).

To overcome the limitations associated with heuristic-based methods for establishing the connectivity matrix (i.e., $E$), alternative methods have been developed that optimize the weight matrix $W$ directly, bypassing the need for defining $E$ (Zhang et al., 2014). The objective (see eq. (12)) ensures a smooth transition between nearby points, balanced clustering, and uniform edge weight distribution[3].

$$\min_{W \geq 0} \underbrace{\sum_{i,j=1}^{n} \|x_i - x_j\| \, W_{ij}}_{\text{Smooth transition}} + \gamma_1 \underbrace{\sum_{j=1}^{n} (d_j - 1)^2}_{\text{Balanced clustering}} + \quad \gamma_2 \underbrace{\sum_{j=1}^{n} W_{ij}^2}_{\text{Uniform edge weight distribution}} \tag{12}$$

In comparison to methods that focus on local attributes (Zhang et al., 2014), a global model (Daitch et al., 2009) tries to fit a graph to a vector of data using the following quadratic objective eq. (13).

$$\min_{W} \sum_{i=1}^{n} \left\| d_j x_i - \sum_{j \neq i}^{n} W_{ij} x_j \right\|^2 \text{ s.t: } d_j \geq 1 \text{ and } \sum_{i} (\max(0, 1 - d_j))^2 \leq \eta n \tag{13}$$

An alternate methodology (Elhamifar & Vidal, 2012) tries to fit a graph to a specified data set by enforcing sparsity on the learned similarity values via $L_1$ regularization term (see eq. (14)).

$$\min_{W} \left( \|X - XW\|_F^2 + \gamma_1 \|W\|_1 \right) \tag{14}$$

In contrast, some researchers have used a low-rank constraint by replacing the sparsity term (i.e., the $L_1$ norm, $\|W\|_1$) with a low-rank term (i.e., the nuclear norm, $\|W\|_*$) (Liu et al., 2010). While other approaches have even combined the low-rank constraint with sparsity (Zhuang et al., 2011), local constraint (Lu et al., 2013), Markov random walk (Liu et al., 2014), B-matching (Li & Fu, 2013; 2014).

Another method, instead of fitting a graph directly on the raw data, tries to alternate between learning a linear transformation matrix (i.e., $P$ in eq. (15)) of the data such that their pairwise comparison is more robust to noise and learning the similarity values (i.e., $W$ eq. (15)) (Zhang et al., 2010) using the locality-preserving principle.

$$\min_{P, W} \sum_{i,j=1}^{n} \|P x_i - P x_j\|^2 W_{ij} + \gamma_1 \sum_{i,j=1}^{n} W_{ij} \ln(W_{ij}), \text{ s.t: } \sum_{i=1}^{n} \|P x_i\|^2 = 1, \text{ and } \sum_{j=1}^{n} W_{ij} = 1, W_{ij} \geq 0. \tag{15}$$

Since $\|P x_i - P x_j\|^2 = \|x_i - x_j\| P^\top P \|x_i - x_j\|$ one can consider that this method is unsupervised Mahalanobis distance learning.

Unlike existing methods, our approach is the first to explore learning a similarity metric for SC without adding any hyperparameters to the loss function.

## 4 Experiments

**Benchmark datasets:** We ran a series of experiments on benchmark diagnostic datasets (see figs. 3 and 5 to 9 in Appendix F) (Fränti & Sieranoja) as well as real-world datasets (see Table 4) to showcase the method's

---

[3]$\gamma_1, \gamma_2, \eta$ in eqs. (12) to (15) are regularizer hyperparameter coefficient.

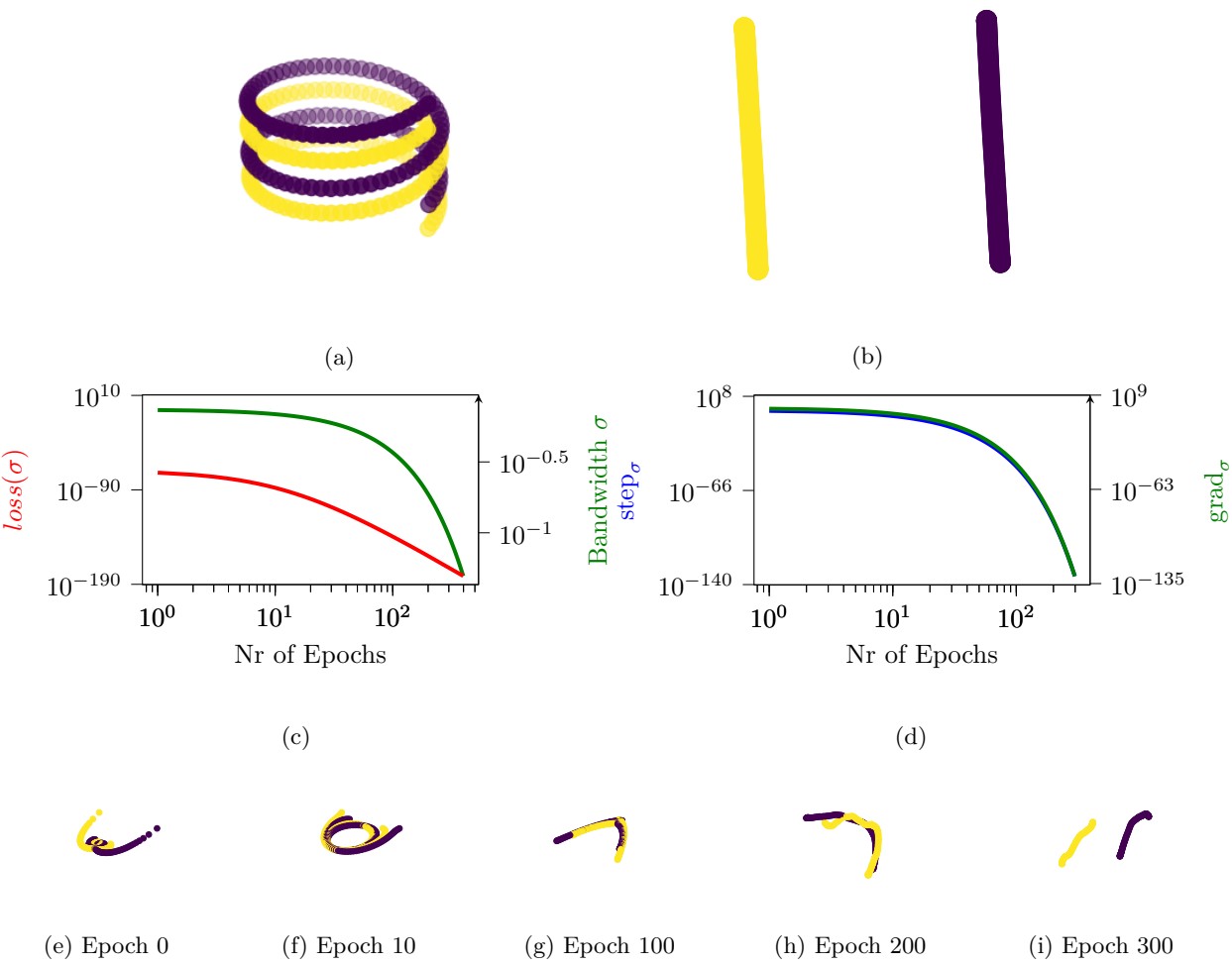

Figure 3: Benchmark (two spirals) dataset in fig. 3a. Our method learns a linear separation of these two spirals fig. 3b. In fig. 3b is the eigen-embedding of the benchmark data. The loss (loss $= \sum_{i=1}^{K} \lambda_i$) along with the bandwidth $\sigma$, decreases consistently with the number of iterations (see fig. 3c). Furthermore, the $\text{grad}_\sigma$ and the $\text{step}_\sigma$ decreases consistently with the number of iterations (see fig. 3d). A snapshot of the optimized trajectory is in figs. 3e to 3i

capabilities at separating non-convex shapes (see fig. 3a and figs. 5a, 6a, 7a, 8a, 9a, 10a, 12a, 13a, 14a and 15a in Appendices F and G) into linearly separable clusters (see fig. 3b and figs. 5b, 6b, 7b, 8b, 9b, 10b, 12b, 13b, 14b and 15b in Appendices F and G). In all scenarios, the method tries to amplify the big pairwise distances within a confined space while reducing the small pairwise distances. Hence, the method finds a persisting gap within the topology of the dataset and tries to linearize it in the embedding space (see fig. 3b and figs. 5b, 6b, 7b, 8b and 9b in Appendix F). The analysis of the optimization trajectory of $\sigma$ reveals a consistently non-increasing and a non-negative first and second derivative (i.e., $\text{grad}_\sigma, \text{step}_\sigma$) throughout the trajectory (see fig. 3d and figs. 5d, 6d, 7d, 8d, 9d, 10d, 12d, 13d, 14d and 15d in Appendices F and G). This behavior empirically supports the convex nature of the proposed algorithm.

The GC of the given Laplacian (i.e., loss $= \sum_{i=1}^{K} \lambda_i$) decreases throughout the training process (see figs. 3c and 4b and figs. 5c, 6c, 7c, 8c, 9c, 10c, 12c, 13c, 14c and 15c in Appendices F and G). Additionally, considering that both the first and second derivatives of the Laplacian matrix (i.e., $\frac{\partial L}{\partial \sigma}$ and $\frac{\partial^2 L}{\partial \sigma^2}$) remain Laplacian matrices (see Lemma 7 in Appendix D), sharing the same eigenvectors (see Lemma 3 in Appendix B), their respective GC are defined by the gradient (i.e, $\text{grad}_\sigma = \sum_{i=1}^{K} f_i^\top \frac{\partial L}{\partial \sigma} f_i$) and step update (i.e., $\text{step}_\sigma =$

$\sum_{i=1}^{K} f_i^\top \frac{\partial^2 \boldsymbol{L}}{\partial \sigma^2} f_i$) (see eqs. (8) and (18) in Appendix C). As a result, as $\sigma$ decreases, the GC decreases for the Laplacian matrix and its first and second derivatives since all characterize the same graph (see fig. 3d and figs. 5d, 6d, 7d, 8d, 9d, 10d, 12d, 13d, 14d and 15d in Appendices F and G). Eventually, as $\sigma$ approaches its optimum, both the gradient (i.e., $\sum_{i=1}^{K} f_i^\top \frac{\partial \boldsymbol{L}}{\partial \sigma} f_i$) and the step update (i.e., $\sum_{i=1}^{K} f_i^\top \frac{\partial^2 \boldsymbol{L}}{\partial \sigma^2} f_i$) become smaller.

**Image and text datasets:** Furthermore, this linearization of the separation boundary reduces the complexity of the downstream tasks one can perform upon these embeddings. Clustering is one popular unsupervised task whose performance is solely the capability for learning the best representation of the data topology (von Luxburg, 2007). Hence, to compare the capability of our unsupervised method, we run K-means + + (Jain, 2010; Arthur & Vassilvitskii) on the eigen-embeddings where the clustering performance would be a proxy evaluator for the learned similarity matrix. We extend to text and image datasets (see Table 1) where the models try to find linear separation between data clusters. In such a real data scenario, the proposed method tries to separate the dataset using persistent low-density (i.e., gap) regions throughout the dataset.

Table 1: Description of the experimented datasets.

| Dataset | Size | Dimensions | Clusters |
|---------|------|------------|----------|
| JAFFE   | 212  | 177        | 7        |
| Umnist  | 575  | 644        | 20       |
| YALE    | 165  | 105        | 15       |
| BA      | 1404 | 320        | 36       |
| COIL20  | 1440 | 1024       | 20       |
| ORL     | 400  | 1024       | 36       |

**Data preparation:** The image datasets are assumed to be correctly registered beforehand; therefore, no feature selection is needed upfront. To ensure the equivariance of the scale of the features, standardization ($\hat{x}_{\text{stand}} = \frac{x - \hat{\mu}}{\sqrt{\hat{\sigma}^2}}$) is performed before on non-binary data.

Once the similarity learning has been completed, we perform K-means++ (Jain, 2010; Arthur & Vassilvitskii) clustering upon the eigen-embeddings. To mitigate the effect of the initialization, K-means + + results are aggregated over 50 different times until convergence.

In Table 2 compares our approach with two graph-based learning methods: Sparse Subspace Clustering (SSC) (Elhamifar & Vidal, 2012) and Low-Rank Representation (LRR) (Elhamifar & Vidal, 2012). Additionally, we consider six models that integrate graph construction with spectral embedding. These include Clustering with Adapting Neighbors (CAN) (Nie et al., 2014) and its variations: Projected Clustering with Adapting Neighbors (PCAN) (Yang et al., 2022) and Self-Weighted Clustering with Adapting Neighbors (SWCAN) (Nie et al., 2020). Additionally, we assess several other methods in our evaluation: LAPIN, which optimizes a bipartite graph for subspace clustering (Nie et al., 2023); DOGC, which learns the clustering discretization and graph learning simultaneously (Han et al., 2020); JGSED, a recent technique that integrates graph construction, spectral embedding, and cluster discretization (Qiao et al., 2018); and JSESR, an approach that combines spectral embedding with spectral rotation (Pang et al., 2020). Last but not least, self-tuning spectral clustering (SelfT) (Zelnik-manor & Perona, 2004) is also included in the comparison.

In contrast to our approach, the effectiveness of other methods depends significantly on selecting appropriate hyperparameters and the type of regularization used. *The aim is to demonstrate the superiority of the proposed method in terms of clustering performance and ease of use.*

Recent advancements in deep clustering methods have demonstrated good performance on well-established image datasets. These methods comprise two stages: unsupervised representation learning, followed by the actual clustering step (Ren et al., 2022; Zhou et al., 2022). The bulk of the research in this area concentrates on representation learning. This phase involves organizing the data such that similar items are closely embedded, whereas dissimilar items are spaced further apart. This configuration simplifies the task for conventional clustering techniques. However, our proposed method focuses on the improvement of the clustering technique itself.

**Evaluation metrics:** Unlike classifiers, where the type of classes and their predictions are kept intact, the prediction cluster indicators are permuted in clustering. Therefore, the clustering evaluation reduces to comparing two different types of sets, i.e., ground-truth labels set and cluster indicator. As a result, more than a single metric is required to assess all aspects of the clustering performance normalized mutual information (NMI) and accuracy (ACC) capture different aspects of the performance.

NMI in eq. (16) compares two sets using entropy as a criterion. It measures how much entropy is required to describe the second set using the entropy of the first set. Normalization scales the metric from zero to one.

$$\text{NMI}(\boldsymbol{X}, \boldsymbol{Y}) = \frac{\sum_{\mathbf{x} \in \boldsymbol{X}} \sum_{\mathbf{y} \in \boldsymbol{Y}} P(\mathbf{x}, \mathbf{y}) \log \frac{P(\mathbf{x}, \mathbf{y})}{P(\mathbf{x}) P(\mathbf{y})}}{\max \left\{ -\sum_{\mathbf{x} \in \boldsymbol{X}} P(\mathbf{x}) \log P(\mathbf{x}), -\sum_{\mathbf{y} \in \boldsymbol{Y}} P(\mathbf{y}) \log P(\mathbf{y}) \right\}} \tag{16}$$

In contrast, accuracy (ACC in cf. eq. (17)) measures the pairwise exactness of the predicted clusters with the ground-truth clusters. This metric requires a bi-partite set alignment as in an unsupervised learning setup; the predicted cluster index does not correspond to the ground-truth label value at the individual data level. Therefore, the two sets ($\boldsymbol{X}$ and $\boldsymbol{Y}$ of size $n$) are aligned through the Kuhn-Munkres algorithm (KMA) (Kuhn, 1955) (*i.e.*, $\hat{\boldsymbol{Y}} = \text{KMA}(\boldsymbol{Y})$) upfront (see eq. (17)).

$$\text{ACC}[\boldsymbol{X}, \hat{\boldsymbol{Y}} = \text{KMA}(\boldsymbol{Y})] = \frac{\sum_{\mathbf{x} \in \boldsymbol{X}, \hat{\mathbf{y}} \in \hat{\boldsymbol{Y}}} \delta(\mathbf{x}, \hat{\mathbf{y}})}{n} \tag{17}$$

Table 2: ACC and NMI over the different datasets. Comping methods have the average performance while ours is only one estimation.

| Data | SelfT | JSESR | LRR | SSC | CAN | PCAN | SWCAN | LAPIN | DOGS | JGSED | Ours |
|------|-------|-------|-----|-----|-----|------|-------|-------|------|-------|------|
| | | | | | **ACC** | | | | | | |
| YALE | 47.65 | 30.18 | 26.67 | 24.48 | 24.24 | 25.45 | 26.67 | 30.45 | 30.93 | 32.12 | **56.96** |
| JAFFE | 54.28 | 27.83 | 29.72 | 29.72 | 28.77 | 28.30 | 30.19 | 30.68 | 31.79 | 32.08 | **86.85** |
| COIL20 | 38.17 | 81.37 | 73.40 | 76.18 | **83.54** | 83.33 | 79.03 | 81.96 | 80.90 | 82.64 | 69.79 |
| BA | 23.67 | 46.59 | 44.59 | 45.87 | 42.24 | 43.59 | 46.08 | 46.83 | 38.53 | 47.86 | **50.21** |
| UMIST | 40.69 | 61.03 | 56.17 | 65.74 | 64.87 | 57.74 | 71.30 | 71.30 | 62.17 | **71.30** | 66.78 |
| ORL | 45.78 | 46.59 | 44.59 | 45.87 | 42.24 | 43.59 | 46.08 | 46.83 | 38.53 | 47.86 | **67.75** |
| | | | | | **NMI** | | | | | | |
| YALE | 59.64 | 33.48 | 27.53 | 25.69 | 20.71 | 23.54 | 24.05 | 32.98 | 33.42 | 33.78 | **64.01** |
| JAFFE | 69.86 | 9.97 | 16.62 | 14.92 | 13.33 | 12.88 | 15.33 | 15.91 | 18.06 | 17.59 | **93.34** |
| COIL20 | 63.15 | 89.25 | 82.90 | 90.29 | 91.09 | 89.08 | 89.19 | 90.03 | 88.96 | **91.17** | 85.32 |
| BA | 38.96 | 60.83 | 57.93 | **62.50** | 54.24 | 49.55 | 57.98 | 59.04 | 49.82 | 60.64 | 60.85 |
| UMIST | 60.93 | 75.18 | 70.22 | 77.68 | 78.21 | 70.05 | 82.95 | 82.95 | 78.55 | 82.95 | **85.00** |
| ORL | 72.75 | 80.37 | 83.20 | 79.26 | 76.45 | 73.16 | 76.51 | 82.74 | 78.09 | 81.80 | **86.30** |

**Results:** The model demonstrates an improvement in the NMI and ACC metric (see Table 2). This improvement suggests that the entropy of the cluster indices generated by our similarity matrix more closely approximates the actual class indices for the dataset under consideration. Given that NMI provides a broader comparative analysis between true labels and predicted labels, clustering accuracy is computed as a supplementary metric to estimate performance at the individual level. It is important to emphasize that although our method facilitates reproducible similarity learning through its operation within a convex setup and the absence of hyperparameters, competing methods lack these combined attributes. Consequently, the performance of these alternative approaches is heavily contingent upon the initialization of the learning process, which introduces considerable variability and precludes the assurance of even average performance levels.

Similarly, optimizing the kernel bandwidth $\sigma$ follows a similar trajectory as in the benchmark dataset. The method progressively tries to reach an optimal value and decreases the gradient (see fig. 4b and figs. 16b,

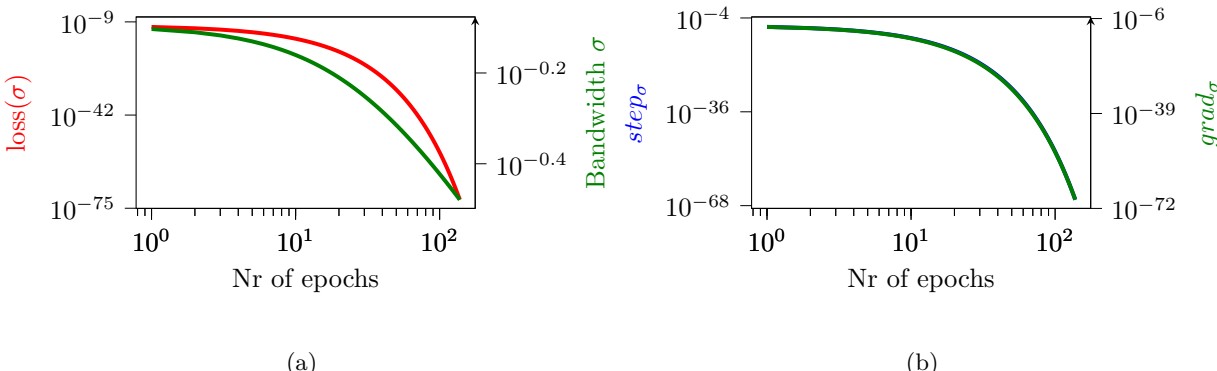

Figure 4: The loss (loss $= \sum_{i=1}^{K} \lambda_i$) over the iterations along with the optimization of the kernel bandwidth $\sigma$ for the JAFFE dataset in fig. 4a. The $_\sigma$ and $step_\sigma$ in fig. 4b.

17b, 18b, 19b, 20b, 21b, 22b and 23b in Appendix H). Likewise, the cumulative magnitude of the first $K$ eigenvalues progresses towards smaller values as in the case of the benchmark data experiments (see fig. 4a and figs. 16a, 17a, 18a, 19a, 20a, 21a, 22a and 23a in Appendix H). This behavior indicates the formation of the clusters as the $\sigma$ progresses towards more optimal values.

## 5 Conclusion

The proposed method learns the optimal bandwidth of the proposed kernel under the optimization objective that maximizes the linear separation for K clusters when using the first K eigenvectors of the Laplacian matrix. Alternatively, arbitrarily selecting the kernel bandwidth risks choosing a suboptimal value that might either underemphasize the distances, resulting in indistinguishable clusters, or overemphasize them, leading to individual data being isolated as its own cluster.

The goal of the proposed method is to function without labels in a fully unsupervised manner that eliminates the dependence on hyperparameters. When working without labeled data, validating any hyperparameters directly through the dataset or the objective function is impossible. Moreover, choosing suboptimal hyperparameter values can negatively impact the effectiveness of clustering in subsequent tasks. Consequently, our work introduces a framework for similarity learning in SC that avoids the introduction of any hyperparameters.

We showcase our method's ability to separate highly non-convex shapes and project the dataset into a linearly separable topology on a series of benchmark datasets. The proposed method is validated on popular datasets and compared with recent similarity learning models. Our proposed method shows an improvement in both normalized mutual information and accuracy. Notice that all alternative models' performance strictly depends on their initial settings. *Since these initial settings are not validated in an unsupervised setting, the optimal performance of the competing methods is never guaranteed.*

As a future work, it should be noted that SC itself inherently requires the predefined specification of the number of clusters $K$ within the dataset. Looking forward, an area for further research would be incorporating methods for autonomously estimating the number of clusters. A promising approach is to identify density peaks within the data's topological structure, an idea inspired by the technique presented in the study by Rodriguez & Laio (2014). Adopting such a strategy could enhance the unsupervised aspect of SC, expanding it from similarity learning to include autonomous cluster number determination.

The proposed method is constrained to the proposed kernel, which requires precise pairwise image alignment, which is impractical for real-world image datasets. To increase applicability, the approach should incorporate deep learning models, which produce translation-invariant embeddings, thus reducing the need for exact pairwise image alignment.

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

# A    Notation

- *ACC*: Accuracy.

- *GC*: Graph cut.

- *SC*: Spectral clustering.

- *MNI*: Normalized mutual information.

- *KMA*: Kuhn-Munkres algorithm.

- *EVD*: Eigenvalue decomposition.

- $u(x, y)$: Distance metric between two data points $x$ and $y$. It utilizes '1 - cosine similarity' for binary data and 'squared Euclidean distance' for other data types.

- $K(x, y)$: Kernel function quantifying the affinity between two individual data points $x$ and $y$.

- $N$: Total number of individual data points in the dataset.

- $D$: The number of dimensions represented in the dataset.

- $K$: The number of clusters defined within the dataset.

- $\mathbf{A}$: Affinity matrix, a square $N \times N$ matrix.

- $\mathbf{D}$: Degree matrix, a diagonal $N \times N$ matrix derived from the affinity matrix.

- $\mathbf{L}$: Laplacian matrix, a square $N \times N$ matrix, defined by $\mathbf{L} = \mathbf{D} - \mathbf{A}$.

- $\mathbf{M}$: Incident matrix associated with the Laplacian matrix, such that $\mathbf{L} = \mathbf{M}^\top \mathbf{M}$.

- $\mathbf{J}$: Constant matrix with all entries set to one dimension $N \times N$.

- $\Delta$: Distance matrix whose entries are determined by the '1-cosine distance' for binary data or by the squared Euclidean distance for other types.

- $\mathbf{C}$: Matrix representing scale and translation modifications applied to the distance matrix $\Delta$.

- $\mathbf{E}$: Connectivity matrix, $N \times N$, indicating the presence of an edge between any two data points $x$ and $y$.

- $\mathbf{W}$: Similarity matrix, $N \times N$, describing the similarity scores between pairs of data points.

- $\mathbf{F}$: Matrix consisting of eigenvectors of the Laplacian matrix $\mathbf{L}$, dimensions $N \times N$.

- $\Lambda$: Diagonal matrix containing the eigenvalues of the Laplacian matrix $\mathbf{L}$, dimensions $N \times N$.

- $\mathbf{P}$: Linear transformation matrix, dimensions $N \times N$.

- $\mathbf{f}$: Specific eigenvector of the Laplacian matrix $\mathbf{L}$, representing embedded data points.

- $\lambda$: Scalar value representing an eigenvalue of the Laplacian matrix $\mathbf{L}$.

- $\sigma$: Bandwidth parameter for the kernel function used in the analysis.

- $\gamma$ is a regularizer hyperparameter coefficient.

- $\eta$ is a regularizer hyperparameter coefficient.

- $c$ is a scaling constant.

## B  The second derivative of the EVD on a symmetric matrix

**Lemma 3.** *Given a real symmetric matrix $\boldsymbol{L}$, the second derivative of its eigenvalues is expressed as*

$$d^2\lambda = f^\top(d^2\boldsymbol{L})f,$$

*where $f$ denotes the eigenvector corresponding to the eigenvalue $\lambda$, that is, $\lambda = f^\top \boldsymbol{L} f$.*

*Proof.* Consider the eigenvalue equation for the matrix $\boldsymbol{L}$ as:

$$\boldsymbol{L}f = \lambda f.$$

Differentiating both sides yields the first derivative of the eigenvalue leads to

$$d\lambda = f^\top(d\boldsymbol{L})f.$$

Here, $f$ is the eigenvector of $\boldsymbol{L}$ corresponding to the eigenvalue $\lambda$, which implies that $f$ is also an eigenvector of the derivative matrix $d\boldsymbol{L}$. Thus, $d\lambda$ is the eigenvalue associated with the eigenvector $f$ for the matrix $d\boldsymbol{L}$.

By extending the same reasoning to the second derivative of the matrix $\boldsymbol{L}$, we deduce that the second derivative of the eigenvalue $\lambda$ w.r.t the parameter is expressed by:

$$d^2\lambda = f^\top(d^2\boldsymbol{L})f.$$

This completes the proof.

$\square$

## C  On the non-negativity of the $\sigma$ gradient.

**Theorem 2.** *The gradient of the loss function is non-negative, i.e., $\frac{\partial loss(\sigma)}{\partial \sigma} \geq 0, \forall \sigma \in \mathbb{R}_{\geq 0}$.*

*Proof.* This is demonstrated by observing that:

$$\frac{\partial \mathrm{loss}(\sigma)}{\partial \sigma} = \sum_{i=1}^{K} \frac{\partial \lambda_i(\sigma)}{\partial \sigma} \geq 0$$

Differentiating the eigenvalues of a symmetric matrix yields $d\lambda = f^\top d\boldsymbol{L} f$, which implies:

$$\frac{\partial \mathrm{loss}(\sigma)}{\partial \sigma} = \sum_{i=1}^{K} f_i^\top \frac{\partial \boldsymbol{L}(\sigma)}{\partial \sigma} f_i \geq 0, \quad \forall \sigma \in \mathbb{R}_{\geq 0} \tag{18}$$

Given that $\frac{\partial \boldsymbol{L}(\sigma)}{\partial \sigma}$ is PSD (see Lemma 4), its Rayleigh quotient is non-negative, thus confirming that $f_i^\top \frac{\partial \boldsymbol{L}(\sigma)}{\partial \sigma} f_i \geq 0$ for all $i \in \{1, \ldots, K\}$.

This results in:

$$\sum_{i=1}^{K} f_i^\top \frac{\partial \boldsymbol{L}(\sigma)}{\partial \sigma} f_i \geq 0, \quad \forall f \in \mathbb{R}^N$$

$\square$

**Lemma 4.** *The first derivative of the Laplacian matrix $\left(\frac{\partial \boldsymbol{L}(\sigma)}{\partial \sigma}\right)$ is a PSD matrix $\forall \sigma \in \mathbb{R}_{\geq 0}$.*

*Proof.* Initially, one can observe that because $\boldsymbol{L}(\sigma) = \boldsymbol{L}^\top(\sigma)$ is symmetric, its derivative also maintains symmetry (see Corollary 2):

$$\frac{\partial \boldsymbol{L}(\sigma)}{\partial \sigma} = \frac{\partial \boldsymbol{L}^\top(\sigma)}{\partial \sigma}$$

Given this symmetry of $\frac{\partial \boldsymbol{L}(\sigma)}{\partial \sigma}$, all its eigenvalues of the derivative are real (Strang, 2006).

Since $\frac{\partial \boldsymbol{L}(\sigma)}{\partial \sigma}$ is another Laplacian matrix (see Lemma 5) it can be represented as the product of an incidence matrix $\mathbf{M}$:

$$\frac{\partial \boldsymbol{L}(\sigma)}{\partial \sigma} = \mathbf{M}^\top \mathbf{M}.$$

For any vector $\mathbf{x}$, this leads to:

$$\mathbf{x}^\top \frac{\partial \boldsymbol{L}(\sigma)}{\partial \sigma} \mathbf{x} = \|\mathbf{M}\mathbf{x}\|^2 \geq 0.$$

Thus, $\frac{\partial \boldsymbol{L}(\sigma)}{\partial \sigma}$ is PSD.

$\square$

**Lemma 5.** *The first derivative of the Laplacian matrix, $\frac{\partial \boldsymbol{L}(\sigma)}{\partial \sigma}$, is also a Laplacian matrix $\forall \sigma \in \mathbb{R}_{\geq 0}$.*

*Proof.* To establish that the derivative of the Laplacian matrix $\frac{\partial \boldsymbol{L}(\sigma)}{\partial \sigma}$ retains the properties of a Laplacian matrix, we must demonstrate that each diagonal entry is equivalent to the negative sum of the off-diagonal entries in its corresponding row and column (by the symmetry of the matrix).

We focus on showing this property for a single row without loss of generality, as the matrix's symmetry implies that this will hold for each row and column.

$$\frac{\partial \boldsymbol{L}_{j,j}(\sigma)}{\partial \sigma} = -\sum_{i \neq j} \frac{\partial \boldsymbol{A}_{j,i}(\sigma)}{\partial \sigma} \tag{19}$$

Given that $\frac{d}{d\sigma} K(\sigma^2) = 2\sigma \frac{d}{d\sigma^2}\left[e^{-\frac{d(x,y)}{\sigma^2}}\right] > 0 \; \forall \sigma, d(x,y) \in \mathbb{R}_{>0}$, we derive two crucial conclusions:

- The diagonal entries of $\frac{\partial \boldsymbol{L}(\sigma)}{\partial \sigma}$ are non-negative, resulting directly from the positive contributions of $\frac{\partial \boldsymbol{A}_{i,j}(\sigma)}{\partial \sigma}$ for $i \neq j$.

- The off-diagonal entries of $\frac{\partial \boldsymbol{L}(\sigma)}{\partial \sigma}$ are non-positive, as they correspond to $-\frac{\partial K(\sigma)}{\partial \sigma}$ which is negative $\forall \sigma, d(x,y) \in \mathbb{R}_{>0}$.

Together with equation equation 19, this confirms that the derivative $\frac{\partial \boldsymbol{L}(\sigma)}{\partial \sigma}$ behaves as a symmetric Laplacian matrix, whose formulation is mirrored in its transpose:

$$\frac{\partial \boldsymbol{L}(\sigma)}{\partial \sigma} = \frac{\partial \boldsymbol{L}^\top(\sigma)}{\partial \sigma} \tag{20}$$

$\square$

# D   Lemma's for the convexity of the $\sigma$ update

**Lemma 6.** *The second derivative of the Laplacian matrix $\left(\frac{\partial^2 \boldsymbol{L}(\sigma)}{\partial \sigma^2}\right)$ is a PSD matrix $\forall \sigma \in \Sigma = (0, \sqrt{6}/3]$.*

*Proof.* Initially once can easily observe that because $\boldsymbol{L}(\sigma) = \boldsymbol{L}^\top(\sigma)$ is symmetric its second derivative is also symmetric $\frac{\partial^2 \boldsymbol{L}(\sigma)}{\partial \sigma^2} = \frac{\partial^2 \boldsymbol{L}^\top(\sigma)}{\partial \sigma^2}$ (see Corollary 2). Since the domain of $\sigma \in \Sigma$, it follows that the derivative of $\frac{\partial^2 \boldsymbol{L}(\sigma)}{\partial \sigma^2}$ constitutes an alternate Laplacian matrix, (see Lemma 7).

Given the new Laplacian matrix (i.e., $\frac{\partial^2 \boldsymbol{L}(\sigma)}{\partial \sigma^2} = \frac{\partial^2 \boldsymbol{L}^\top(\sigma)}{\partial \sigma^2}$) is symmetric, its eigenvalues are real values (Strang, 2006). As any other Laplacian matrix, $\frac{\partial^2 \boldsymbol{L}(\sigma)}{\partial \sigma^2}$ can be written as a product of incidence matrix (i.e., $\boldsymbol{M}$) as:

$$\frac{\partial^2 \boldsymbol{L}(\sigma)}{\partial \sigma^2} = \boldsymbol{M}^\top \boldsymbol{M}.$$

Whenever the eigenvalues are real numbers, one can safely say that:

$$\forall \boldsymbol{x}, \boldsymbol{x}^\top \frac{\partial^2 \boldsymbol{L}(\sigma)}{\partial \sigma^2} \boldsymbol{x} = \rho \in \mathbb{R}_{\geq 0}.$$

Reusing:

$$\frac{\partial^2 \boldsymbol{L}(\sigma)}{\partial \sigma^2} = \boldsymbol{M}^\top \boldsymbol{M}$$

into

$$\boldsymbol{x}^\top \frac{\partial^2 \boldsymbol{L}(\sigma)}{\partial \sigma^2} \boldsymbol{x} = \boldsymbol{x}^\top \boldsymbol{M}^\top \boldsymbol{M} x = ||\boldsymbol{M}\boldsymbol{x}||^2 = \rho \rightarrow \rho \geq 0.$$

Hence, the new Laplacian matrix $\frac{\partial^2 \boldsymbol{L}(\sigma)}{\partial \sigma^2}$ is PSD. □

To prove Lemma 6 we need Lemma 7.

**Lemma 7.** *The second derivative of Laplacian matrix $\frac{\partial^2 \boldsymbol{L}(\sigma)}{\partial \sigma^2}$ is another Laplacian matrix $\forall \sigma \in \Sigma = (0, \sqrt{6}/3]$.*

*Proof.* To prove that $\frac{\partial^2 \boldsymbol{L}(\sigma)}{\partial \sigma^2}$ is another Laplacian matrix, one must initially demonstrate that each diagonal entry is the negative sum of the non-diagonal entries in its corresponding row (or equivalently, its column), owing to the symmetry of the matrix. Since the matrix is symmetric, we choose row-wise, equivalent to column-wise, and vice versa. Furthermore, it suffices to show for just one row; without losing generality, it is equivalent to the rest of the rows.

$$\frac{\partial^2 L_{j,j}(\sigma)}{\partial \sigma^2} = \frac{\partial^2 \sum_{i=0}^{N-1} A_{j,i}(\sigma)}{\partial \sigma^2} = \sum_{i=0}^{N-1} \frac{\partial^2 A_{i,j}(\sigma)}{\partial \sigma^2} \tag{21}$$

Given that the proposed kernel

$$\frac{\partial^2 K(\sigma)}{\partial \sigma^2} \geq 0, \forall d(x,y) \in \mathbb{R}_{>0}, \forall \sigma \in \Sigma = (0, \sqrt{6}/3]$$

(see Corollary 1), this results in two important implications:

- Firstly, the diagonal values for $\frac{\partial^2 L_{i,i}(\sigma)}{\partial \sigma^2}$ are always positive since:

$$\frac{\partial^2 A_{i,j|i\neq j}(\sigma)}{\partial \sigma^2} > 0, \forall i \in [0, N-1], \forall d(x,y) \in \mathbb{R}_{>0}, \forall \sigma \in \Sigma = (0, \sqrt{6}/3].$$

- Secondly, off-diagonal of entries of

$$\frac{\partial^2 L_{i,j|i\neq j}(\sigma)}{\partial \sigma^2} = -\frac{\partial^2 K(\sigma)}{\partial \sigma^2} < 0, \forall \sigma \in \Sigma = (0, \sqrt{6}/3], \forall d(x,y) \in \mathbb{R}_{>0}.$$

Hence, together with eq. (21), one can conclude that the new matrix is a symmetric Laplacian matrix (see eq. (22)).

$$\frac{\partial^2 \boldsymbol{L}(\sigma)}{\partial \sigma^2} = \frac{\partial^2 \boldsymbol{L}^\top(\sigma)}{\partial \sigma^2} \tag{22}$$

$\square$

To prove Lemma 7, we need Corollary 1.

**Corollary 1.** *The second derivative of the proposed kernel* $\frac{\partial^2 K(\sigma)}{\partial \sigma^2} \geq 0, \forall d(x,y) \in \mathbb{R}_{>0}, \forall \sigma \in \Sigma = (0, \sqrt{6}/3].$

*Proof.* To prove this in-equality $\frac{\partial^2 K(\sigma)}{\partial \sigma^2} \geq 0, \forall \sigma \in \Sigma$ we need to prove that:

$$\frac{\partial^2 K(\sigma)}{\partial \sigma^2} = 2e^{\frac{-d(x,y)}{\sigma^2}} \frac{d(x,y)}{\sigma^6} \left\{ 2d(x,y) - 3\sigma^2 \right\} \tag{23}$$

The first two terms in eq. (23) are always non-negative $\left( i.e., e^{\frac{-d(x,y)}{\sigma^2}} \geq 0, \frac{d(x,y)}{\sigma^6} \geq 0, \forall \sigma \in \Sigma, \forall d(x,y) \in R_{>0} \right)$.
However, the last term

$$2d(x,y) - 3\sigma^2 = 2(u(x,y)+1) - 3\sigma^2 = \underbrace{2 - 3\sigma^2}_{\frac{2}{3} \geq \sigma^2} + 2\underbrace{u(x,y)}_{\geq 0} \geq 0, \forall \|\sigma\| \leq \frac{\sqrt{6}}{3}, \forall u(x,y) \in R^+$$

is guaranteed to be non-negative. Hence the proof.

$\square$

**Corollary 2.** *The first and second derivative of Laplacian matrix* $\frac{\partial^2 \boldsymbol{L}(\sigma)}{\partial \sigma^2}$ *is a symmetric matrix.*

*Proof.* Starting by computing the derivative of the kernel:

$$\frac{\partial K(\sigma)}{\partial \sigma} = \frac{\partial e^{\frac{-d(x,y)}{\sigma^2}}}{\partial \sigma} = e^{\frac{-d(x,y)}{\sigma^2}} \frac{2d(x,y)}{\sigma^3} \tag{24}$$

As a result of symmetricity in eq. (24), the new matrix resulting from the differentiation of the Laplacian would still be another symmetricity matrix.

$$\frac{\partial \boldsymbol{L}(\sigma)}{(\partial \sigma)} = \left\{ \frac{\partial \boldsymbol{L}(\sigma)}{\partial \sigma} \right\}^\top = \frac{\partial \boldsymbol{L}^\top(\sigma)}{\partial \sigma} \tag{25}$$

and the second derivative is:

$$\frac{\partial^2 K(\sigma)}{\partial \sigma^2} = \frac{\partial}{\partial \sigma}\left\{\frac{\partial e^{\frac{-d(x,y)}{\sigma^2}}}{\partial \sigma}\right\}$$

$$= \frac{\partial}{\partial \sigma}\left\{e^{\frac{-d(x,y)}{\sigma^2}}\frac{2d(x,y)}{\sigma^3}\right\}$$

$$= e^{\frac{-d(x,y)}{\sigma^2}}\frac{\partial}{\partial \sigma}\left\{\frac{2d(x,y)}{\sigma^3}\right\} + \frac{2d(x,y)}{\sigma^3}\frac{\partial}{\partial \sigma}\left\{e^{\frac{-d(x,y)}{\sigma^2}}\right\}$$

$$= e^{\frac{-d(x,y)}{\sigma^2}}\left\{\frac{-3 \times 2d(x,y)}{\sigma^4}\right\} + e^{\frac{-d(x,y)}{\sigma^2}}\left\{\frac{2d(x,y)}{\sigma^3}\right\}^2$$

$$= e^{\frac{-d(x,y)}{\sigma^2}}\left\{\left(\frac{2d(x,y)}{\sigma^3}\right)^2 - \frac{3 \times 2d(x,y)}{\sigma^4}\right\}$$

$$= e^{\frac{-d(x,y)}{\sigma^2}}\left\{\frac{(2d(x,y))^2 - 3\sigma^2 2d(x,y)}{\sigma^6}\right\}$$

$$\frac{\partial^2 K(\sigma)}{\partial \sigma^2} = 2e^{\frac{-d(x,y)}{\sigma^2}}\frac{d(x,y)}{\sigma^6}\left\{2d(x,y) - 3\sigma^2\right\} \tag{26}$$

As a result of symmetricity eq. (26), the new matrix resulting from the second differentiation of the Laplacian would still be another symmetricity matrix:

$$\frac{\partial^2 \boldsymbol{L}(\sigma)}{(\partial \sigma^2)} = \left\{\frac{\partial^2 \boldsymbol{L}(\sigma)}{\partial \sigma^2}\right\}^\top = \frac{\partial^2 \boldsymbol{L}^\top(\sigma)}{\partial \sigma^2}.$$

$\square$

# E   Scaling of the affinity matrix

To illustrate the dependence of SC on relative distances, we note that uniformly increasing all distances in the Euclidean metric by a unit value corresponds to a linear scaling of the affinity matrix when employing the RBF kernel (see eq. (2)). Furthermore, this scaling of the affinity matrix modifies the eigenvalues of the Laplacian matrix while leaving its eigenvectors unchanged (see Lemma 8).

**Lemma 8.** *Given an affinity matrix* $\mathbf{A}$ *and its corresponding Laplacian matrix* $\mathbf{L}$*, scaling the affinity matrix* $\mathbf{A}_{scaled} = c\mathbf{A}$ *s.t* $c > 0$ *preserves the eigenvectors of the resulting Laplacian matrix* $\mathbf{L}_{scaled}$*, while the eigenvalues are simply scaled by* $c$*, i.e.,* $\lambda_{scaled} = c\lambda$*.*

*Proof.* Consider the eigenvector matrix $\mathbf{F}$ of $\mathbf{L}$ such that

$$\mathbf{F}^\top \mathbf{L} \mathbf{F} = \Lambda,$$

where $\Lambda$ is a diagonal matrix containing the eigenvalues $\lambda_{1,\dots,N}$. The same eigenvectors $\mathbf{F}$ can diagonalize $\mathbf{L}_{\text{scaled}}$ to:

$$\mathbf{F}^\top \mathbf{L}_{\text{scaled}} \mathbf{F} = c\Lambda.$$

Notice that the new degree matrix for $\mathbf{A}_{\text{scaled}}$ is $\mathbf{D}_{\text{scaled}} = c\mathbf{D}$, leading to:

$$\mathbf{L}_{\text{scaled}} = c\mathbf{D} - c\mathbf{A} = c(\mathbf{D} - \mathbf{A})c\mathbf{L}$$

Then we prove that the orthogonal matrix containing the eigenvectors $\mathbf{F}$ diagonalizes $\mathbf{L}_{\text{scaled}}$ to:

$$F^{\top}\mathbf{L}_{\text{scaled}}\mathbf{F} = \mathbf{F}^{\top}\{c\mathbf{L}\}\mathbf{F} = c\mathbf{F}^{\top}L\mathbf{F} = c\Lambda = \Lambda_{scaled}$$

$\square$

With the $\mathbf{F}^{\top}\mathbf{L}_{\text{scaled}}\mathbf{F} = \Lambda_{scaled}$, and the new scaled Laplacian being symmetric $\Lambda_{scaled} = \Lambda_{scaled}^{\top}$ it can be safely stated that $\mathbf{F}$ are the eigenvectors of $\mathbf{L}_{\text{scaled}}$ as well Strang (2006). Since the eigenvalues $\Lambda_{scaled} = c\Lambda$ are uniformly scaled, their order remains unchanged. Consequently, the order of the eigenvectors corresponding to the smallest eigenvalues also stays unchanged for both $\mathbf{L}_{\text{scaled}}$ and $L$, thereby maintaining consistent GC.

## F    Experiments on benchmark dataset

Herein, we present a series of sanity checks on popular benchmark clustering datasets (see Table 3). The purpose of these experimentations is to showcase the method's capability to separate clusters under different types of convexity. Apart from different types of separation, the proposed method also handles an asymmetric number of data per cluster as in fig. 8. For each tested dataset, we present the final eigen-embeddings, the loss (i.e., loss $= \sum_{i=1}^{K} \lambda_i$) over iterations together with the trajectory of the bandwidth $\sigma$ along with $grad_{\sigma}$ and $step_{\sigma}$. A visual representation of the eigen-embeddings illustrates how the method facilitates the linear separation of individual clusters.

Table 3: Description of the benchmark datasets.

| Dataset | Size | Dimensions | Clusters |
|---|---|---|---|
| Two spirals (see fig. 3) | 500 | 3 | 2 |
| Two rings (see fig. 5) | 500 | 3 | 2 |
| Three spirals (see fig. 6) | 312 | 2 | 3 |
| Two circles (see fig. 7) | 500 | 2 | 2 |
| Two (asymmetric) half moons (see fig. 7) | 373 | 2 | 2 |
| Two (symmetric) half moons (see fig. 8) | 500 | 2 | 2 |

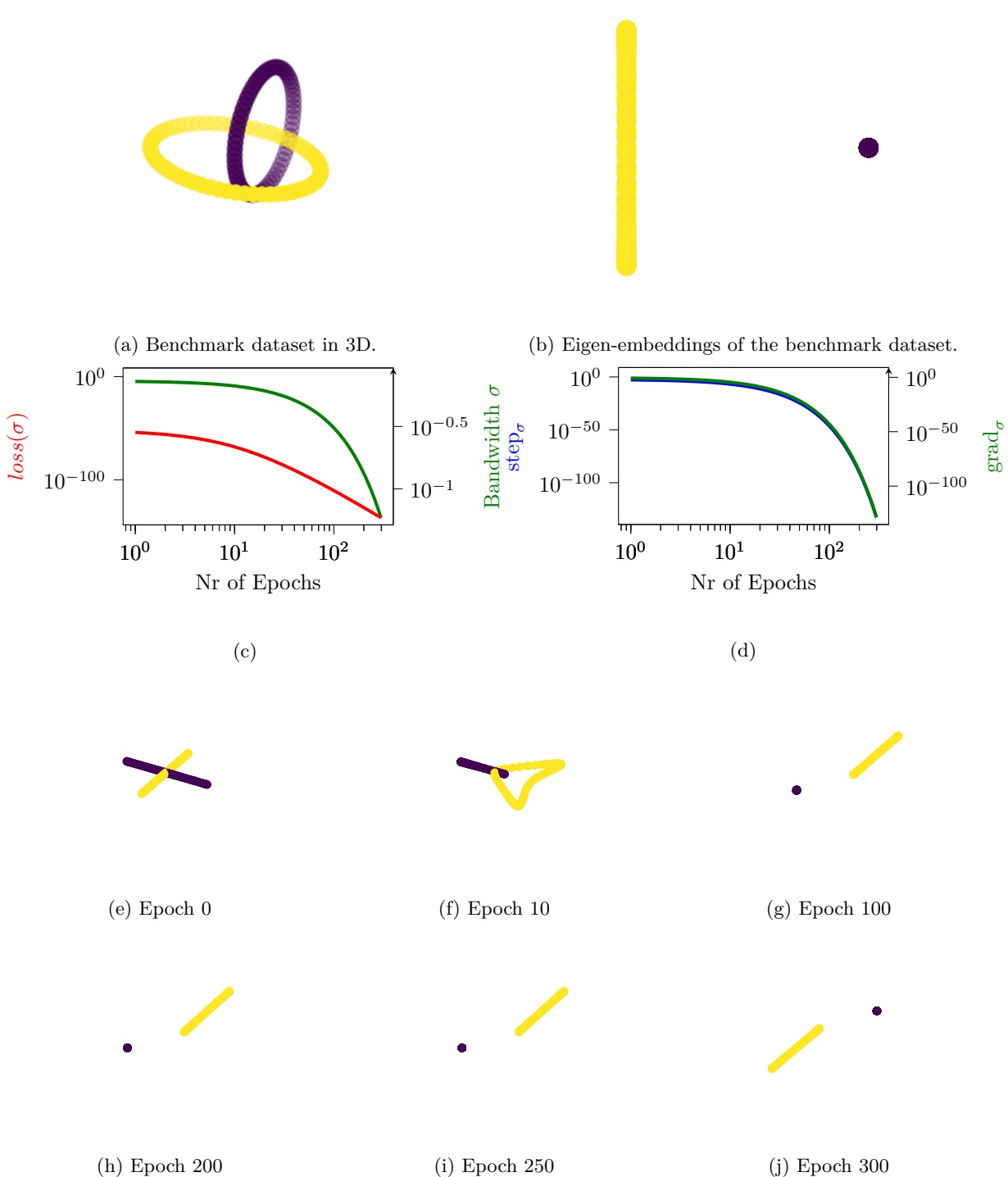

(a) Benchmark dataset in 3D.   (b) Eigen-embeddings of the benchmark dataset.

(c)                (d)

(e) Epoch 0     (f) Epoch 10     (g) Epoch 100

(h) Epoch 200     (i) Epoch 250     (j) Epoch 300

Figure 5: Benchmark dataset two rings in 3D (see fig. 5a). Despite the shape not being convex, the method learns a linear separation of these two rings (see fig. 5b). The loss (loss $= \sum_{i=1}^{K} \lambda_i$) along with the bandwidth $\sigma$, decreases consistently with the number of iterations (see fig. 5c). Furthermore, the $\text{grad}_\sigma$ and the $\text{step}_\sigma$ decreases consistently with the number of iterations (see fig. 5d). A snapshot of the optimized trajectory is in figs. 5e to 5j.

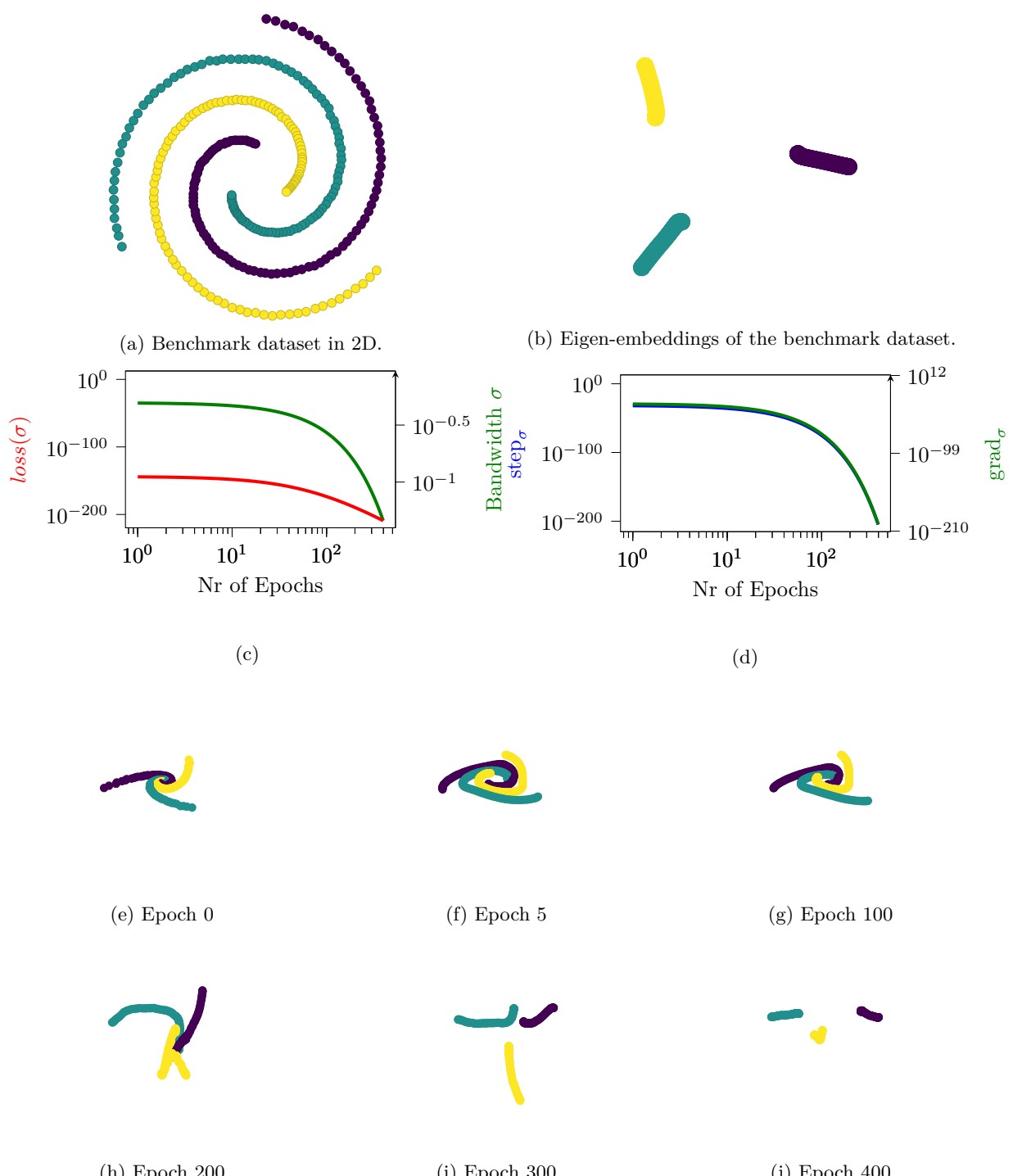

(a) Benchmark dataset in 2D.

(b) Eigen-embeddings of the benchmark dataset.

(c)

(d)

(e) Epoch 0

(f) Epoch 5

(g) Epoch 100

(h) Epoch 200

(i) Epoch 300

(j) Epoch 400

Figure 6: Benchmark dataset three spirals in 2D (see fig. 6a). Despite the shape not being convex, the method learns a linear separation of these three spirals (see fig. 6b). The loss (loss $= \sum_{i=1}^{K} \lambda_i$) along with the bandwidth $\sigma$, decreases consistently with the number of iterations (see fig. 6c). Furthermore, the $\text{grad}_\sigma$ and the $\text{step}_\sigma$ decreases consistently with the number of iterations (see fig. 6d). A snapshot of the optimized trajectory is in figs. 6e to 6i.

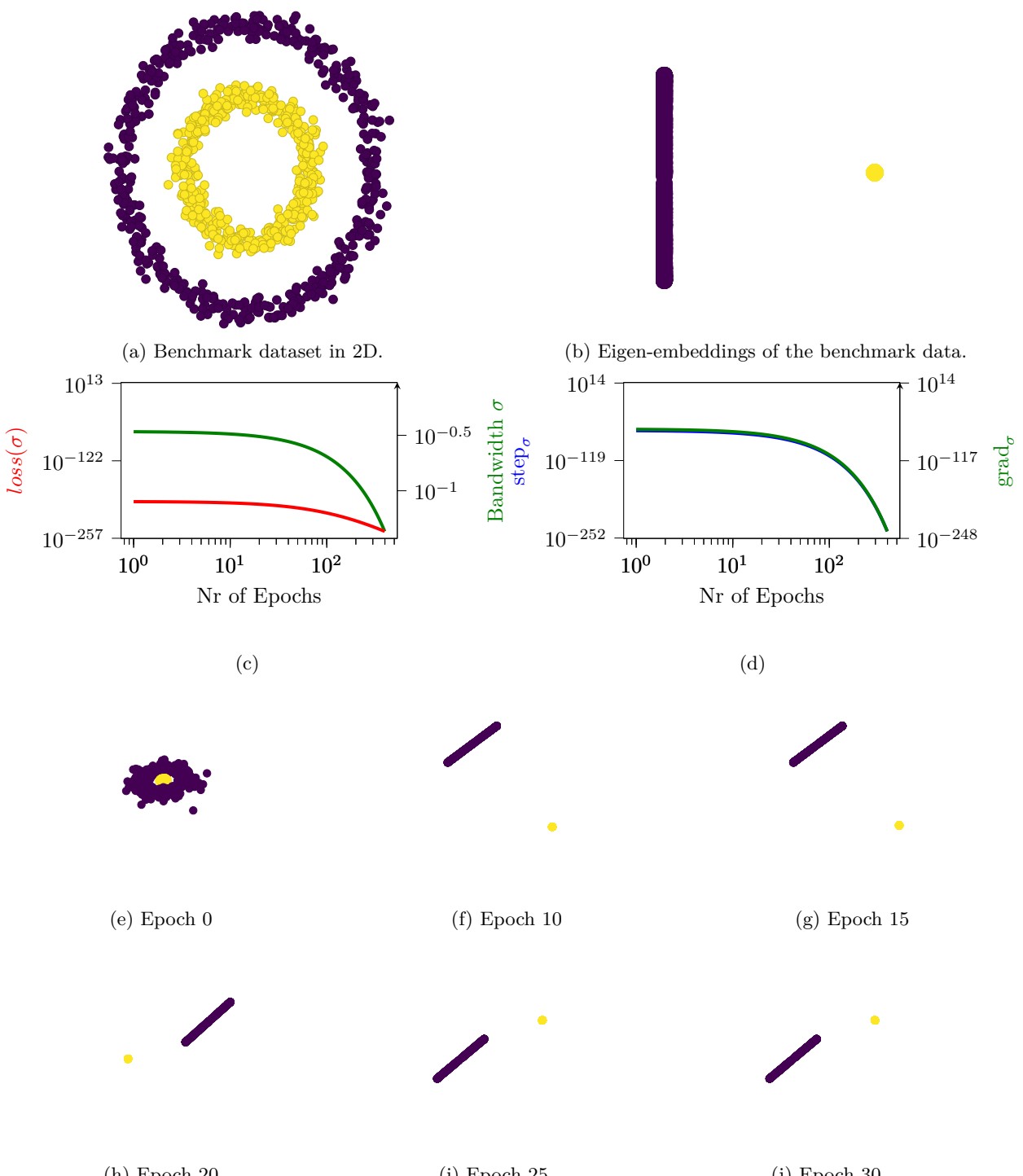

(a) Benchmark dataset in 2D.

(b) Eigen-embeddings of the benchmark data.

(c)

(d)

(e) Epoch 0

(f) Epoch 10

(g) Epoch 15

(h) Epoch 20

(i) Epoch 25

(j) Epoch 30

Figure 7: Benchmark dataset two rings in 2D (see fig. 7a). Despite the shape not being convex, the method learns a linear separation of these two rings (see fig. 7b). The loss (loss $= \sum_{i=1}^{K} \lambda_i$) along with the bandwidth $\sigma$, decreases consistently with the number of iterations (see fig. 7c). Furthermore, the $\mathrm{grad}_\sigma$ and the $\mathrm{step}_\sigma$ decreases consistently with the number of iterations (see fig. 7d). A snapshot of the optimized trajectory is in figs. 7e to 7j.

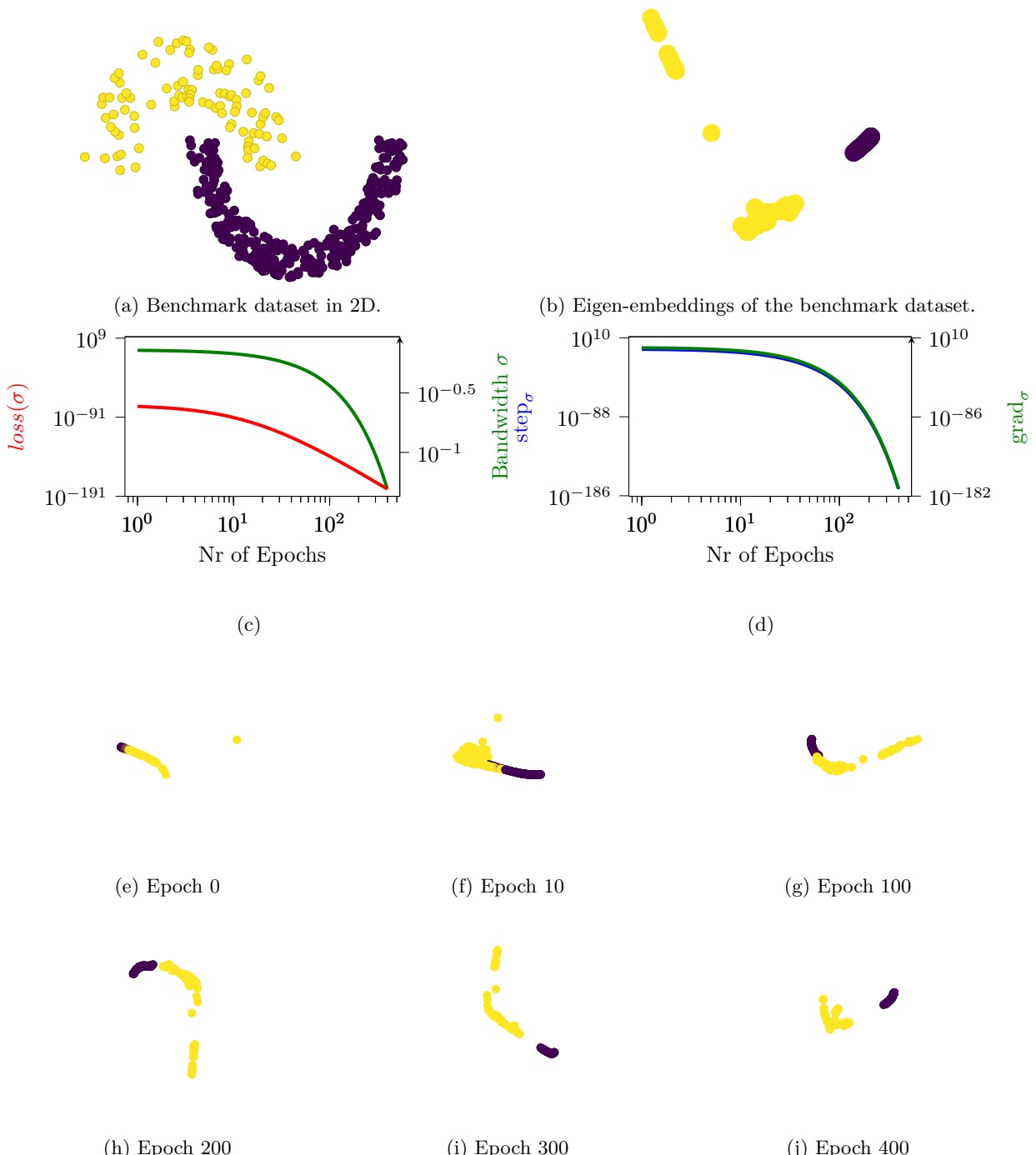

Figure 8: Benchmark dataset two (asymmetric) half moons in 2D (see fig. 8a). Despite the shape not being convex, the method learns a linear separation of these two half moons (see fig. 8b). The loss (loss $= \sum_{i=1}^{K} \lambda_i$) along with the bandwidth $\sigma$, decreases consistently with the number of iterations (see fig. 8c). Furthermore, the $\text{grad}_\sigma$ and the $\text{step}_\sigma$ decreases consistently with the number of iterations (see fig. 8d). A snapshot of the optimized trajectory is in figs. 8e to 8j.

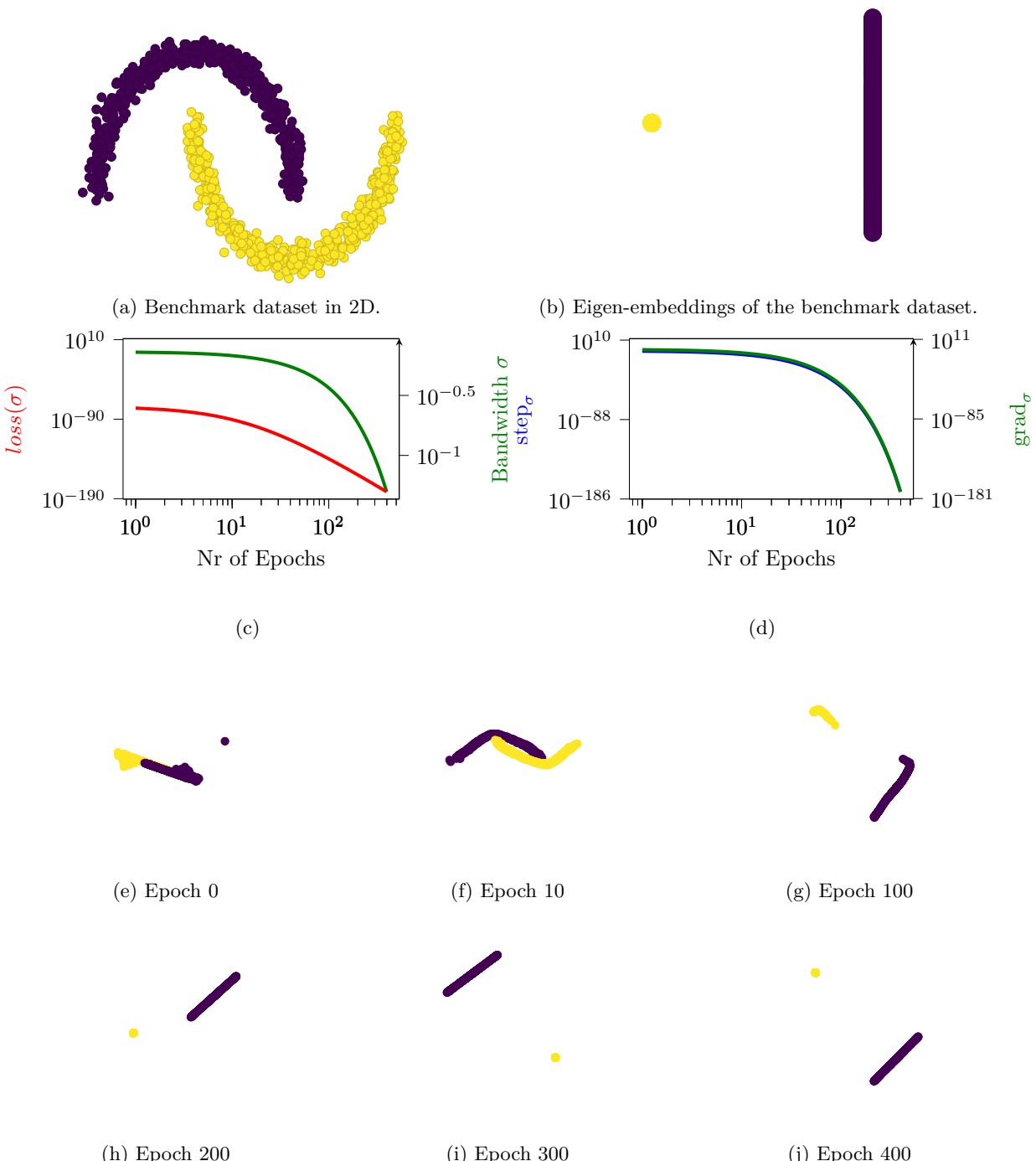

(a) Benchmark dataset in 2D.

(b) Eigen-embeddings of the benchmark dataset.

(c)

(d)

(e) Epoch 0

(f) Epoch 10

(g) Epoch 100

(h) Epoch 200

(i) Epoch 300

(j) Epoch 400

Figure 9: Benchmark dataset two moons in 2D (see fig. 9a). Despite the shape not being convex, the method learns a linear separation of these two moons (see fig. 9b). The loss (loss $= \sum_{i=1}^{K} \lambda_i$) along with the bandwidth $\sigma$, decreases consistently with the number of iterations (see fig. 9c). Furthermore, the $\text{grad}_\sigma$ and the $\text{step}_\sigma$ decreases consistently with the number of iterations (see fig. 9d). A snapshot of the optimized trajectory is in figs. 9e to 9j.

## G   Additional experiments with outlier exposure

In our method, we strategically reduce the bandwidth $\sigma$ towards zero, enabling SC to adhere to the maximum margin principle (Hofmeyr, 2020). This strategy, however, encounters a challenge when isolated data points, distant from the main data body, form their "singleton" clusters (Hofmeyr, 2020). Since the eigenvectors enable a bi-partition on the dataset with a GC cost indicated by its corresponding eigenvalues (von Luxburg, 2007), the singletons, which are notably detached from the primary dataset, can be effectively separated using a GC that harnesses the eigenvector corresponding to the smallest eigenvalue (Hofmeyr, 2020). This separation mechanism is scalable to multiple outliers, allowing their separation through the initial set of eigenvectors equivalent in number to the outliers (see Algorithm 2). Given that, these singleton occurrences can be easily identified by sequentially applying GC on the foremost eigenvector, the count of these vectors directly represents the number of singletons. Consequently, once we have identified the singletons, we utilize the subsequent eigenvectors to partition the remainder of the dataset into non-singleton clusters. Furthermore, we experimented on different datasets containing outliers and showcased that method's ability to perform well.

To demonstrate the method's capability of managing isolated data points, we augmented each benchmark dataset in table 3 (Fränti & Sieranoja) by adding two isolated points (see figs. 10 to 15). During our experiments with these enhanced datasets, we modified the loss function to include additional eigenvalues equivalent to the number of isolated points (i.e., loss = $\sum_{i=1}^{K+Nr_{Singletons}} \lambda_i$ see Algorithm 2). For visualization, we omitted the first two eigenvectors since they only separate the singletons and are ineffective in separating the spiral patterns, focusing instead on the subsequent eigenvectors (see figs. 10 to 15).

---

**Algorithm 2** Similarity Learning with Singleton Detection

---

1: **procedure** SINGLETONS(Data $\mathbf{X}^{[N,D]}$, Number of Clusters $K$)
2:      $\sigma, f_{1,\dots,K} \leftarrow$ SIMILARITYLEARNING($\mathbf{X}^{[N,D]}, K$)         ▷ Compute initial cluster partition
3:      $Nr_{Singletons} \leftarrow 0$                                                  ▷ Set the singleton counter to zero
4:      **for** $i = 1$ to $K$ **do**
5:          $u_{\geq 0} \leftarrow f_i \geq 0$          ▷ Data partition with non-negative entries of the $i$-th eigenvector
6:          $u_{<0} \leftarrow f_i < 0$              ▷ Data partition with negative entries of the $i$-th eigenvector
7:          **if** len($u_{\geq 0}$) == 1 or len($u_{<0}$) == 1 **then**         ▷ Check for singletons in both partitions
8:              $Nr_{Singletons} \leftarrow Nr_{Singletons} + 1$
9:          **else**
10:             break[4]                             ▷ Stop if no singletons found in this eigenvector
11:     **if** $Nr_{Singletons} == K$ **then**                          ▷ All partitions contain singletons
12:         **return** SINGLETONS($\mathbf{X}^{[N,D]}, K + Nr_{Singletons}$)        ▷ Recurse with increased cluster count
13:     **else**
14:         **return** SIMILARITYLEARNING($\mathbf{X}^{[N,D]}, K + Nr_{Singletons}$) ▷ Adjust number of clusters and re-run

---

Considering that the eigenvectors are arranged in order of increasing eigenvalues, it follows logically that if no singleton is observed in the partition corresponding to the given eigenvector, then it is assured that no subsequent eigenvector partitions will contain a singleton either. Hence, breaking the looping would guarantee a correct numbering of singeltons.

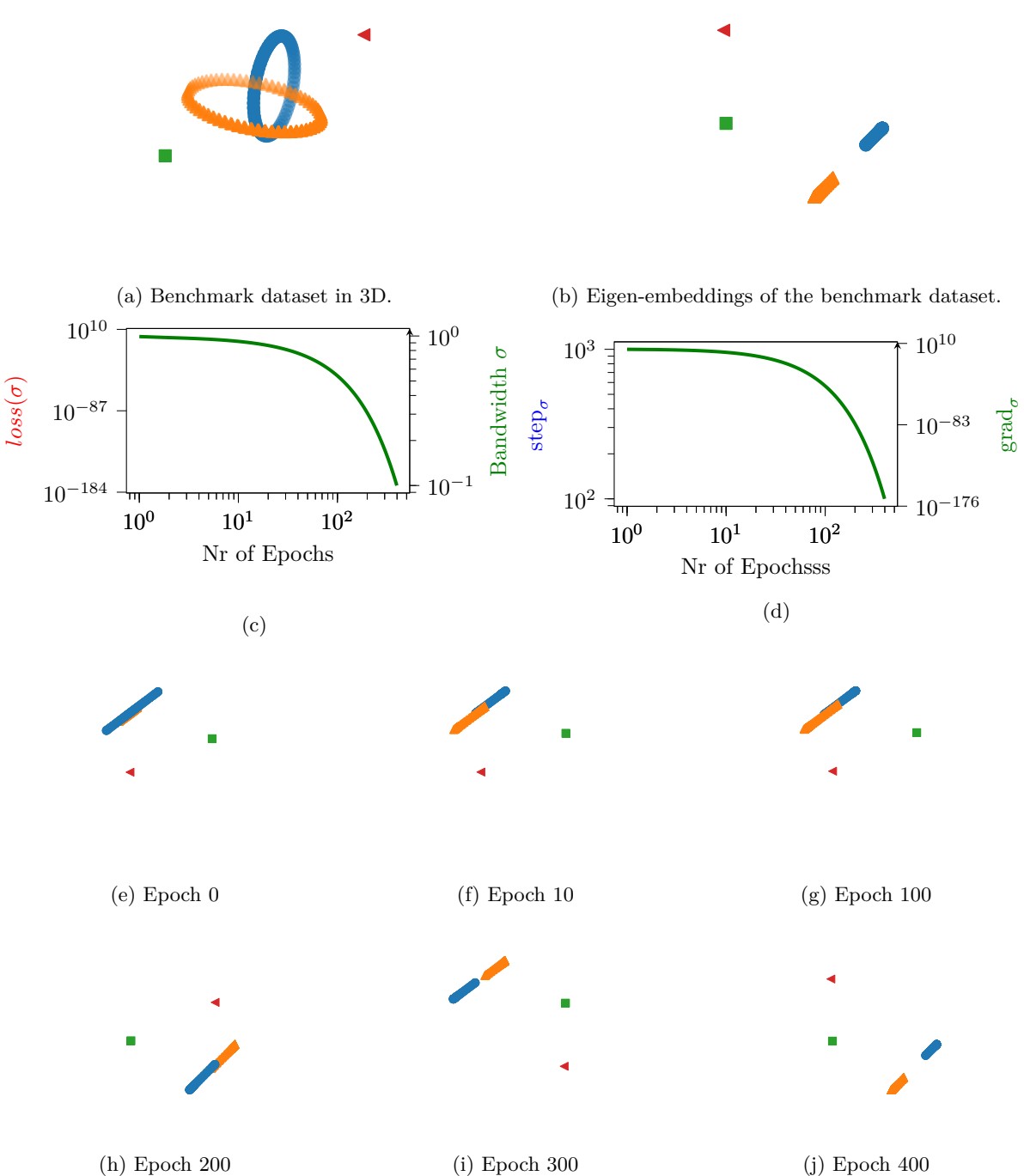

Figure 10: Benchmark dataset two rings in 3D (see fig. 10a). Despite the presence of the outliers and the shape not being convex, the method learns a linear separation of these two rings (see fig. 10b). The loss (loss = $\sum_{i=1}^{K} \lambda_i$) along with the bandwidth $\sigma$, decreases consistently with the number of iterations (see fig. 10c). The loss (loss = $\sum_{i=1}^{K} \lambda_i$) along with the bandwidth $\sigma$, decreases consistently with the number of iterations (see Optimizing kernel bandwidth $\sigma$ using the gradient descent stabilizes as the $\sigma$ gradient diminishes (see fig. 10d). A snapshot of the optimized trajectory is in figs. 10e to 10j, where the first two eigenvectors (corresponding to the outliers) are omitted, and the subsequent ones are visualized instead.

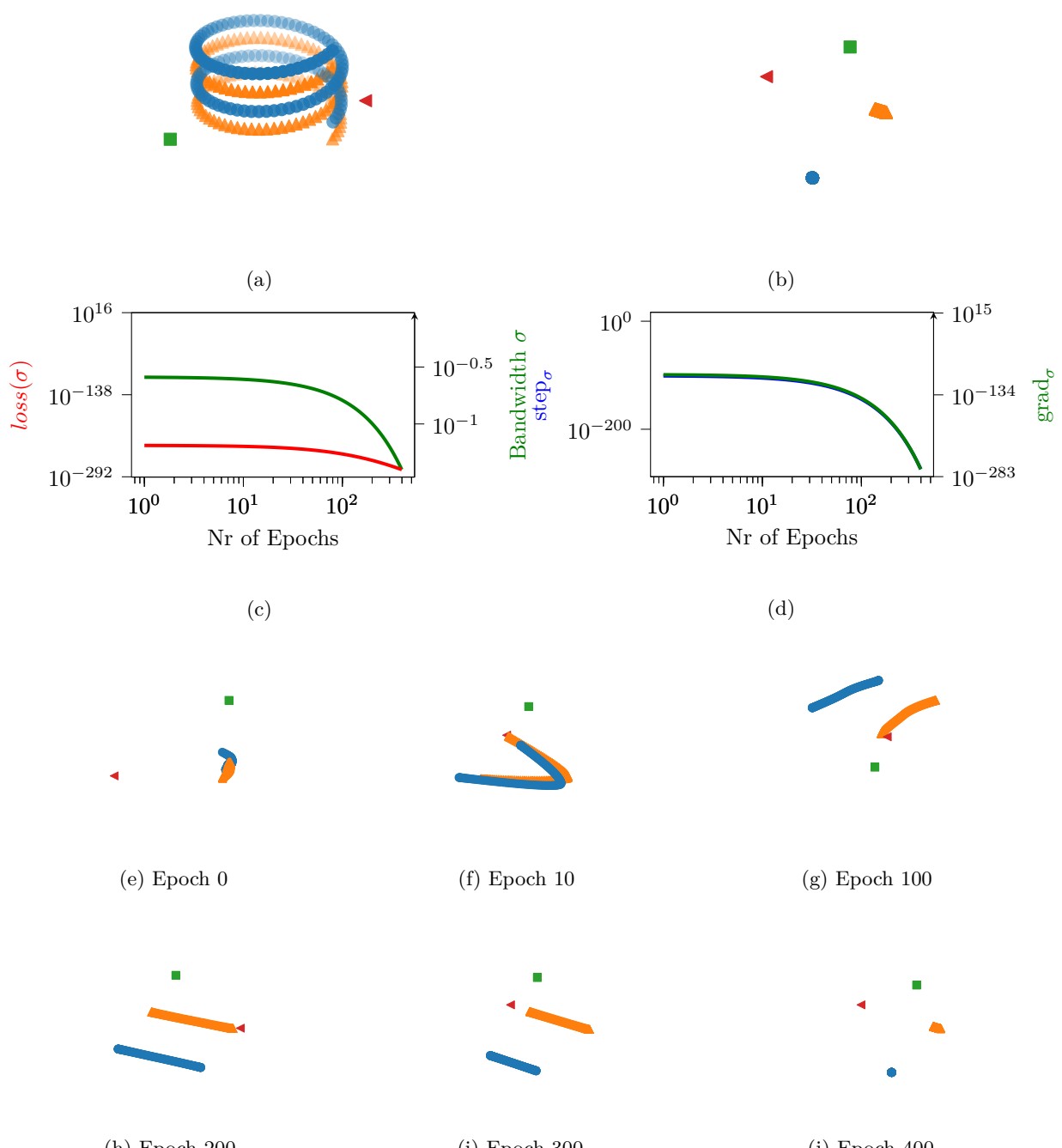

Figure 11: Our methodology successfully differentiates the two intertwined spirals despite the presence of two outliers, as depicted in fig. 11a, by learning a linear separation demonstrated in fig. 11b. The eigen-embedding of the benchmark data, shown in fig. 11c, reveals the intrinsic structure captured by our approach. The progression of the loss function (loss $= \sum_{i=1}^{K} \lambda_i$) over training iterations is depicted in fig. 11b, illustrating the convergence pattern. Our optimization process includes tuning the kernel bandwidth $\sigma$ via gradient descent, as shown in fig. 11c. Furthermore, a series of snapshots portraying the optimized separation trajectory can be seen in figs. 11e to 11h and 11j, where the first two eigenvectors (corresponding to the outliers) are omitted, and the subsequent ones are visualized instead.

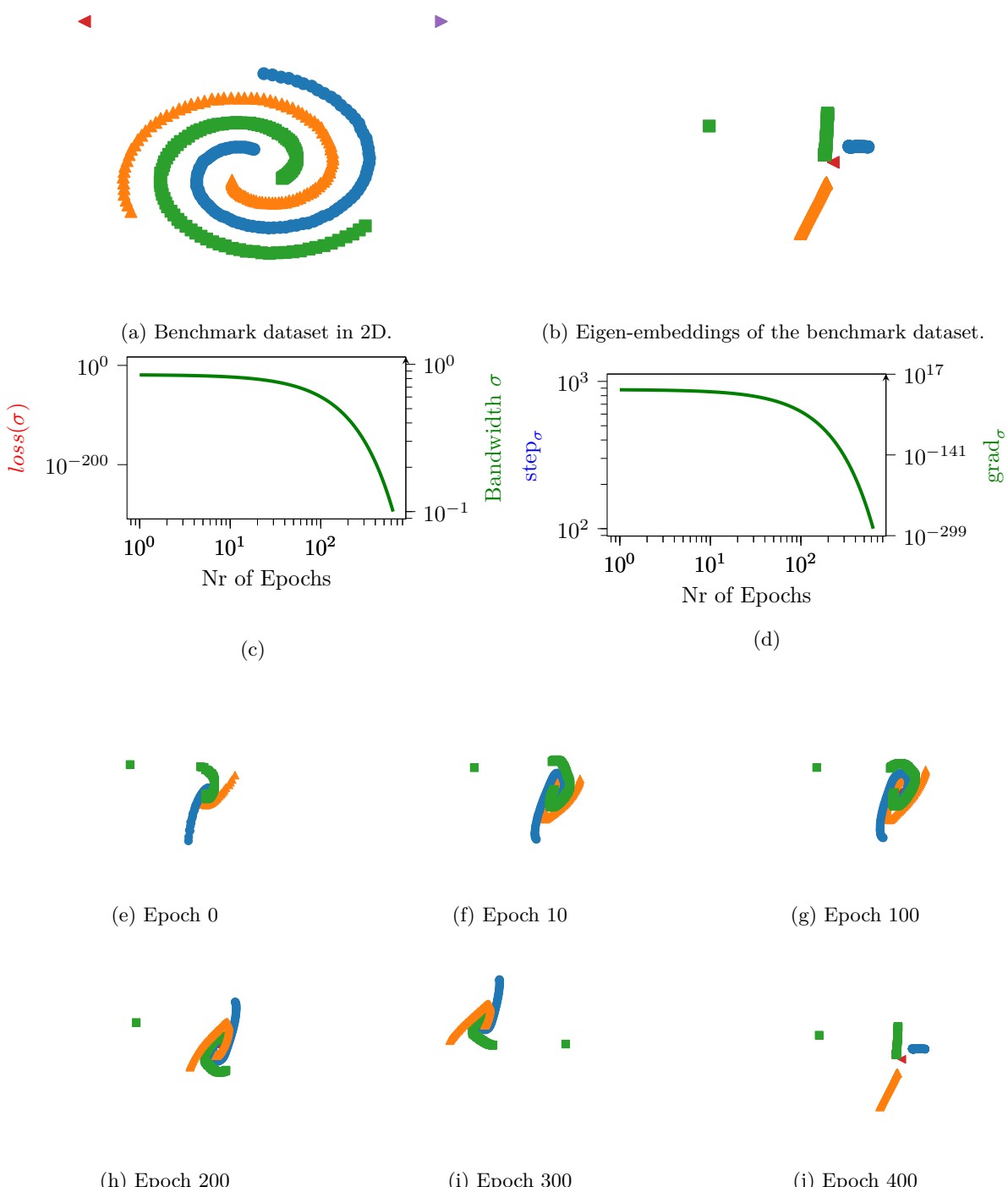

Figure 12: Benchmark dataset three spirals in 2D (see fig. 12a). Despite the presence of the outliers and the shape not being convex, the method learns a linear separation of these three spirals (see fig. 12b). The loss (loss $= \sum_{i=1}^{K} \lambda_i$) along with the bandwidth $\sigma$, decreases consistently with the number of iterations (see fig. 12c). Optimizing kernel bandwidth $\sigma$ using the gradient descent stabilizes as the $\sigma$ gradient diminishes (see fig. 12d). A snapshot of the optimized trajectory is in figs. 12e to 12i, where the first two eigenvectors (corresponding to the outliers) are omitted, and the subsequent ones are visualized instead.

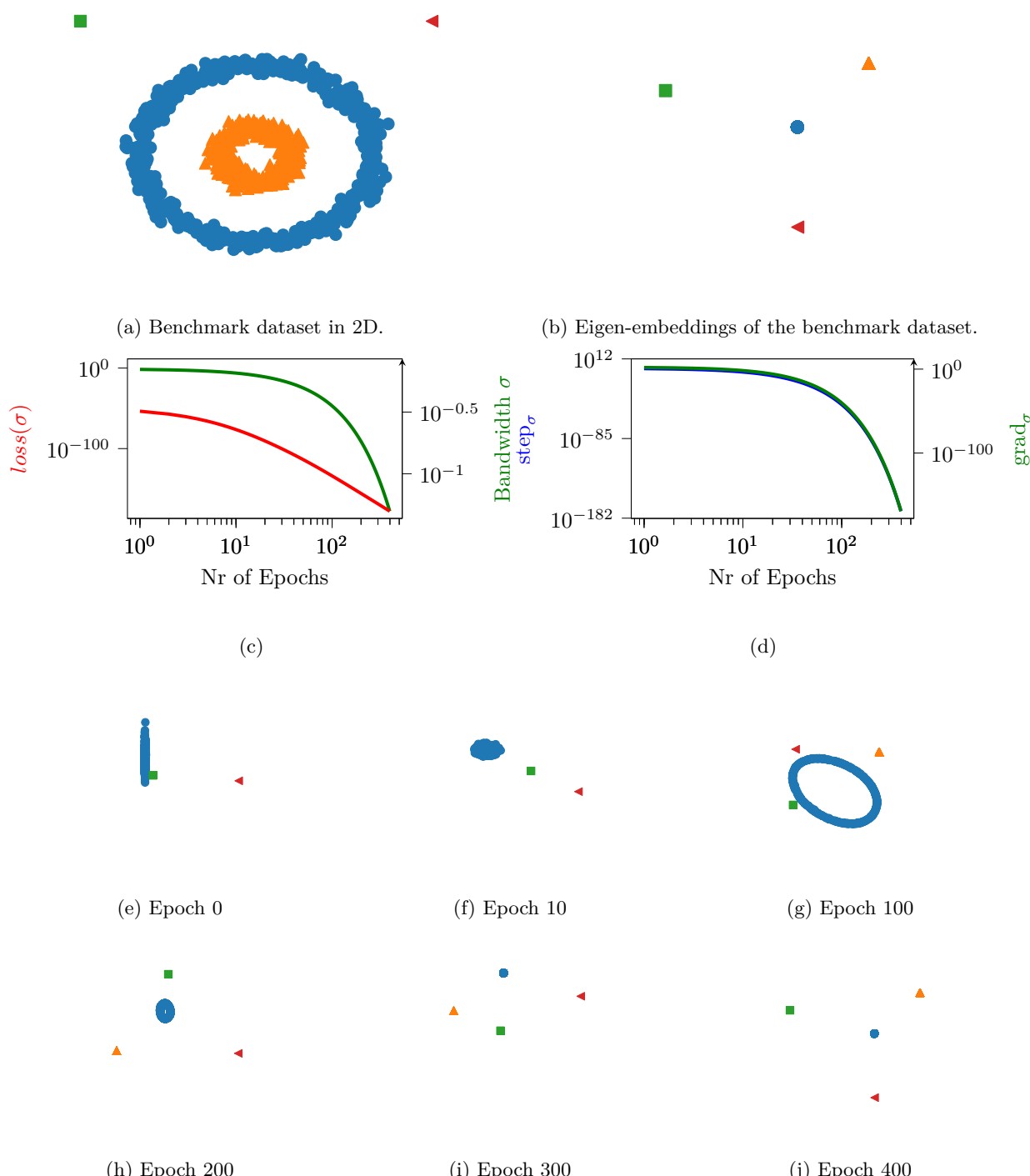

Figure 13: Benchmark dataset two moons in 2D (see fig. 13a). Despite the presence of the outliers and the shape not being convex, the method learns a linear separation of these two moons (see fig. 13b). The loss (loss $= \sum_{i=1}^{K} \lambda_i$) along with the bandwidth $\sigma$, decreases consistently with the number of iterations (see fig. 13c). Furthermore, the $\text{grad}_\sigma$ and the $\text{step}_\sigma$ decreases consistently with the number of iterations (see fig. 13d). A snapshot of the optimized trajectory is in figs. 13e to 13j, where the first two eigenvectors (corresponding to the outliers) are omitted, and the subsequent ones are visualized instead.

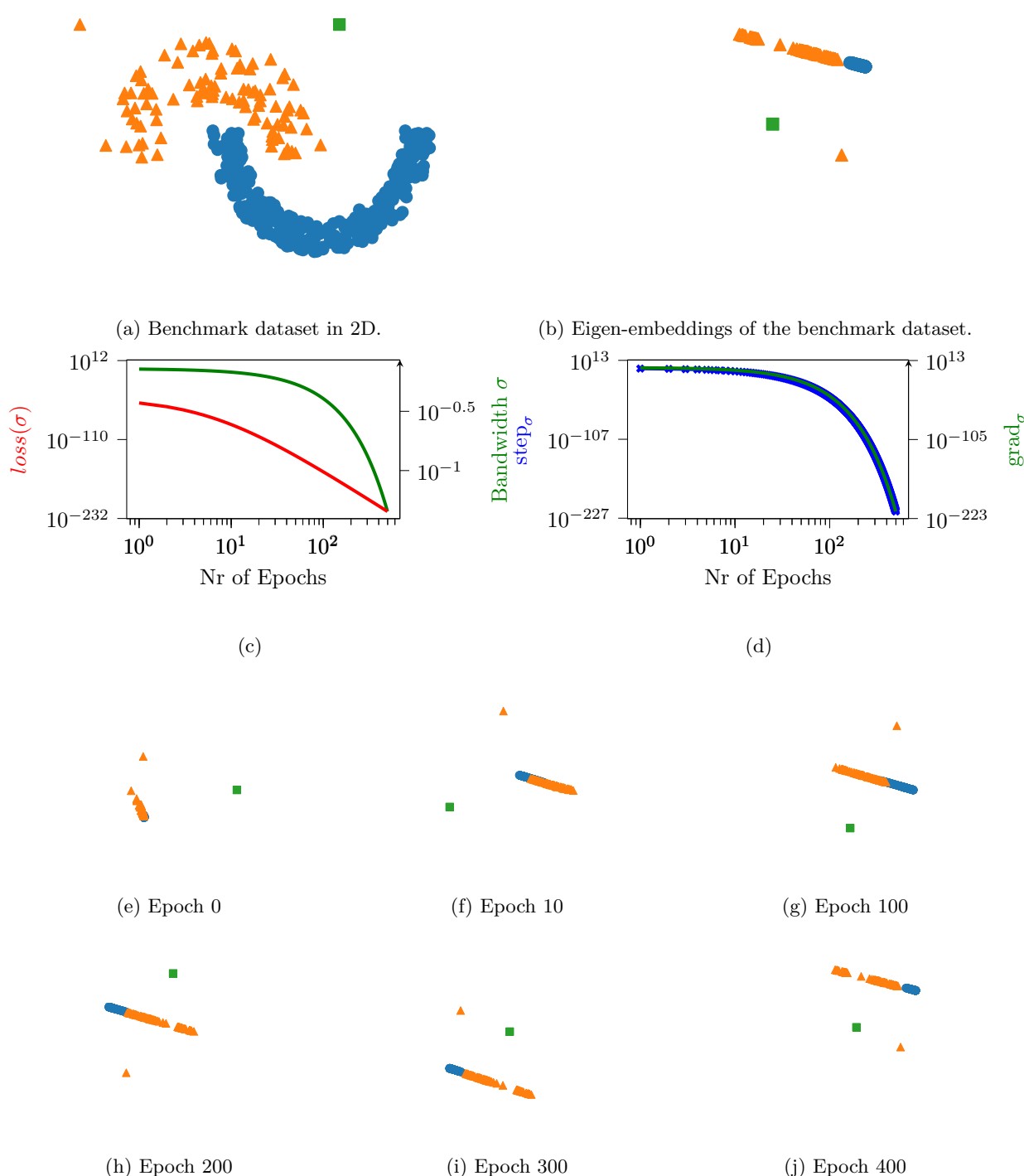

Figure 14: Benchmark dataset two (asymmetric) half moons in 2D (see fig. 14a). Despite the presence of the outliers and the shape not being convex, the method learns a linear separation of these two half moons (see fig. 14b). The loss (loss $= \sum_{i=1}^{K} \lambda_i$) along with the bandwidth $\sigma$, decreases consistently with the number of iterations (see fig. 14c). Furthermore, the $\text{grad}_\sigma$ and the $\text{step}_\sigma$ decreases consistently with the number of iterations (see fig. 14d). A snapshot of the optimized trajectory is in figs. 14e to 14j, where the first two eigenvectors (corresponding to the outliers) are omitted, and the subsequent ones are visualized instead.

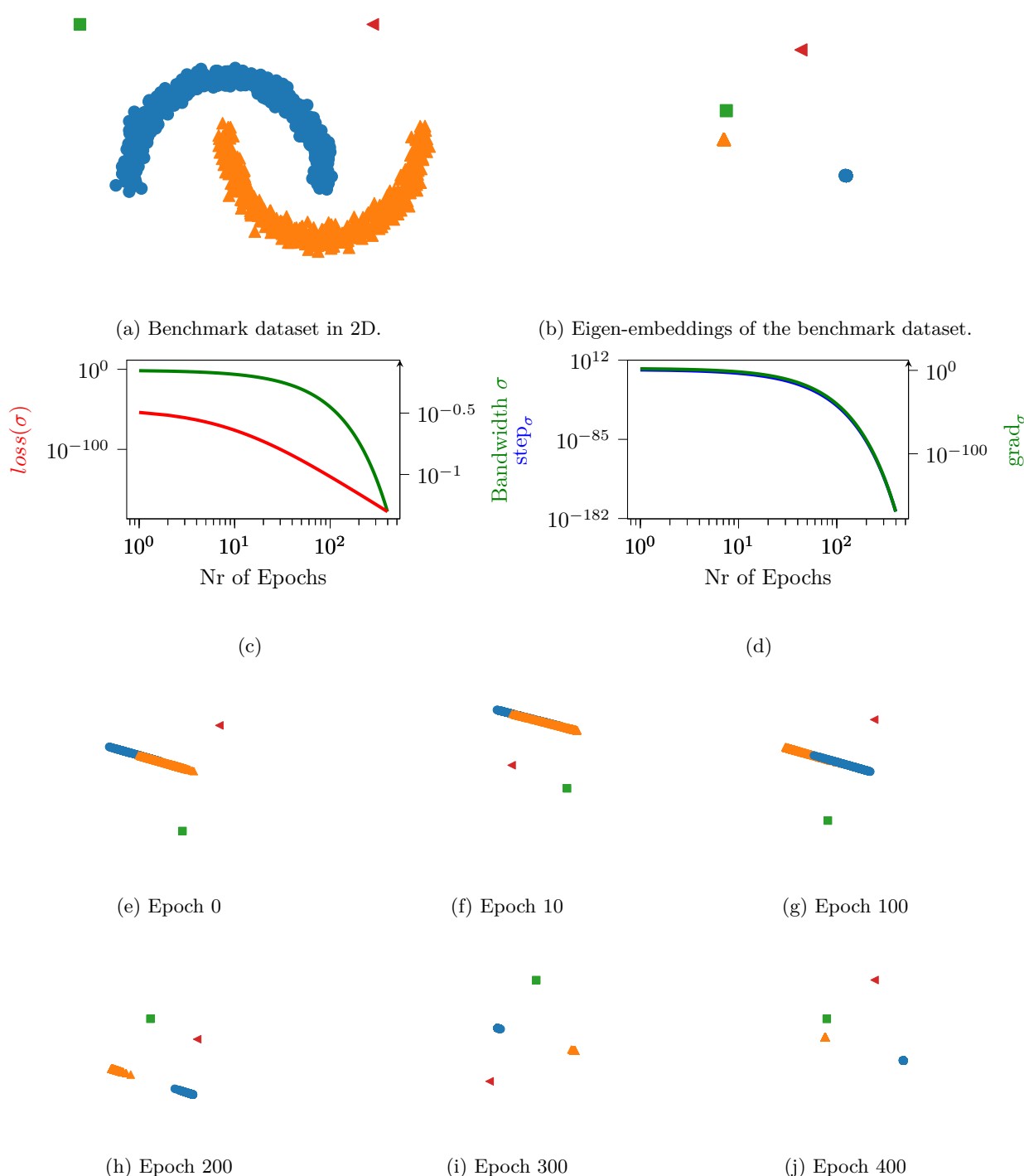

Figure 15: Benchmark dataset two moons in 2D (see fig. 15a). Despite the presence of the outliers and the shape not being convex, the method learns a linear separation of these two moons (see fig. 15b). The loss (loss = $\sum_{i=1}^{K} \lambda_i$) along with the bandwidth $\sigma$, decreases consistently with the number of iterations (see fig. 15c). Furthermore, the $\text{grad}_\sigma$ and the $\text{step}_\sigma$ decreases consistently with the number of iterations (see fig. 15d). A snapshot of the optimized trajectory is in figs. 15e to 15j, where the first two eigenvectors (corresponding to the outliers) are omitted, and the subsequent ones are visualized instead.

# H  Additional experiments with traditional clustering techniques

Unlike our proposed method, *the competing ones have a hyperparameter(s) dependency that cannot be objectively tuned.* Hence, for a fairer comparison with our method, we report the performance of the competing method as the average across a range of hyperparameters. Moreover, we also report the maximal performance figures of the alternative methods (see table 5). Two different kernels of choice have been tested across 12 different hyperparameter combinations. RBF kernel across different bandwidth ($\{0.01, 0.05, 0.1, 1, 10, 50, 100\}$) along with a polynomial kernels $K(x, y) = (a + x^\top y)^b$ with four different hyperparameter values (i.e., $a = \{0, 1\}$ and $b = \{2, 4\}$). The proposed method is entirely reproducible; therefore, we run just one experiment per dataset and report just one evaluation metric.

**Competing methods:** In table 5, we assess the effectiveness of the proposed method in a clustering scenario. Therefore we employ an SC setup in which the spectral embeddings derived from the acquired similarity matrix exhibit better performance compared to standard clustering methods. As competing methods, we utilize a range of clustering methods that incorporate kernel tricks with various hyperparameters, as well as kernel-free similarity learning methods.

Kernel K-means (KKM) (Dhillon et al., 2004) can perform the mean computation in the kernel reproducible space. This method uses a single kernel and demands arbitrary hyperparameter settings that must be set manually. Robust Kernel K-Means (RKKM) (Du et al., 2015) tries to attain maximum clustering performance by combining multiple kernels on a K-Means setup. Multiple kernel k-means (MKKM) (Huang et al., 2012b) is similar to KKM while aggregating multiple kernels to attain a compounding effect in the final clustering performance.

Spectral Clustering (SC) (Ng et al., 2001) is the first method in the evaluation. Since our proposal is essentially built on top of SC, we can show that it is possible to optimize the RBF kernel, given the data. Therefore, vanilla SC serves as the baseline for the proposed method. Affinity aggregation spectral clustering (AASC) algorithm (Huang et al., 2012a) tries to consolidate multiple affinities matrix to refine the clustering result. Twin Learning for Similarity and Clustering (TLSC) (Kang et al., 2017) is another approach that does not require any kernel function but directly learns the similarity matrix and the clustering indexing. Similarity Learning via Kernel Preserving Embeddings (SLKE) (Kang et al., 2019) is similar to TLSC, which does not learn clustering indicator vectors but solely the similarity matrix. The SLKE algorithm has been experimented with in two distinct regularizations: Sparse SLKE (SLKE-S) and Low-ranked SLKE (SLKE-R). SC is performed on top of the learned similarity matrix via SLKE.

**Dataset description:** The datasets described in Table 4 are widely utilized to assess clustering performance as no registration is needed upfront. The first three, JAFFE (Lyons et al., 1998), YALE (of Yale, 1997), ORL (Cambridge, 1994), contain human faces obtained at different illumination conditions or different facial expressions or with and without glasses. The last image dataset, COIL20 (Nene et al., 1996)) contains images of toys acquired at different orientations. While the rest of the data (BA, TR11, TR41, TR45) are text corpus (TR).

For each tested dataset, we present the loss (loss $= \sum_{i=1}^{K} \lambda_i$) together with the trajectory of the bandwidth $\sigma$ over iterations (see figs. 16a, 17a, 18a, 19a, 20a, 21a, 22a and 23a) along with $grad_\sigma$ and $step_\sigma$ (see figs. 16b, 17b, 18b, 19b, 20b, 21b, 22b and 23b).

Table 4: Description of the experimented datasets.

| Dataset | Size | Dimensions | Clusters |
|---------|------|------------|----------|
| YALE | 165 | 1024 | 15 |
| JAFFE | 213 | 676 | 10 |
| ORL | 400 | 1024 | 40 |
| COIL20 | 1440 | 1024 | 20 |
| BA | 1404 | 320 | 36 |
| TR11 | 414 | 6429 | 9 |
| TR41 | 878 | 7454 | 10 |
| TR45 | 690 | 8261 | 10 |

Table 5: ACC and NMI over the different datasets. The average performance is outside the brackets, while the highest performance from multiple runs is in the bracket for each method. MKKM and AASC results are an aggregation of multiple realizations.

| | | | | ACC | | | | | |
|------|------|------|------|------|------|------|------|------|------|
| Data | SC | KKM | MKKM | RKKM | AASC | TLSC | SLKE-S | SLKE-R | Ours |
| YALE | 40.53(49.42) | 38.97(47.12) | 45.70 | 39.71(48.09) | 40.64 | 45.35(56.96) | 38.89(61.82) | 51.28(66.24) | **52.62** |
| JAFFE | 54.03(74.88) | 67.09(74.39) | 74.55 | 67.89(75.61) | 30.35 | 86.64(99.83) | 70.77(96.71) | **90.89**(99.85) | 86.85 |
| ORL | 46.65(58.96) | 45.93(53.53) | 47.51 | 46.88(54.96) | 27.20 | 50.50(62.35) | 45.32(77.00) | 59.00(74.75) | **67.75** |
| COIL20 | 43.65(67.60) | 50.74(59.49) | 54.82 | 51.89(61.64) | 34.87 | 38.03(72.71) | 56.83(75.42) | 65.55(84.03) | **69.79** |
| BA | 26.25(31.07) | 33.66(41.20) | 40.52 | 34.35(42.17) | 27.07 | 39.50(47.72) | 36.35(50.74) | 35.79(44.37) | **50.21** |
| TR11 | 43.32(50.98) | 44.65(51.91) | 50.13 | 45.04(53.03) | 47.15 | 54.79(71.26) | 46.87(69.32) | **55.07**(74.64) | 50.96 |
| TR41 | 44.80(63.52) | 46.34(55.64) | **56.10** | 46.80(56.76) | 45.90 | 43.18(65.60) | 47.91(71.19) | 53.51(74.37) | 55.69 |
| TR45 | 45.96(57.39) | 45.58(58.79) | 58.46 | 45.69(58.13) | 52.64 | 53.38(74.02) | 50.59(78.55) | 58.37(79.89) | **63.32** |
| | | | | NMI | | | | | |
| YALE | 44.79(52.92) | 42.07(51.34) | 50.06 | 42.87(52.29) | 46.83 | 45.07(56.50) | 40.38(59.47) | 52.87(64.29) | **64.01** |
| JAFFE | 59.35(82.08) | 71.48(80.13) | 79.79 | 74.01(83.47) | 27.22 | 84.67(99.35) | 60.83(94.80) | 81.56(99.49) | **93.34** |
| ORL | 66.74(75.16) | 63.36(73.43) | 68.86 | 63.91(74.23) | 43.77 | 63.55(78.96) | 58.84(86.35) | 75.34(85.15) | **86.30** |
| COIL20 | 54.34(80.98) | 63.57(74.05) | 70.64 | 63.70(74.63) | 41.87 | 73.26(82.20) | 65.40(80.61) | 73.53(91.25) | **85.32** |
| BA | 40.09(50.76) | 46.49(57.25) | 56.88 | 46.91(57.82) | 42.34 | 52.17(63.04) | 55.06(63.58) | 50.11(56.78) | **60.58** |
| TR11 | 31.39(43.11) | 33.22(48.88) | 44.56 | 33.48(49.69) | 39.39 | 37.58(58.60) | 30.56(67.63) | 45.39(70.93) | **48.68** |
| TR41 | 36.60(61.32) | 40.37(59.88) | **57.75** | 40.86(60.77) | 43.05 | 43.18(65.50) | 34.82(70.89) | 47.45(68.50) | 51.15 |
| TR45 | 33.22(48.03) | 38.69(57.87) | **56.17** | 38.96(57.86) | 41.94 | 44.36(74.24) | 38.04(72.50) | 50.37(78.12) | 55.93 |

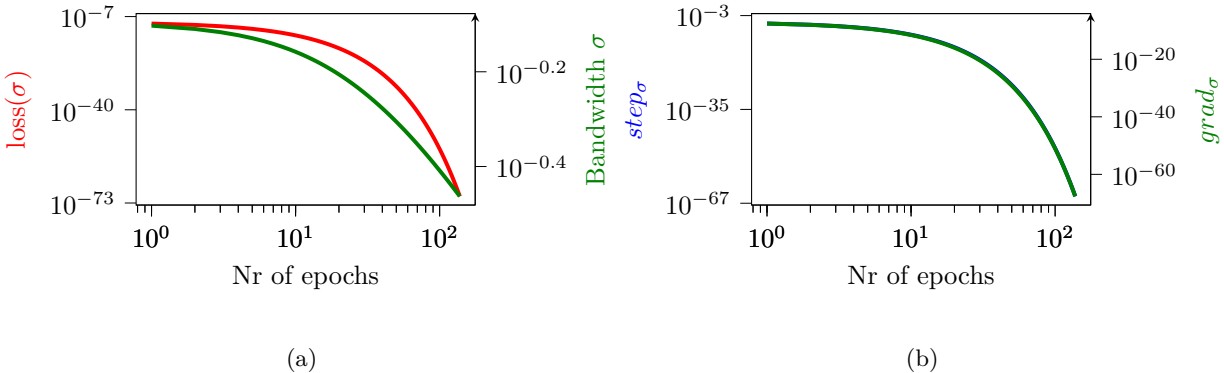

(a)                                                    (b)

Figure 16: The loss (loss $= \sum_{i=1}^{K} \lambda_i$) and the bandwidth $\sigma$ over the subsequent iterations for the COIL20 dataset in fig. 16a. Optimization of the bandwidth $\sigma$ on the COIL20 dataset in fig. 16b.

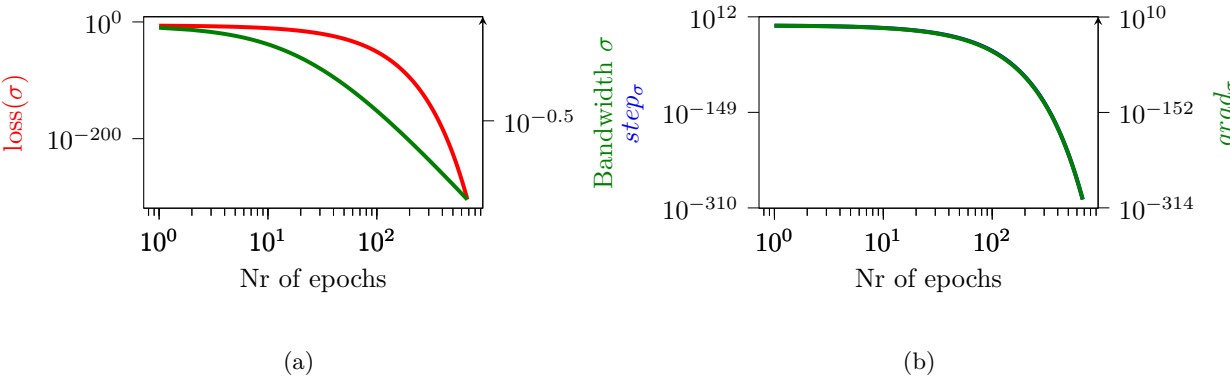

(a)                                                                (b)

Figure 17: The loss values (loss $= \sum_{i=1}^{K} \lambda_i$) over the iterations for the ORL dataset in fig. 17a. Optimization of the bandwidth $\sigma$ on the ORL dataset in fig. 17b.

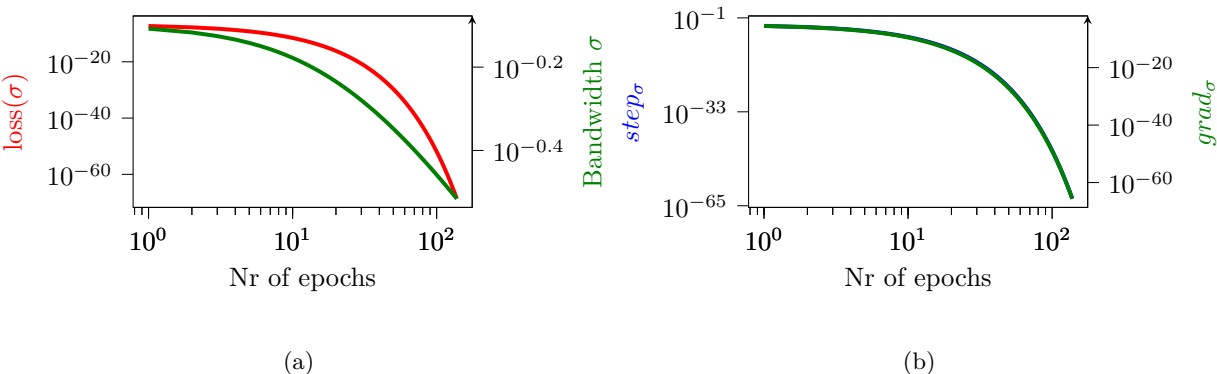

(a)                                                                (b)

Figure 18: The loss (loss $= \sum_{i=1}^{K} \lambda_i$) and the bandwidth $\sigma$ over the subsequent iterations for the YALE dataset in fig. 18a. Optimization of the bandwidth $\sigma$ on the YALE dataset in fig. 18b.

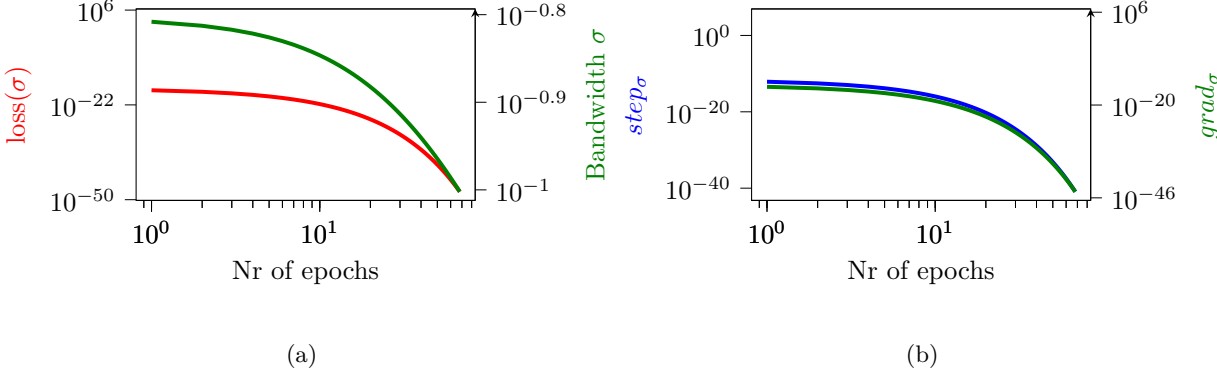

(a)                                                                (b)

Figure 19: The loss (loss $= \sum_{i=1}^{K} \lambda_i$) and the bandwidth $\sigma$ over the subsequent iterations for the BA dataset in fig. 19a. Optimization of the bandwidth $\sigma$ on the BA dataset in fig. 19b.

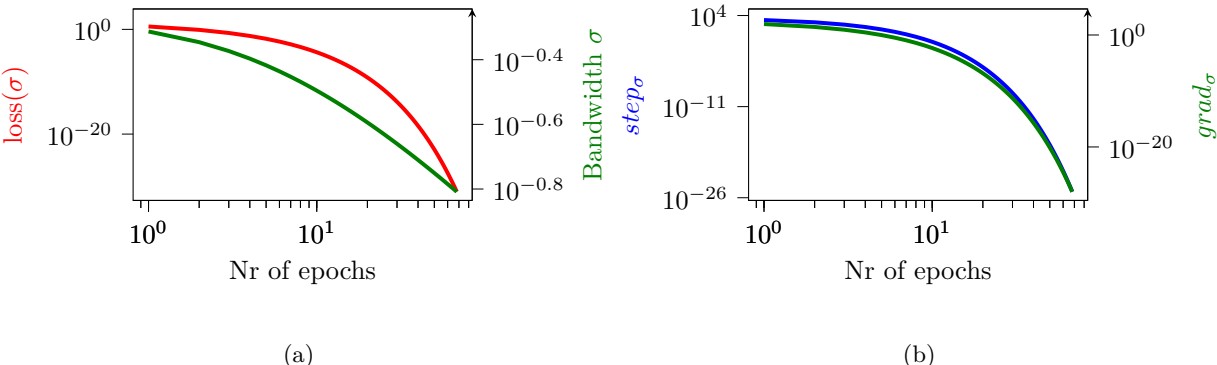

(a)                                                                (b)

Figure 20: The loss (loss $= \sum_{i=1}^{K} \lambda_i$) and the bandwidth $\sigma$ over the subsequent iterations for the TR11 dataset in fig. 20a. Optimization of the bandwidth $\sigma$ on the TR11 dataset in fig. 20b.

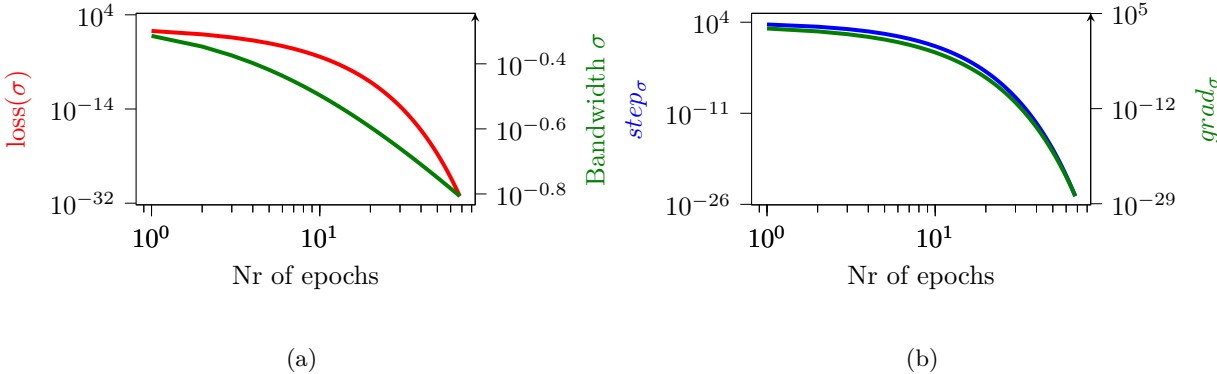

(a)                                                                (b)

Figure 21: The loss (loss $= \sum_{i=1}^{K} \lambda_i$) and the bandwidth $\sigma$ over the subsequent iterations for the TR41 dataset in fig. 21a. Optimization of the bandwidth $\sigma$ on the TR41 dataset in fig. 21b.

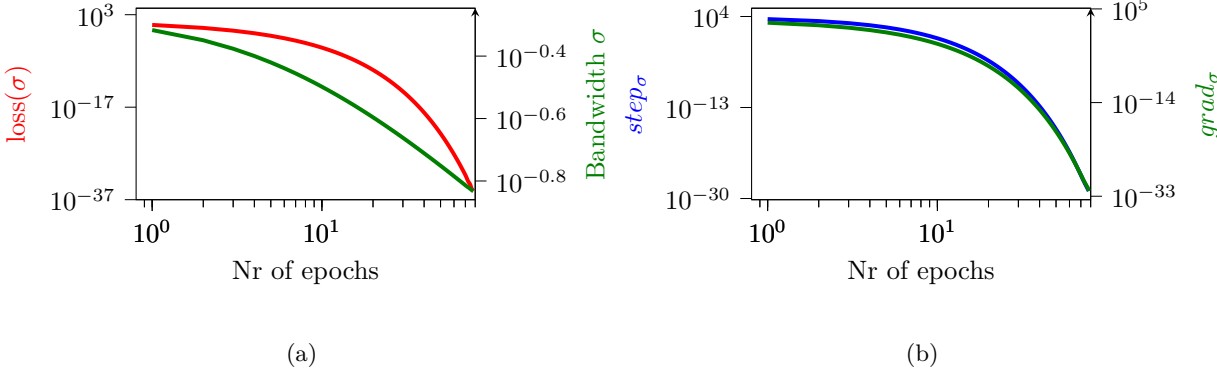

(a)                                                                (b)

Figure 22: The loss (loss $= \sum_{i=1}^{K} \lambda_i$) and the bandwidth $\sigma$ over the subsequent iterations for the TR45 dataset in fig. 22a. Optimization of the bandwidth $\sigma$ on the TR45 dataset in fig. 22b.

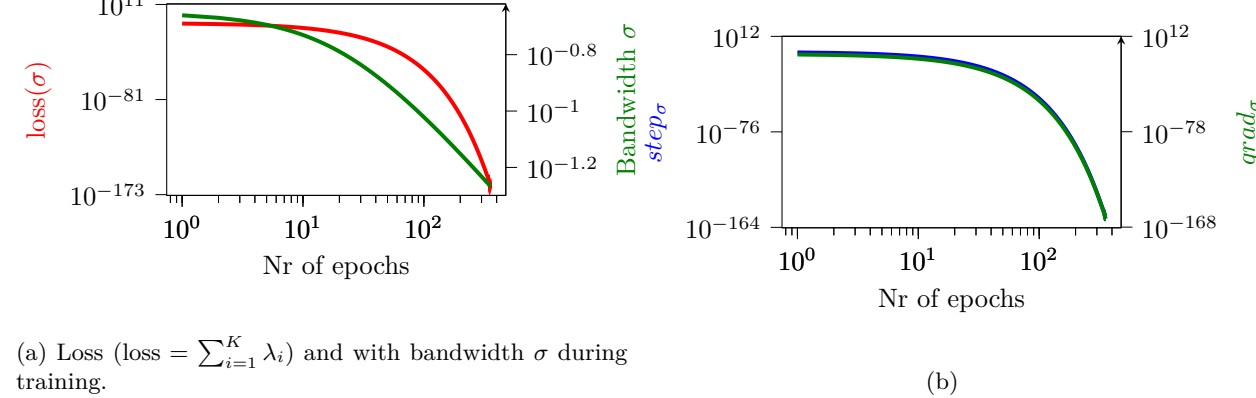

(a) Loss (loss $= \sum_{i=1}^{K} \lambda_i$) and with bandwidth $\sigma$ during training.

(b)

Figure 23: The loss (loss $= \sum_{i=1}^{K} \lambda_i$) and the bandwidth $\sigma$ over the subsequent iterations for the UMNIST dataset in fig. 23a. Optimization of the bandwidth $\sigma$ on the UMNIST dataset in fig. 23b.

