# OpenReview forum: "Unsupervised Similarity Learning for Spectral Clustering"
_TMLR — Rejected by TMLR_

### Review · Reviewer_kuMq · 2024-12-24

**Summary Of Contributions:**

The paper proposes a spectral clustering method that aims to eliminate the need for hyperparameters. Specifically, the authors use a similarity matrix constructed with a Radial Basis Function kernel, where the scale parameter $\sigma$ of the kernel is determined through a Newton’s optimization algorithm. The objective of this optimization is to minimize the sum of the smallest K eigenvalues of the graph Laplacian, which corresponds to splitting the data into K sub-clusters.

The contributions of the authors was to change the RBF kernel (in particular, the distance function) such that the graph laplacian stays positive semi-definite and the loss function remain convex. They were able to compare empirically their methods with other spectral clustering algorithms.

**Audience:**

Yes

**Broader Impact Concerns:**

No ethical concerns

**Claims And Evidence:**

No

**Requested Changes:**

More clarity is needed about:

- The claim that the decomposition of the distance $d(x, y) = u(x, y) + 1$ ensures convexity.
- How the domain of $\sigma$: $[0, \frac{\sqrt{6}}{3}]$ has been established, and if Lemma 7 still holds or needs a small modification.
- How the restrictive domain is not a limitation of the proposed method, which could be suboptimal for other datasets.

**Strengths And Weaknesses:**

**Strengths**

- The problem of spectral clustering is well motivated, and reducing the number of hyper parameters in a unsupervised context is a good approach.
- The method has been compared with other spectral clustering techniques, and empirically, it seems to outperforms other methods.

**Weaknesses**

**Critical Evaluation of the Theory and Methodology**

-  Lemma 7 fails when $u(x, y) = \cos(x, y)$ because $\cos(x, y)$ can be negative, as mentioned in the Method section. When $u(x, y) = d(x, y) \geq 0$, the expression $ 2 - 3\sigma^2 + 2u(x, y) \geq 0$ holds true only if $\sigma \leq \sqrt{\frac{2}{3}}$. It is unclear why the term $\frac{\sqrt{6}}{3}$ is used, as it does not directly appear in the theorem’s proofs (as a result of an inequality for example). This makes it difficult to understand why this term is necessary, even though it is important for defining the valid range of $\sigma$.

- It is not clear why breaking down the distance as $d(x, y) = u(x, y) + 1$ ensures that the loss function is convex. Can you show how this specific decomposition guarantees convexity?

-  Limiting $\sigma$ to the range $[0, \frac{\sqrt{6}}{3}]$ is restrictive. While this restriction might help the optimizer to converge, it does not necessarily ensure that the solution is the best possible (i.e., the loss could be lower outside the domain).

-  In terms of empirical results, other methods have shown better performance. For example, Bai et al. (2023) in their paper "Spectral Clustering with Robust Self-Learning Constraints" published in *Artificial Intelligence* (320:103924), performed better on the same dataset. This suggests that the current method may need improvement and should be compared more thoroughly with existing techniques.


**Slightly overstated claims:**
- The abstract and introduction state that the method "eliminates predefined metrics." However, it still uses a modified RBF kernel with a single adjustable parameter, sigma. The basic structure of the similarity metric—the RBF kernel—remains the same. This means the flexibility of the "learned" similarity function is still limited by how the kernel is designed. In this context, the assertion that the method "eliminates predefined metrics" appears to be inaccurate.

- The claim that the method can manage "highly non-convex shapes" appears somewhat misleading, as it suggests that the method is capable of effectively handling complex data in comparison to other existing approaches. The datasets employed (e.g., YALE, JAFFE, COIL20) along with examples such as rings and spirals are standard benchmarks commonly utilized in clustering research.

**Minor comments and typos:**
- Page 2, paragraph 1: Missing date at citation (Fan et al.).
- Page 2, paragraph 2: due to dependence on -> due to [their] dependence on
- Page 2, paragraph 3: a novel alternation -> a novel variation (or modification) in
- Can you introduce C, J and E before the algorithm 1?
- Table 2: 83.54 (CAN method for ACC) is the best value, and should be in bold.

---

> ### Author Response · Authors · 2025-01-16
> **Comment regarding the restrictive domain.**
>
> > **Comment 3:** How the restrictive domain is not a limitation of the proposed method, which could be suboptimal for other datasets?
>
> We thank the reviewer for this important comment.
>
> In such a scenario where the optimal $\sigma$ is outside the restricted domain (i.e., $\sigma_{optimal}\notin \Sigma$), then $\sigma=\frac{\sqrt{6}}{3}$ provides all eigenvalues equal to zero. In such a case, that dataset contains very high values of pairwise distances between the data pairs.
>
> To mitigate this scenario (and improve the numerical stability of the method), we perform two steps: first, the normalization of the data  (mean subtraction and division by the standard deviation), followed by the normalization of the matrix containing the distances (rescaled such as maximum value is one).
>
> These consecutive normalization steps drastically increase the chances that optimal $\sigma$ is inside the restricted domain (i.e., $\sigma_{optimal}\in \Sigma$).
>
> Notice, however, that if the loss at $loss(\sigma=\frac{\sqrt{6}}{3})>0$ and because its gradient is always positive (i.e., $loss(\sigma_{1})\geq loss(\sigma_{2}),\forall \sigma_{1},\sigma_{2}\in R_{\geq 0},s.t:\sigma_{1}\geq\sigma_{2}$) then $loss(\forall \sigma>\frac{\sqrt{6}}{3})\geq loss(\sigma=\frac{\sqrt{6}}{3})$ thus $\sigma_{optimal}\in \Sigma$.

---

> ### Author Response · Authors · 2025-01-16
>
> > **Comment 1 and 2:** The claim that the decomposition of the distance $d(x,y)=u(x,y)+1$ ensures convexity. How the domain of $\sigma\in [0,\frac{\sqrt(6)}{2}]$ has been established, and if Lemma 7 still holds or needs a small modification.
>
> We thank the reviewer for these important questions.
>
> To ensure the convexity of the proposed loss, we need the convexity of the proposed kernel $\(\frac{\partial^{2}K(\sigma^{2})}{\partial\sigma^{2}}\geq0$.
> The latter is ensured if and only if $2d(x,y)^{2}-3\sigma^{2}\geq 0$
> We cannot, however, find any range of values for $\sigma$ to guarantee $2d(x,y)^{2}-3\sigma^{2}\geq 0$ since it would require  $2d(x,y)^{2}\geq 3\sigma^{2},\forall d(x,u)\in R_{\geq0}$.
>
> However by modifying the distance to $d(x,y)=u(x,y)+1$ then we get the following $2(u(x,y)+1)-3\sigma^{2}=2-3\sigma^{2}+2u(x,y)$
>
> Notice that $u(x,y)\geq 0$ not only for squared Euclidean distance but even for cosine distance since $cos_{sim}(X, Y)=\frac{\sum_{i=1}^{N}X_{i}Y_{i}}{ |X| |Y|}$since for any two binary data vector X and Y $\forall X, Y\in [0,1]^{N},\sum_{i=1}^{N}X_{i}Y_{i}\geq 0, |X|\geq 0, |Y|\geq 0$ (why Lemma 2 holds).
>
> Leaving with the only requirement of $2-3\sigma^{2}\geq0\to 2\geq3\sigma^{2}\to \frac{2}{3}\geq\sigma^{2}\to \sqrt{\frac{2}{3}}\geq\sigma\to \sqrt{\frac{2\times3}{3\times3}}\geq\sigma\to \frac{\sqrt{6}}{3}\geq\sigma>0$. Hence the domain $\sigma\in (0,\frac{\sqrt(6)}{2}]$.

---

> ### Author Response · Authors · 2025-01-22
> **Typo found in the text.**
>
> We wanted to let the reviewers know that after double-checking the experiments, we noticed an important 'typo' in the text.
>
> The utilized distance $u(x,y)$ for binary data in the experimental code is $u(x, y) = 1-\cos_{sim}(x, y)$ while in the text, we wrote that $ u(x, y) = \cos_{sim}(x, y)$.
>
> Notice since data are binary, then $0\leq u(x,y)\leq1$ in both cases, although the utilized distance is $u(x,y)=0$ for identical pairs and $u(x,y)=1$ for the orthogonal ones.
>
> We are updating this on the updated report together with the recommended changes.

---

> ### Author Response · Authors · 2025-01-23
> **Reply on minor comments and typos**
>
> We thank the reviewer for the minor comments and typos that further helped us improve the manuscript.
> >Page 2, paragraph 1: Missing date at citation (Fan et al.).
>
> We updated the citation.
> >Page 2, paragraph 2: due to dependence on -> due to [their] dependence on
>
> We rewrote this part as suggested
> >Page 2, paragraph 3: a novel alternation -> a novel variation (or modification) in
>
> We rewrote this part as suggested
> >Can you introduce C, J, and E before the algorithm 1?
>
> We wrote a notation section in the Appendix following Reviewer 5tpp that includes the description of matrix C, J, and E as suggested.
> >Table 2: 83.54 (CAN method for ACC) is the best value and should be in bold.
>
> The values suggested have been highlighted in bold as suggested.
>
> Thank you for your time, and please let us know if there is any additional feedback on our replies.

---

> ### Author Response · Authors · 2025-01-24
> **Reply on slightly overstated claims**
>
> >**Comment 1:** The abstract and introduction state that the method "eliminates predefined metrics." However, it still uses a modified RBF kernel with a single adjustable parameter, sigma. The basic structure of the similarity metric—the RBF kernel—remains the same. This means the flexibility of the "learned" similarity function is still limited by how the kernel is designed. In this context, the assertion that the method "eliminates predefined metrics" appears to be inaccurate.
>
> We thank the reviewer for the constructive comment, and since it is in line with the comment from **Reviewer 5tpp**, to make the statement more coherent, we altered it as follows:
> "Instead of predefining this metric upfront **as a fixed parametric function**, we introduce a novel approach that learns **the optimal parameters of a similarity function** through parameter optimization."
>
> >**Comment 2:** The claim that the method can manage "highly non-convex shapes" appears somewhat misleading, as it suggests that the method is capable of effectively handling complex data in comparison to other existing approaches. The datasets employed (e.g., YALE, JAFFE, COIL20) along with examples such as rings and spirals are standard benchmarks commonly utilized in clustering research.
>
> We thank the reviewer for the comment. We conducted experiments on spirals, rings, and others to emphasize the method's capabilities in identifying boundaries that follow "non-convex trajectories," although the data themselves are simple in complexity. Thus, the main ability of the method is to identify these non-convex separations boundaries and linearize them. This does not necessarily translate directly into handling complex datasets as the latter would require feature extraction using DL or foundation models, which has been left as future work. Nevertheless we have changed it  "non-convex shapes" instead not to produce any further confusion.
>
> We thank you again for these very constructive comments and we look forward to your feedback.

---

> ### Author Response · Authors · 2025-01-24
> **Reply on Spectral Clustering with Robust Self-Learning Constraints**
>
> > In terms of empirical results, other methods have shown better performance. For example, Bai et al. (2023) in their paper "Spectral Clustering with Robust Self-Learning Constraints" published in Artificial Intelligence (320:103924), performed better on the same dataset. This suggests that the current method may need improvement and should be compared more thoroughly with existing techniques.
>
> We thank the reviewer for pointing out this related work.
>
> Indeed, some methods perform better than our dataset, as shown in our experimental tables as well. However, the main advantages of the proposed methods relative to all the existing ones in an unsupervised clustering setting are (a) the removal of hyperparameters in the loss or the similarity function and (b) the convex optimization setting. Both of which remove the dependency of the final result from the initialization. Notice that Bai et al. (2023)  work has two hyper-parameters that cannot be validated if no label is provided, thus making their clustering performance strictly dependent on the handpicked initial values of these hyperparameters.
>
>
> Thank you for your time looking forward to your feedback.

---

> > ### Author Response · Authors · 2025-01-25
> > **Revised Manuscript**
> >
> > Dear **Reviewer kuMq**
> >
> > We greatly appreciate your insightful questions and constructive feedback.
> >
> > We have responded to your comments and updated the manuscript accordingly.
> >
> > We look forward to receiving your feedback and are ready to address any further concerns you might have.
> >
> > Thank you for your time.

---

> > > ### Comment · Reviewer_kuMq · 2025-02-17
> > > **Final decision**
> > >
> > > I would like to thank the authors for their extensive feedback and the additional explanations provided.
> > >
> > > My primary concern is that the paper would benefit from a clearer presentation. The assumptions, hypotheses, and results are not consistently delineated, which may lead to misunderstandings regarding the motivations and contributions of the work.
> > >
> > > **For example**, the key assumption that $d(x, y)$ is always greater than 1 is introduced without sufficient justification, while it plays a central role in determining the range of $\sigma$. In Corollary 1, this assumption appears somewhat arbitrary, and the resulting formulation—particularly the introduction of $u(x,y)$—seems unnecessarily complex. In fact, one could simplify Corollary 1 to: $\frac{\partial^2 K(\sigma^2)}{\partial \sigma^2} \geq 0 \iff \sigma \leq \frac{\sqrt{6}}{3} \sqrt{d(x,y)}$. Additionally, I noticed an error in Corollary 1: the expression: $2(u(x,y)+1) -  3\sigma^2$ should be written as $2(u(x,y)+1)^2 -  3\sigma^2$. (Note that this error doesn't affect the overall definition of the range of $\sigma$, but contributes to the overall lack of clarity.)
> > >
> > > Because of these issues, I am inclined to lean towards rejection in its current form. I am confident that with significant effort towards simplification, this work could be accepted.

---

> ### Author Response · Authors · 2025-02-17
> **Response to the final decision part 1**
>
> We thank the reviewer for taking the time to follow up on our replies, and we wanted to take this opportunity to improve our work further.
>
> >My primary concern is that the paper would benefit from a clearer presentation. The assumptions, hypotheses, and results are not consistently delineated, which may lead to misunderstandings regarding the motivations and contributions of the work.
>
> Although a clearer presentation is posed as a primary concern, we would have greatly appreciated it if the authors had raised this issue from the beginning so we could have had the opportunity to address it further.
>
> Regarding the **assumptions** and **hypothesis**, we wanted to emphasize that **this is a follow-up work on SC where, with a modification on the RBF kernel (Lemma 1), we were able to propose a convex optimization setting for it is bandwidth kernel, which is otherwise treated as a handpicked hyperparameter**.
>
> The **motivation** and **contribution** of this work remain **optimizing the kernel bandwidth**.
> The expectation is to get **better clustering results** as the method is very data-driven and has minimal human intervention.
>
> >For example, the key assumption that $d(x,y)$ is always greater than 1 is introduced without sufficient justification, while it plays a central role in determining the range of $\sigma$. In Corollary 1, this assumption appears somewhat arbitrary, and the resulting formulation—particularly the introduction of $u(x,y)$—seems unnecessarily complex. In fact, one could simplify Corollary 1 to: $\frac{\partial^2K(\sigma^2)}{\partial^2}\geq 0\iff \sigma\leq \frac{\sqrt{6}}{3}\sqrt{d(x,y)}$
>
> We have to observe a convex behavior of the RBF kernel (i.e., $\frac{\partial^2K(\sigma)}{\partial\sigma^2}\geq 0$) to ensure the convexity of the GC loss w.r.t $\sigma$.
>
> However using $0<\sigma\leq \frac{\sqrt{6}}{3}\sqrt{d(x,y)}$ it is not possible to define a range of $\sigma$  to guarantee this inequality since $d(x,y)\in R_{\geq 0}$. A more concrete example, in the case of $d(x,y)=0$ (in the case of duplicated data), it requires $0<\sigma\leq0$.
>
> Therefore, to ensure a range of $\sigma$ that guarantees convexity of the GC loss w.r.t $\sigma$, we alter the distances to $d(x,y)=u(x,y)+1$ and we only require $\forall u(x,y)\geq 0$.
>
> We hope we addressed the reviewer's concerns regarding the **assumptions, hypotheses, and results,** and we are ready to answer any additional concerns the reviewer might have.
>
> Thank you again for your time.

---

> ### Author Response · Authors · 2025-02-17
> **Response to the final decision part 2**
>
> We thank the reviewer for their time.
>
> > Additionally, I noticed an error in Corollary 1: the expression: $2(u(x,y)+1)-3\sigma^{2}$ should be written as $2(u(x,y)+1)^{2}-3\sigma^{2}$. (Note that this error doesn't affect the overall definition of the range of
> $\sigma$, but contributes to the overall lack of clarity.)
>
> We investigated again the second derivative of the RBF kernel $\frac{\partial^2K(\sigma)}{\partial\sigma^2}$ in eqn 24 till 26 in the Corollary 2, and we could not find any mistake.
>
> We get  $2(u(x,y)+1)-3\sigma^{2}$ as the correct answer.
>
> We have included an extended derivation below.
>
> **First Derivative:**
> $$ \frac{\partial K(\sigma)}{\partial \sigma} = \frac{\partial}{\partial \sigma} \left( e^{-\frac{d(x,y)}{\sigma^2}} \right) $$
>
> Using the chain rule:
> $$ \frac{\partial K(\sigma)}{\partial \sigma} = e^{-\frac{d(x,y)}{\sigma^2}} \cdot \frac{\partial}{\partial \sigma} \left( -\frac{d(x,y)}{\sigma^2} \right) $$
>
> Compute the derivative inside:
> $$ \frac{\partial}{\partial \sigma} \left( -\frac{d(x,y)}{\sigma^2} \right) = -\frac{d(x,y) \cdot (-2)}{\sigma^3} = \frac{2d(x,y)}{\sigma^3} $$
>
> So:
> $$ \frac{\partial K(\sigma)}{\partial \sigma} = e^{-\frac{d(x,y)}{\sigma^2}} \cdot \frac{2d(x,y)}{\sigma^3} $$
>
> **Second Derivative:**
> Now, take the derivative of the first derivative:
> $$ \frac{\partial^2 K(\sigma)}{\partial \sigma^2} = \frac{\partial}{\partial \sigma} \left( e^{-\frac{d(x,y)}{\sigma^2}} \cdot \frac{2d(x,y)}{\sigma^3} \right) $$
>
> Use the product rule:
> $$ \frac{\partial^2 K(\sigma)}{\partial \sigma^2} = \frac{\partial}{\partial \sigma} \left( e^{-\frac{d(x,y)}{\sigma^2}} \right) \cdot \frac{2d(x,y)}{\sigma^3} + e^{-\frac{d(x,y)}{\sigma^2}} \cdot \frac{\partial}{\partial \sigma} \left( \frac{2d(x,y)}{\sigma^3} \right) $$
>
> We already know:
> $$ \frac{\partial}{\partial \sigma} \left( e^{-\frac{d(x,y)}{\sigma^2}} \right) = e^{-\frac{d(x,y)}{\sigma^2}} \cdot \frac{2d(x,y)}{\sigma^3} $$
>
> Now compute the second term:
> $$ \frac{\partial}{\partial \sigma} \left( \frac{2d(x,y)}{\sigma^3} \right) = 2d(x,y) \cdot \frac{\partial}{\partial \sigma} \left( \sigma^{-3} \right) = 2d(x,y) \cdot (-3)\sigma^{-4} = -\frac{6d(x,y)}{\sigma^4} $$
>
> Putting it all together:
> $$ \frac{\partial^2 K(\sigma)}{\partial \sigma^2} = \left( e^{-\frac{d(x,y)}{\sigma^2}} \cdot \frac{2d(x,y)}{\sigma^3} \right) \cdot \frac{2d(x,y)}{\sigma^3} + e^{-\frac{d(x,y)}{\sigma^2}} \cdot \left( -\frac{6d(x,y)}{\sigma^4} \right) $$
>
> Simplify:
> $$ \frac{\partial^2 K(\sigma)}{\partial \sigma^2} = e^{-\frac{d(x,y)}{\sigma^2}} \left( \frac{4d(x,y)^2}{\sigma^6} - \frac{6d(x,y)}{\sigma^4} \right) $$
>
> Factor out $\frac{2d(x,y)}{\sigma^6}$:
> $$ \frac{\partial^2 K(\sigma)}{\partial \sigma^2} = e^{-\frac{d(x,y)}{\sigma^2}} \cdot \frac{2d(x,y)}{\sigma^6} \left( 2d(x,y) - 3\sigma^2 \right) $$
>
> **Final Result:**
> $$ \frac{\partial^2 K(\sigma)}{\partial \sigma^2} = 2e^{-\frac{d(x,y)}{\sigma^2}} \cdot \frac{d(x,y)}{\sigma^6} \left( 2d(x,y) - 3\sigma^2 \right) $$
> $$ \frac{\partial^2 K(\sigma)}{\partial \sigma^2} = 2e^{-\frac{d(x,y)}{\sigma^2}} \cdot \frac{d(x,y)}{\sigma^6} \left[ 2(u(x,y)-1) - 3\sigma^2 \right] $$
>
> We welcome any feedback from the reviewer and thank the reviewer for allowing us to clarify Corollary 1 further.

---

> ### Comment · Reviewer_kuMq · 2025-02-17
>
> Regarding the mistake, following your equation, you noted, in Corollary 1:
>
> $2d(x,y)^2-3\sigma^2 = 2 (u(x,y)+1) - 3 \sigma^2 $. This equality should be, if $d(x,y)=u(x,y)+1$, $2d(x,y)^2-3\sigma^2 = 2 (u(x,y)+1)^2 - 3 \sigma^2 $. Again, this error doesn't affect the end result.

---

> > ### Author Response · Authors · 2025-02-18
> > **Reply regarding a typo Corollary 1**
> >
> > We thank the reviewer very much for the follow-up comment.
> > >**Comment:** Regarding the mistake, following your equation, you noted, in Corollary 1: $2d(x,y)^{2}-3\sigma^{2}=2(u(x,y)+1)-3\sigma^{2}$ This equality should be, if $d(x,y)=u(x,y)+1, 2d(x,y)^{2}-3\sigma^{2}=2(u(x,y)+1)^{2}-3\sigma^{2}$. Again, this error doesn't affect the end result.
> >
> > The squared distance in this equation is a typo, as the correct last term in the second derivative of the kernel as in eqn 26 is as follows:
> > $\textcolor{blue}{2d(x,y)}-3\sigma^{2}=2(u(x,y)+1)-3\sigma^{2}$.
> >
> > We thank the reviewer for spotting this and wanted to re-iterate that the last term remains without a squared (i.e., $\textcolor{blue}{2d(x,y)}-3\sigma^{2}$), even though it does not affect the results.
> >
> > We updated this in the manuscript and thank the reviewer for the thorough investigation.

---

### Review · Reviewer_5tpp · 2025-01-07

**Summary Of Contributions:**

This paper considers the spectral clustering problem. The success of the spectral clustering algorithm hinges on the choice of the similarity function. The paper proposes a modified radial basis function (RBF) for the similarity function with a bandwidth that is learned through a convex algorithm. The main contributions are as follows:

* The paper introduces a convex algorithm for learning the bandwidth, making the resulting algorithm simple and free of hyperparameters, in contrast to existing methods.

* Extensive numerical experiments, covering various datasets and baseline methods, demonstrate how the proposed method outperforms the existing baselines.

**Audience:**

Yes

**Broader Impact Concerns:**

I have no concerns.

**Claims And Evidence:**

Yes

**Requested Changes:**

- The review recommends addressing some of the weaknesses listed above.

> Minor: Suggestions to improve writing/typesetting

- Page 2: When referring to $d(x,y)= u(x,y)+1$, u(x,y)$ is not defined.
- Page 2: $cos_{sim}(x,y)$ should be in math mode.
- Put full stop after equation 1. Such that should be abbreviated as s.t.
- Define what a kernel is before stating Lemma 1
- What happen as $\sigma \rightarrow \infty$ in eq. 1?
- In section 2.1, define the abbreviation GC before using it.
- The normalization condition in 2.1 should be $f^Tf = 1$, and not $ff^T$.
- Page 16: For the last line in the proof of Theorem 2, should not f and f^T be swapped?
- ln function in equation 15 should be in math mode
- Have the authors explored if strict convexity can be attained?
- Is the standardization procedure also applied for all baselines? How were the parameters of the other baselines tuned? Have the authors explored the effect of different data pre-processing (e.g., scaling to [0,1], normalizing)?
- For small sigma, it might be good to add discussion of how floating point issues might be avoided in the updates of Algorithm 1?
- Paper might benefit from the addition of a notation section
- max in equation 13 should be in math mode
- Table 5: Any intuition/explanation why the method does not outperform on TR41 ad TR45?

**Strengths And Weaknesses:**

**Strengths**

- The proposed algorithm for updating the learnable parameter is straightforward and is based on optimizing a convex function, which is easily interpretable through Ky-Fan's theorem. Moreover, the necessary components of the algorithm, such as the derivatives, are provided in closed form.
- The authors have conducted extensive experiments and comparisons with baseline methods. Additionally, several additional experiments are provided in the supplementary material. The reviewer finds these experiments convincing and believes they effectively demonstrate the appeal of the method, which works without the need for tuning.

**Weaknesses**

- Why does the vanishing of the mixed derivative in equation (4) suffice as a criterion for identifying the minimizer of the left-hand side (LHS) of equation (4)? Right after equation (4), the paper notes that "... an alternation between each parameter is adopted." Does this alternating minimization resemble or reduce to non-convex approaches, as discussed in the introduction? The reviewer would appreciate clarification on this point.
- The abstract is somewhat misleading, as it states, "to learn similarity metrics," while the proposed algorithm actually learns the bandwidth of a modified RBF function. Could the authors clarify this discrepancy?
- Regarding equation (6), my understanding of Ky-Fan's theorem, as applied to spectral clustering, is that the maximizer of $trace(X^T H X)$ over $X^T X = I$ (for Hermitian $X$) is attained at the first k eigenvectors of $H$ (up to unitary transformation). In equation (6), do $f_1, \dots, f_k$ refer to the eigenvectors? Perhaps explicitly stating what GC represents in this case would help avoid confusion, especially in contrast to the simpler case of a solo quadratic form in equation (3).
- The paper states, "Since SC is predominantly affected by relative rather than absolute distances, a uniform translation of the Euclidean metric by a unit value does not alter the hierarchy of pairwise distances within a dataset. Consequently, such a shift does not impact the SC algorithm’s results." The reviewer would appreciate a justification for this claim, particularly the last sentence.

---

> ### Author Response · Authors · 2025-01-15
> **Comment regarding the convexity and the alternation procedure.**
>
> > **Comment 1:** "Why does the vanishing of the mixed derivative in equation (4) suffice as a criterion for identifying the minimizer of the left-hand side (LHS) of equation (4)? Right after equation (4), the paper notes that "... an alternation between each parameter is adopted." Does this alternating minimization resemble or reduce to non-convex approaches, as discussed in the introduction? The reviewer would appreciate clarification on this point."
>
> Dear Reviewer,
>
> Thank you for your very insightful question.
>
> The process begins with an alternating gradient descent (analytical) strategy between $\sigma$ and $f$ to minimize $G(\sigma, f)$.
>
> However, once we derive a closed-form solution through eigenvalue decomposition (EVD) for $f$ s.t $f=EVD(L(\sigma))$, we no longer need to perform gradient descent on $f$.
> This leaves us solely with gradient descent on $\sigma$.
>
> The search for $\sigma$ remains within the convex space if, and only if, $f$ remains restricted as the eigenvector of $L$.
>
> To exemplify this concept more simply, consider minimizing $\arg\min_{a,b} f(a,b)$ where $b$ is directly obtained through the function $b=g(a)$. Subsequently, the process simplifies to executing gradient descent solely on $a$ in $\arg\min_a f(a, g(a))$ where $f(a, g(a))$ is convex w.r.t $a$.
>
> In our case, the substitution becomes $f = \text{EVD}(L(\sigma))$, which must be computed (and not searched) independently.
>
> In other words, the alternation remains in place, but one step is computation (i.e., $f = \text{EVD}(L(\sigma))$), and the other step is search through gradient descent (i.e., $\arg\min_{\sigma}G(\sigma, \text{EVD}(L(\sigma)))$.

---

> > ### Comment · Reviewer_5tpp · 2025-01-17
> > **alternation procedure**
> >
> > Thank you for the clarification. I believe incorporating part of the discussion above into the paper would be beneficial. In particular, emphasizing that 'the search for $\sigma$ remains within the convex space if, and only if, $f$ remains restricted as the eigenvector of $L$' would be valuable.

---

> ### Author Response · Authors · 2025-01-15
> **Comments regarding the dependence of SC on relative distances**
>
> > **Comment 4:** The paper states, "Since SC is predominantly affected by relative rather than absolute distances, a uniform translation of the Euclidean metric by a unit value does not alter the hierarchy of pairwise distances within a dataset. Consequently, such a shift does not impact the SC algorithm’s results." The reviewer would appreciate a justification for this claim, particularly the last sentence.
>
> We thank the reviewer for another very important comment.
>
> To illustrate the dependence of SC on relative distances, we note that uniformly increasing all distances in the Euclidean metric by a unit value corresponds to a linear scaling of the affinity matrix when employing the RBF kernel (see eqn 2 in the main text).
> Furthermore, this scaling of the affinity matrix modifies the eigenvalues of the Laplacian matrix while leaving its eigenvectors unchanged (see Lemma).
>
> **Lemma**
> Given an affinity matrix $ A $ and its corresponding Laplacian matrix $ L$, scaling the affinity matrix $ A_{scaled} = \gamma A$ s.t $\gamma>0$ preserves the eigenvectors of the resulting Laplacian matrix $ L_{scaled} $, while the eigenvalues are simply scaled by $ \gamma $, i.e., $ \lambda_{scaled} = \gamma\lambda $.
>
>
> **Proof**
>
> Consider the eigenvector matrix $ F $ of $ \mathbf{L} $ such that
> $$
> F^{\top} \mathbf{L} F = \Lambda,
> $$
> where $ \Lambda $ is a diagonal matrix containing the eigenvalues $ \lambda_{1, \ldots, N} $.
>
> Notice that the new degree matrix for $ A_{scaled} $ is $ D_{scaled} = \gamma D $, leading to the following:
>
> $$L_{scaled}= D_{scaled} - A_{scaled}=\gamma D - \gamma A=\gamma (D - A)=\gamma L$$
>
> Then we prove that the orthogonal matrix containing the eigenvectors $F$ diagonalizes $L_{scaled}$ to:
>
> $$F^{\top} L_{scaled} F= F^{\top} \{\gamma L\} F = \gamma F^{\top} L F = \gamma\Lambda= \Lambda_{scaled}$$
>
> With the $F^{\top} L_{scaled} F= \Lambda_{scaled}$, it can be safely stated that $F$ are the eigenvectors of $L_{scaled}$ as well.
>
> Since the eigenvalues $ \Lambda_{scaled} = \gamma\Lambda $ are uniformly scaled, their order remains unchanged.
>
> Consequently, the order of the eigenvectors corresponding to the smallest eigenvalues also stays unchanged for both $ L_{scaled} $ and $ L $, thereby maintaining consistent GC.
>
> **We are incorporating this in the Appendix as well**

---

> ### Author Response · Authors · 2025-01-20
> **Revision of the abstract**
>
> > **Comment 2:** "The abstract is somewhat misleading, as it states, "to learn similarity metrics," while the proposed algorithm actually learns the bandwidth of a modified RBF function. Could the authors clarify this discrepancy?"
>
> We appreciate the reviewer's constructive question.
>
> What we stated, "to learn similarity metrics," is a more generic version of "to learn the parameters of the similarity metrics." We decided it would make it more suitable for the abstract section to enhance its readability
>
> However, we have revised the abstract and altered it to "to learn the parameters of the similarity metrics" for greater clarity and consistency.
>
> Thank you for your comment.

---

> ### Author Response · Authors · 2025-01-20
> **Comment regarding the notation in the proposed objective**
>
> > **Comment 3:** "Regarding equation (6), my understanding of Ky-Fan's theorem, as applied to spectral clustering, is that the maximizer of  $trace(X^{T}HX)$ over $X^{T}X=I$  over  (for Hermitian $X$) is attained at the first k eigenvectors of(up to unitary transformation). In equation (6), do $f_{1},\cdots,f_{k}$ refer to the eigenvectors? Perhaps explicitly stating what GC represents in this case would help avoid confusion, especially in contrast to the simpler case of a solo quadratic form in equation (3)."
>
> We appreciate the question and your comment.
>
> Ky-Fan's theorem allows us to optimize the expression $trace(X^T H X)$ s.t $X^T X = I$ where $H$ is a Hermitian matrix.
>
> In our specific context, $L\in R^{N\times N}\$ represents the Hermitian matrix (since $L = L^T$ and no complex value is present), and $f_1, \dots, f_k$ are its bottom orthogonal eigenvectors.
>
> Accordingly, $f_1, \dots, f_k$ correspond to the eigenvectors of the Laplacian matrix, which is Hermitian.
> We will clarify this point in our text to ensure its emphasis.
>
> Thank you for your valuable feedback.

---

> ### Author Response · Authors · 2025-01-23
> **Reply on some of the suggestions to improve writing/typesetting**
>
> We thank the reviewer for the very thorough suggestions which we adopted as follow.
>
> >Page 2: When referring to $d(x,y)= u(x,y)+1$, u(x,y)$ is not defined.
>
> We added a :**where $u(x, y) = \|x - y\|^2$ is used for non-binary data and $u(x, y) = 1-\cos_{sim}(x, y)$ for binary data.**
>
> >Page 2: $cos_{sim}(x,y)$ should be in math mode.
>
> Changed to $\cos_{sim}(x, y)$
> >Put full stop after equation 1. Such that should be abbreviated as s.t.
>
> Done.
> >Define what a kernel is before stating Lemma 1
>
> Before Lemma 1 we added: **Notice that a kernel function $K$ maps two inputs $x$ and $y$ to a scalar that represents their inner product in a higher-dimensional space, expressed as $K(x, y) = \langle \phi(x), \phi(y) \rangle $, where $\phi$ is an implicit mapping.**
> >What happen as $\sigma \rightarrow \infty$ in eq. 1?
>
> In this case $\sigma \rightarrow \infty$ every edge weight similarity becomes very $\infty$ big namely all the data points collapse into a single point in the eigenspace. Thus the data topology is lost.
> >In section 2.1, define the abbreviation GC before using it.
>
> Done.
> >The normalization condition in 2.1 should be $f^Tf = 1$, and not $ff^T$.
>
> Done.
> >Page 16: For the last line in the proof of Theorem 2, should not f and f^T be swapped?
>
> Yes and we dod the swap.
> >ln function in equation 15 should be in math mode
>
> Done.
>
> >Paper might benefit from the addition of a notation section
>
> Added a notation section at the start of the Appendix
> >max in equation 13 should be in math mode
>
> Done

---

> ### Author Response · Authors · 2025-01-24
> **Reply on Minor: Suggestions to improve writing/typesetting**
>
> >Have the authors explored if strict convexity can be attained?
>
> Following up with the discussion with **Reviewer QMhR** the loss function is already strictly convex.
> Nevertheless, due to the limited precision, the model cannot be guaranteed to behave as such since some edge similarity values might get truncated towards zero. In this case, truncating the bottom eigenvalues faster than the rest, nevertheless ensuring the convexity.
>
> >Is the standardization procedure also applied for all baselines?
>
> Mean subtraction is applied for all baselines for all non-binary datasets. Division by variance is solely applied in our case to avoid floating point issues.
> >How were the parameters of the other baselines tuned?
>
> In general, we select a range of values and then average the ACC and NMI. For example, when using RBF, we select 7 different bandwidth values $\sigma=(0.01, 0.05, 0.1, 1, 10, 50, 100)$ and average their performance.
> >Have the authors explored the effect of different data pre-processing (e.g., scaling to [0,1], normalizing)?
>
> The proposed method cannot work without  **mean subtraction**. However, if the floating point allows it, it converges slowly when **no division by standard deviation.**  We did not try scaling [0,1] however, this would require mean subtraction to work, although might be a replacement for standard deviation division
>
> >For small sigma, it might be good to add discussion of how floating point issues might be avoided in the updates of Algorithm 1?
>
> This is indeed a limitation of the RBF kernel in general and of the proposed method in particular.
> However, we are still thinking about how to perform the optimization in the log scale while maintaining the convexity, but we do not have a concrete direction yet.
> Nevertheless, we perform two consecutive scaling operations to somehow mitigate this challenge. Namely, division by standard deviation and scaling of the distance matrix to [0,1] as indicated in line 3 in Algorithm 1.
>
> >Table 5: Any intuition/explanation why the method does not outperform on TR41 ad TR45?
>
> This comes down to the chosen hyper-parameter for the competing methods. Our method cannot underperform if the hyperparameters of the competing methods are handpicked very closely or at exactly their optimal magnitude.
>
> Thank you very much for your comments, as they help us make the submission stronger.

---

> > ### Author Response · Authors · 2025-01-25
> > **Revised Manuscript**
> >
> > Dear **Reviewer 5tpp**,
> >
> > Thank you for your insightful questions and valuable feedback.
> >
> > We have addressed your comments and made the appropriate revisions to our manuscript.
> >
> > We are eager to hear any additional feedback you may have and are prepared to respond to further concerns.
> >
> > We appreciate your time and effort.
> >
> > Best regards.

---

> > > ### Comment · Reviewer_5tpp · 2025-01-29
> > > **Thank you**
> > >
> > > I would like to thank the authors for diligently addressing my suggestions, as well as those of the other reviewers. I believe the manuscript has significantly improved as a result of these changes.

---

### Review · Reviewer_QMhR · 2025-01-14

**Summary Of Contributions:**

The authors investigate an alternating optimization to determine the variance hyperparameter $\sigma^2$ in spectral clustering. The cost function is the sum of the smallest K eigenvalues of the graph Laplacian, where K is the dimensionality of the embedding space, and it is minimized by alternately computing these eigenvalues and refining the choice of $\sigma^2$. The authors present empirical results on several data sets.

**Audience:**

No

**Broader Impact Concerns:**

None.

**Claims And Evidence:**

No

**Requested Changes:**

The authors should address the previously mentioned concerns regarding the sensibility of the loss function and the practical usage of this approach.

Here are further suggestions to improve the clarity of the paper:

* Lemma 1 follows immediately from elementary properties of kernels; see Propositions 13.1 and 13.2 in Learning with Kernels (Scholkopf & Smola, 2002). The proof does not require a half-page of exposition.

* The constant $\sqrt{6}/3$ appears mysteriously throughout section 2. Somewhere at the top the authors should simply observe that the function $f(\sigma) = \exp(-a/\sigma^2)$ is convex over the interval $(0,\sqrt{6}/3)$ for all $a$ in some appropriate interval.

* For completeness, the text should precisely and prominently state how the kernel matrix K, the adjacency matrix A, and the Laplacian are related to one another.

* The authors claim in Theorem 1 that the sum of K eigenvalues from the graph Laplacian is a convex function of $\sigma$. It is a well-known result in convex optimization that the sum of the top K eigenvalues of a symmetric matrix is a convex function of its matrix elements; see page 118 in Convex Optimization (Boyd & Vandenberghe, 2003). Can the authors (drastically) simplify the proof of Theorem 1 using this property, along with the observation that $\exp(-a/\sigma^2)$ is convex in $\sigma$ for sufficiently small $\sigma$? Or if not -- if the proof contains other essential ideas -- can the authors more clearly highlight the additional or alternative steps that are required?

* Graph Laplacian embeddings are computed from bottom eigenvectors; the paper should mention this explicitly.

* How do you prevent "over-stepping" in Newton's method (eq. 9) so that $\sigma_i$ does not obtain negative values?

Perhaps worth considering or mentioning:

* Can the authors generalize their method to estimate a positive-definite covariance matrix $\Sigma$ rather than a single variance $\sigma^2$?

**Strengths And Weaknesses:**

* One strength of the submission is its empirical evaluation. The proposed method is benchmarked on six data sets, compared to ten competing methods, and evaluated using two different metrics - normalized mutual information (NMI) and accuracy (ACC).

* The main weaknesses of the paper concern its mathematical development, the soundness of the loss function, and the proposed usage of the algorithm by practitioners.

* Is the loss function sensible? Theorem 2 shows that the loss is non-decreasing for positive $\sigma$. On page 5, the authors state that "although reducing the magnitude of $\sigma$ does lead to a lower GC value in SC, setting $\sigma$ arbitrarily close to zero is not advisable." Isn't it the case that the minimum is obtained at $\sigma=0$? If so, doesn't this suggest that the loss function is not capturing some important aspects of the problem? Are the empirical results simply an artifact of when the optimization is terminated?

* The scope of the paper seems limited as the authors investigate the learning of a single hyperparameter $\sigma^2$. So at best (if I understand correctly), the significance of the method is that it can learn this hyperparameter with a few iterations of Newton's method as opposed to a simple one-dimensional sweep of reasonable hyperparameter values. But it is not clear that the proposed optimization is much faster than a simple sweep because one of the alternating steps of the optimization is precisely to compute the bottom K eigenvectors of the graph Laplacian, and it seems (from Figures 3 and 4) that hundreds of steps (epochs) are needed per optimization.

* The authors claim that the validation of hyperparameters is not possible in an unsupervised setting. I understand that it is not possible to hold out a labeled validation set when the data has no labels.  But the claim does not reflect how unsupervised methods (and particularly, methods for clustering) are used in practice. Typically, the practitioner examines the clusters that are discovered and evaluates them using some domain-specific prior knowledge; moreover, this type of evaluation is not especially onerous in the case of a single hyperparameter.

---

> ### Author Response · Authors · 2025-01-16
> **Comment regarding loss function capturing important aspects of the problem**
>
> > **Comment 1:** "Is the loss function sensible? Theorem 2 shows that the loss is non-decreasing for positive $\sigma$. On page 5, the authors state that "although reducing the magnitude of $\sigma$ does lead to a lower GC value in SC, setting
> $\sigma$ arbitrarily close to zero is not advisable." Isn't it the case that the minimum is obtained at $\sigma=0$? If so, doesn't this suggest that the loss function is not capturing some important aspects of the problem? Are the empirical results simply an artifact of when the optimization is terminated?"
>
> We are grateful to the reviewer for posing this insightful question and allowing us to clarify this important aspect.
>
> We wanted to showcase that there is a distinction between **minimizing GC (what we do)** and achieving a **minimal GC (manually setting $\sigma$ to extremely low value )** in generating meaningful clusters.
>
> **Minimal GC**:
> A very small GC is possible by setting $\sigma$ to an extremely low value (e.g., $\sigma=0$). Doing so greatly amplifies all distances to the extent that, after applying the exponential function, all edge similarity values in the graph become uniformly minimal. This results in each data point being isolated, essentially forming its own singleton cluster. This approach not only equalizes all the edge similarity values in the dataset but also disrupts the data structures, as it erases the ordering of the edge similarities and the hierarchical relationships among data points. Consequently, the data topology is lost, leading to all eigenvalues turning to zero, not just the first $K$ ones.
>
> **Minimizing GC**:
> Conversely, beginning with a high GC keeps the hierarchical structure of edge similarities and maintains the data topology.
> The exponentiation process disproportionally magnifies larger distances more than smaller ones.
> By gradually reducing $\sigma$ (guided through the minimization of GC), we selectively amplify the largest distances first 'one by one,' causing their corresponding edge similarity values to become equally low after exponentiation.
> This ' sequential pruning' of the farther distances first would keep the order of smaller distances (i.e., bigger edge similarities) and help preserve their relational hierarchy, thereby retaining the meaningful structure of the data.
>
> This gradual reduction of both GC and $\sigma$ continues until the first $K$ eigenvalues become significantly small. Applying the bottom $K$ eigenvectors ensures the formation of the most distinct $K$ clusters, achieving a minimal GC that reflects the most sensible cluster divisions.
>
> **Through this guided process, we iteratively adjust $\sigma$ and cease its modification once the corresponding gradient is negligible. By doing so, we achieve an optimally minimal GC**

---

> > ### Comment · Reviewer_QMhR · 2025-01-16
> > **Loss function**
> >
> > Thank you very much for these comments and explanations.
> >
> > I think we agree on the following: (1) a finite but non-zero $\sigma$ is interpretable as one that modifies distances in an "empirically optimal" way for clustering; (2) the proposed loss function is convex and has a minimum at zero; (3) the proposed method does not seek this "mathematical" optimum, but terminates according to some criterion on the gradient.
> >
> > I think where we disagree is the following: (1) to me, it feels unsatisfactory, even misleading, to formulate and *advertise* the problem as a convex optimization (with all the attending benefits of convexity), and then to admit not only that the global minimum can be trivially identified but also that this global minimum corresponds to a flawed solution. (2) By your own explanation, what you want is a value of $\sigma$ that reduces the value of the first $K$ eigenvalues, but does not obliterate the distinction between all the eigenvalues. This suggests that the loss function or optimization is itself incomplete; it seems that the loss function needs either (a) another term referring to the remaining eigenvalues or (b) a regularizer (e.g., $\log\frac{1}{\sigma}$)  to rule out the trivial minimum at $\sigma=0$. In case (a) , I suspect that the loss function would no longer remain convex (though it would be very interesting if it did); in case (b), the regularizer would introduce another hyperparameter (i.e., the constant by which it is multiplied). (3) In the current framework, there seems to be an implicit hyperparameter, which is the gradient-cutoff criterion to terminate the optimization. This cutoff must be chosen wisely, and it's not clear why it would or should have the same value for all problems. This seems to undermine the main (mathematical) contribution of the paper that $\sigma$ can be determined by a convex optimization with no tunable hyperparameters. Or perhaps there is a mathematical argument why the gradient-termination criterion should not vary across problems? That also would be very interesting.

---

> ### Author Response · Authors · 2025-01-16
> **Comment regarding parameter sweeping**
>
> > **Comment 2:** "The scope of the paper seems limited as the authors investigate the learning of a single hyperparameter.
> So, at best (if I understand correctly), the significance of the method is that it can learn this hyperparameter with a few iterations of Newton's method as opposed to a simple one-dimensional sweep of reasonable hyperparameter values.
> But, it is not clear that the proposed optimization is much faster than a simple sweep because one of the alternating steps of the optimization is precisely to compute the bottom K eigenvectors of the graph Laplacian, and it seems (from Figures 3 and 4) that hundreds of steps (epochs) are needed per optimization."
>
> We thank the reviewer for posing another very important question.
>
> When conducting parameter sweeping, there is a risk of overshooting, resulting in a very small $\sigma$. This scenario often leads to the formation of many singleton clusters. Equally likely, undershooting the parameter may excessively prolong the process or fail to achieve an optimal GC within a reasonable number of iterations. In both scenarios, the cluster quality suffers as the process either yields non-sensible clusters composed of multiple singletons or fails to achieve the best possible separation of the K clusters inherent in the data.
>
> While our method requires the computation of the bottom K eigenvectors of the graph Laplacian at each step, it provides the gradient (i.e., for the direction of steepest descent) and second derivative (to determine the optimal step size).
> This combination of gradient and optimal step size calculation is crucial for guided and accelerated convergence of $\sigma$.
> This strategic approach not only accelerates the optimization process but also enhances the precision and sensitivity of the clustering results, thereby ensuring the best possible separation of the K clusters inherent in the data.

---

> > ### Comment · Reviewer_QMhR · 2025-01-16
> > **Newton's method**
> >
> > Thanks very much for these comments. I agree that Newton's method could potentially converge much faster than a uniform sweep.
> >
> > Two follow-up questions:
> > 1) The plots suggest that Newton's method is converging to $\sigma=0$. Do you believe this to be the case?
> > 2) If I understand the plots correctly, the method is running for more than one hundred iterations (epochs) of Newton's method. Is this correct? But in most one-dimensional problems, Newton's method (when it is stable) converges much faster than this because it doubles the number of significant digits at each iteration. Can you help to reconcile these observations?
> >
> > Thanks in advance for your answers.

---

> > > ### Author Response · Authors · 2025-01-17
> > > **Newton's method**
> > >
> > > Thank you very much for these critical questions.
> > >
> > > >**Question 1:** The plots suggest that Newton's method is converging to $\sigma=0$. Do you believe this to be the case?
> > >
> > > Due to the log-log scale nature of the plot, it can be a bit misleading as it shows $\sigma$ becoming slowly small (hence the steep ascent in the log scale), but it can serve as a basis to conclude that the method is heading towards zero. We can only say that $\sigma$ is becoming smaller.
> > >
> > > >**Question 2:** If I understand the plots correctly, the method is running for more than one hundred iterations (epochs) of Newton's method. Is this correct? But in most one-dimensional problems, Newton's method (when it is stable) converges much faster than this because it doubles the number of significant digits at each iteration. Can you help to reconcile these observations?
> > >
> > > We tried to explain this in the previous extended reply.
> > > Both $\sigma$ and the computed eigenvectors $f_i$ contribute to reducing $ loss(\sigma) = \sum_{i=0}^{K} f_i^T L(\sigma) f_i $.
> > > Unlike traditional optimization problems where only the parameter is updated, in our case, even the $ loss(\sigma)$ function itself changes as we  $f_{i}$ updated at every step using EVD on $L(\sigma)$.
> > > In other words, the plateau of the loss function becomes progressively flatter. This increasingly flat landscape necessitates numerous iterations to achieve convergence, even when using the efficient Newton method.
> > >
> > > Thank you for your time, and looking forward to any further questions.

---

> ### Author Response · Authors · 2025-01-17
> **Loss function**
>
> We thank the reviewer for these thorough comments.
>
> Considering the multiple points raised, we will answer them by explaining the nature loss function and why Newton's method does not reach $\sigma=0$ in addition to discussing its convergence properties and speed.
>
> **1. Convexity of the loss:**
> Indeed, the loss function is convex (as per Theorem 1) but not strictly convex.
> The $loss(\sigma)$ function would possess only one global minimum in a strictly convex scenario. However, in our case, an open interval of minima exists.
>
> Specifically, within the  the convex domain $\Sigma=(0,\frac{\sqrt(6)}{3}]$ there exists an optimal $\exists\sigma_{optima}>0,s.t:\sigma_{optima}\in\\Sigma$ where $loss(\sigma)=\sum_{i=0}^{K}\lambda_{i}=0$ but also its gradient $\frac{\partial loss(\sigma)}{\partial\sigma}=0$ is also zero.
> Furthermore, the loss and the gradient remain zero $loss(\sigma)=0,\frac{\partial loss(\sigma)}{\partial\sigma}=0$ over an open interval $\forall\sigma\in\Sigma_{1} =(0,\sigma_{optima}]$.
>
> In the complementary set $\Sigma_{2}=(\sigma_{optima},\frac{\sqrt(6)}{3}],s.t: \Sigma =\Sigma_{1}\cup\Sigma_{2}$ the loss and its gradient and its second derivative are all positive $loss(\sigma)>0,\frac{\partial loss(\sigma)}{\partial\sigma}>0,\frac{\partial^2 loss(\sigma)}{\partial\sigma^2}>0$
>
> In the open interval $\Sigma_{1}$, however, we get not only $loss(\sigma)=\sum_{i=0}^{K}\lambda_{i}=0$ everywhere but also $\sum_{i=0}^{N}\lambda_{i}=0$ at $\sigma=0$ ($N\to$ total number of eigenvalue) and everything in between.
>
> The loss function thus achieves a minimum value (zero) in $\Sigma_{1}$ is strictly convex in  $\Sigma_{2}$ while being convex across $ \Sigma =\Sigma_{1}\cup\Sigma_{2}.$
>
> We focus on $\sigma_{optima}$ because it represents the threshold beyond which more than the first $K$ eigenvalues reach zero.
>
> **2. Newton's method does not reach $\sigma=0$**
>
> Now we try to answer why Newton's method stops at $\sigma_{optima}$ and why it does not overstep it (or heads) towards zero (and by extension negative values) when we start at $\sigma=\frac{\sqrt(6)}{3}.$
>
> Notice that Newton's method follows the gradient with adaptive stepping.
> Starting from $\sigma_{i}$ the subsequent  $\sigma_{i+1}, $ i is determined by the intersection of the tangent to the $loss(\sigma)$ curve at $\sigma_{i}$ with the $\sigma-$ axis (illustrated with a figure in the supplementary material for clarity).
> With $\frac{\partial^2 loss(\sigma)}{\partial\sigma^2}>0$, the loss function’s convexity ensures that every tangent lies below the loss curve.
>
> Reaching any $\sigma$ beyond $\sigma_{optima}$ $, i.e.,\sigma<\sigma_{optima} (i.e., \sigma\in\Sigma_{1})$ using this subsequent tangents would require starting from a position $\sigma>\sigma_{optima},(i.e., \sigma\in\Sigma_{2})$ and overstepping $\sigma_{optima}$ it needs to go above the $loss(\sigma)$ curve.
> This scenario is prevented by the convexity of the $loss(\sigma)$ in $\Sigma$ and strictly convex in $\Sigma_{2}$.
> Thus, it has to reach $\sigma_{optima}$.
>
> Moreover, once it reaches $\sigma_{optima}$, it cannot escape it as it attains the gradient zero, and 'automatically' the update stops.
>
> **Hence, $\sigma$ does not head towards zero or negative values as overstepping the first point that it reaches zero gradients is not possible using Newton update in a convex set.**
>
> **3. Speed and Convergence:**
>
> **Speed:** The optimization speed is primarily influenced by the continuous changing of $L(\sigma)$ and its eigenvectors within the loss function $ \text{loss}(\sigma) = \sum_{i=0}^{K} f_i^T L(\sigma) f_i $, where $f_i = \text{EVD}(L)$.
> Since not only $\sigma$ reduces the $loss(\sigma)$ but also the computed eigenvectors $f_{i}$ the changing landscape of $loss(sigma) $ becomes progressively flatter, necessitating numerous iterations, even with the efficient Newton method.
>
> **Converngence and gradient:**
> We do not need and cannot define a gradient threshold to stop the optimization, as it is strictly data-dependent.
> However, as we train, we stop if we hit the $\sigma_{optimal}$ and get no further update as its gradient is precisely zero or we run out of the floating points relation for computation of the eigenvalue decomposition and or computation of the gradient and or second derivative.
>
> Thank you for your time, and we look forward to your feedback.

---

> > ### Comment · Reviewer_QMhR · 2025-01-21
> > **Convexity of the loss function**
> >
> > In response to my questions about the loss function, the authors have suggested that:
> >  (a) the loss function is not strictly convex;
> >  (b) because the loss function is not strictly convex, it has a connected interval of global minima.
> >  (c) at the right endpoint of this interval, the bottom K eigenvalues of the graph Laplacian vanish, but the remaining eigenvalues do not vanish, and therefore the right endpoint is a sensible value of $\sigma$ to select.
> >
> > This argument, if correct, would resolve certain issues. However, I do not believe that this argument is correct, and in any case, it cannot be merely asserted, but must in fact be mathematically proven. I will express my doubts regarding (a) and (b) and also give a short proof that point (c) is incorrect.
> >
> > First, on claim (a): isn't the function $\exp(-a/\sigma^2)$ strictly convex in $\sigma$ for $a>0$ and $\sigma$ sufficiently small? If so, it seems that the loss function should inherit this property.
> >
> > On claim (b): even if a convex function is not strictly convex, it does not necessarily follow that it has a connected interval of global minima. For example, the function $f(x) = |x|$ is convex, but not strictly convex, and it has a single global minimum.
> >
> > Most importantly, on claim (c): the authors are claiming (I think) that for different values of $\sigma$ the graph Laplacian has a different number of zero eigenvalues. But a graph Laplacian only has a multiplicity of zero eigenvalues if the underlying graph is disconnected, and if I understand correctly, the edge weights of the graph Laplacian do not vanish for any finite value of the $\sigma$.  Thus the topological connectivity of the weighted graph does not depend on the value of $\sigma$, and thus neither can the number of connected components (or equivalently) or the number of zero eigenvalues.
> >
> > It would be helpful if in their reply the authors could explicitly express the graph Laplacian weights as a function of $\sigma$. Maybe this would resolve some of the confusions and points of contention in our discussion.

---

> > > ### Author Response · Authors · 2025-01-23
> > > **Reply on the convexity of the loss function**
> > >
> > > We thank the reviewer for the insightful and helpful follow-up discussion.
> > >
> > > We wanted to agree with the reviewer's points a and b (although we meant specifically, not in general).
> > >
> > > However, we wanted to argue that the $\sigma$ still plays an important role in lowering the GC and emphasizing the linear separation of the lowest cuts.
> > >
> > > Given that the proposed kernel is strictly convex in $\Sigma$ and produces a symmetric Laplacian matrix ($L(\sigma)$), its eigenvalues are strictly non-decreasing.
> > > Since the first and second derivative ($, i.e., \frac{\partial L(\sigma)}{\partial \sigma}, \frac{\partial^{2} L(\sigma)}{\partial \sigma^{2}}$) are also Laplacian matrices that share identical eigenvectors, their corresponding eigenvalues are also non-decreasing.
> > >
> > > $$eqn_1:0=\lambda_{1}(\sigma)\leq\cdots\leq\lambda_{K}(\sigma)\leq\cdots\leq\lambda_{N}(\sigma)$$
> > >
> > > $$eqn_2:0=\frac{\partial \lambda_{1}(\sigma)}{\partial \sigma}\leq\cdots\leq\frac{\partial \lambda_{K}(\sigma)}{\partial \sigma}\leq\cdots\leq\frac{\partial \lambda_{N}(\sigma)}{\partial \sigma}$$
> > >
> > > $$eqn_3:0=\frac{\partial^{2} \lambda_{1}(\sigma)}{\partial \sigma^{2}}\leq\cdots\leq\frac{\partial^{2} \lambda_{K}}{\partial \sigma}(\sigma^{2})\leq\cdots\leq\frac{\partial^{2} \lambda_{N}(\sigma)}{\partial \sigma^{2}}$$
> > >
> > >
> > > Now lets define the loss $L_{j}(\sigma)=\sum_{i=1}^{j}\lambda_{i}(\sigma)$ s.t: $j\in{1,N}$ where
> > >
> > > $$eqn_4:0=L_{1}(\sigma)< \cdots <  L_{K}(\sigma)< \cdots <  L_{N}(\sigma)$$
> > >
> > >
> > > $$eqn_5:0=\frac{\partial L_{1}(\sigma)}{\partial \sigma}<\cdots < \frac{\partial L_{K}(\sigma)}{\partial \sigma}<\cdots < \frac{\partial L_{N}(\sigma)}{\partial \sigma}$$
> > >
> > > $$eqn_6:0=\frac{\partial^{2} L_{1}(\sigma)}{\partial \sigma^{2}}<\cdots < \frac{\partial^{2} L_{K}(\sigma)}{\partial \sigma^{2}}<\cdots < \frac{\partial^{2} L_{N}(\sigma)}{\partial \sigma^{2}}$$
> > >
> > > For $\sigma=0$ we get $0=\lambda_{1}(\sigma)=\cdots=\lambda_{K}(\sigma)=\cdots=\lambda_{N}(\sigma)$, we are not interested as we obliterate the data topology.
> > >
> > >
> > > Since $\sigma=0\notin\Sigma$ (under the assumption of infinite precision), and every edge weight is positive
> > > $$eqn_7: w(\sigma)=e^{\frac{-d}{\sigma^{2}}}>0,\frac{\partial w(\sigma)}{\partial \sigma}>0,\frac{\partial^{2} w(\sigma)}{\partial \sigma^{2}}>0,\forall\sigma\in\Sigma,\forall d\in R_{>0}.$$
> > > Then $\forall j,s.t: 1<j\leq N,\forall\sigma\in\Sigma$ we get
> > >
> > > $$eqn_8: 0<\lambda_{j}(\sigma),0<\frac{\partial \lambda_{j}(\sigma)}{\partial \sigma},0<\frac{\partial^{2} \lambda_{j}(\sigma)}{\partial \sigma^{2}},$$ resulting in
> > > $$eqn_9: L_{j}(\sigma)=\sum_{i=1}^{j}\lambda_{i}(\sigma)>0,\forall\sigma\in\Sigma$$
> > >
> > > $$eqn_{10}: \frac{\partial L_{j}(\sigma)}{\partial \sigma}=\sum_{i=1}^{j}\frac{\partial \lambda_{i}(\sigma)}{\partial \sigma}>0,\forall\sigma\in\Sigma.$$
> > >
> > > $$eqn_{11}: \frac{\partial^{2} L_{j}(\sigma)}{\partial \sigma^{2}}=\sum_{i=1}^{j}\frac{\partial^{2} \lambda_{i}(\sigma)}{\partial \sigma^{2}}>0,\forall\sigma\in\Sigma$$
> > >
> > > Considering that the edge weights are part of an ordered set of non-decreasing magnitude $W=[w_{1}(\sigma)\leq\cdots \leq w_{N}(\sigma)]$.
> > > Notice that all the edge $(\forall w\in W)$ values are part of an exponential curve.
> > >
> > >
> > > Similarly, the eigenvalues and the losses in $eqn_{1},eqn_{4}$ inherent in the properties of the edge weights, hence part of an ordered set whose values are part of an exponential curve.
> > >
> > > **1. Update of $\sigma$ and disconnected components:**
> > >
> > > As $\sigma$ decreases significantly in magnitude, the smaller edge values in $W$ get **truncated** towards zero one by one **due to the limited precision** while maintaining their non-decreasing order.
> > > As a result, the first eigenvalues get also truncated towards zero while maintaining their order in $eqn_{1}$.
> > > Under this consideration, we wanted to let the reviewer know why we 'conjectured' the first $K$ eigenvalues get towards zero faster than the latter ones.
> > >
> > > **2. Update of $\sigma$ and no disconnected components:**
> > > In such a situation, as $\sigma$ keeps on decreasing, then edge weights also decrease in magnitude but also increase relative consecutive gap ($i.e., w_{i+1}(\sigma)-w_{i}(\sigma)<w_{i+2}(\sigma)-w_{i+1}(\sigma)$).
> > > **The latter is true since the smaller weights become exponentially smaller than the bigger weights.**
> > > This would extend towards the eigenvalues in $eqn_{1}$.
> > > As a result, as of this, the graph cut decreases in magnitude ($L_{K}(\sigma)$), hence providing better linear separation; the **relative eigen-gap** would increase for each step.
> > >
> > > Thus, in both cases (1 and 2), the absolute value for the GC would decrease, and the **relative eigen-gap** would increase.
> > >
> > > We hope to answer the reviewer's question regarding the importance of the loss and the $\sigma$ for the data topology and looking forward to the feedback.

---

> > > > ### Comment · Reviewer_QMhR · 2025-01-28
> > > > **Convexity of the loss function (con't)**
> > > >
> > > > I thank the authors for their previous answers. Here I wish to confirm my understanding of the proposed method.
> > > >
> > > > 1) I understand that in practice that the optimization cannot be carried out to infinite precision, and also that it cannot be performed for an arbitrarily large number of iterations. But if the above were possible, would the optimization converge to a global minimum at $\sigma=0$?
> > > >
> > > > 2) Though the loss function does not refer explicitly to the $relative$ eigengap between the bottom $K$ eigenvalues and the other eigenvalues, in practice, when there is sufficient precision, the relative eigengap is the quantity of interest?

---

> > > > > ### Author Response · Authors · 2025-01-28
> > > > > **Convexity of the loss function (con't)**
> > > > >
> > > > > We thank the reviewer for the valuable input and this opportunity to follow up on the explanation of the method.
> > > > >
> > > > > >1. I understand that in practice that the optimization cannot be carried out to infinite precision, and also that it cannot be performed for an arbitrarily large number of iterations. But if the above were possible, would the optimization converge to a global minimum at $\sigma=0$?
> > > > >
> > > > >
> > > > > In the case of **infinite precision** and a **finite number** of steps, the **model does not converge at $\sigma=0$**.
> > > > > This is due to the convexity of the loss and the Newton update natyre.
> > > > >
> > > > > A proof sketch we could provide is by contradiction.
> > > > >
> > > > > Using Newton update, assuming the model converges $\sigma_{nr_{Final}}=0$. This update means that there is a **straight line** that goes from $\sigma_{nr_{Final-1}}>0\to\sigma_{nr_{Final}}=0$.
> > > > >
> > > > > Because of the nature of the Newton update, this **straight line** intersects the $\sigma-axis$ at $\sigma_{nr_{Final}}=0$.
> > > > >
> > > > > Furthermore, since $loss(\sigma_{nr_{Final}}=0)=0$, one can safely say that the loss also touches the $\sigma-axis$ at $\sigma_{nr_{Final}}=0$.
> > > > >
> > > > > Hence this **straight line** touches the loss twice, once at $\sigma_{nr_{Final-1}}>0$ and once more at $\sigma_{nr_{Final}}=0$.
> > > > >
> > > > > This **straight line**, however, is a tangent of the loss function at $\sigma_{nr_{Final-1}}>0$.
> > > > >
> > > > > Since the loss is strictly convex in $(0,\frac{\sqrt{6}}{3})$, this tangent has to always be below the loss function. Thus, this straight line cannot touch the loss twice.
> > > > >
> > > > > One can safely say that every straight line from Newton update touches the strictly convex loss function only once $\forall \sigma\in(0,\frac{\sqrt{6}}{3})$.
> > > > >
> > > > > Hence, the proof by contradiction.
> > > > >
> > > > > **Notice that at $\infty$ steps, $\sigma\to 0$ asymptotically.**
> > > > >
> > > > > >2. Though the loss function does not refer explicitly to the $relative$ eigengap between the bottom $K$ eigenvalues and the other eigenvalues, in practice, when there is sufficient precision, the relative eigengap is the quantity of interest?
> > > > >
> > > > > Relative eigengap is an important auxiliary quantity of interest while $loss=\sum_{i=1}^{K}\lambda_{i}$ is more of a primary importance.
> > > > > The proposed loss is more of an explicit quality measure, while the eigengap is more of an implicit quality measure of the first K clusters.
> > > > >
> > > > > Minimizing $loss=\sum_{i=1}^{K}\lambda_{i}$ would enable the eigenprojection of the data using the bottom $K$ eigenvectors to have maximum interclass separability whilst having minimum intraclass variability.
> > > > > This configuration provides better embeddings and facilitates downstream tasks such as $K$-means.
> > > > >
> > > > > On the other hand, the relative eigengap helps determine the optimal number of clusters.
> > > > > The relatively big eigengap between $\lambda_{K}$ and $\lambda_{K+1}$ indicates the first $K$ clusters are more distinguishable than any of the follow-up $N-K$ clusters.
> > > > > However, this relatively big eigengap does not guarantee good data embeddings of the $K$ clusters since the  $loss=\sum_{i=1}^{K}\lambda_{i}$ can be of big magnitude.
> > > > >
> > > > > Thus, to $directly$ $improve$ $the$ $quality$ of these $K$ clusters, $minimizing$ $directly$ $the$ $proposed$ $loss=\sum_{i=1}^{K}\lambda_{i}$.
> > > > >
> > > > > We hope our responses address the questions posed and thank the reviewer for their time.
> > > > >
> > > > > We are happy to address any follow-up questions.

---

> ### Author Response · Authors · 2025-01-20
> **The proposed usage of the algorithm by practitioners**
>
> > **Comment 3:** "The authors claim that the validation of hyperparameters is not possible in an unsupervised setting. I understand that it is not possible to hold out a labeled validation set when the data has no labels. But the claim does not reflect how unsupervised methods (and particularly, methods for clustering) are used in practice. Typically, the practitioner examines the clusters that are discovered and evaluates them using some domain-specific prior knowledge; moreover, this type of evaluation is not especially onerous in the case of a single hyperparameter."
>
> We appreciate the reviewer for raising another important question.
>
> Removing hyper-parameters from the clustering process can enhance the adaptability of spectral clustering in various settings.
>
> For instance, consider the application of clustering point clouds obtained from radar mounted on vehicles. In such scenarios, where timely processing is crucial for accurately identifying moving vehicles, the point cloud data distribution changes continuously. Adhering to a fixed hyper-parameter in these dynamic conditions makes it challenging to maintain optimal performance.
>
> Another situation involves dealing with complex datasets with exceptionally high sensitivity to clustering hyperparameter.
> These datasets often require numerous iterations of labeling by domain experts, as the clustering outcomes can be highly variable and difficult to stabilize. Furthermore, predicting the number of necessary iterations for fine-tuning the optimal hyper-parameter settings becomes difficult since it depends heavily on the extent of labeling provided by the experts.
>
> A third scenario includes using this method to self-supervise deep learning classifiers using contrastive learning. While generating positive examples might be straightforward through data augmentations, deriving high-fidelity negative examples poses a challenge. Using the clustering centers to obtain high-quality negative examples can significantly enhance this process.
> In such a continuously changing data distribution situation, ensuring optimal performance by utilizing a fixed hyper-parameter value is hard.

---

> ### Author Response · Authors · 2025-01-23
> **Reply to the requested changes**
>
> We thank the reviewer for the following requested changes. We did the address them as follow:
>
> >Lemma 1 follows immediately from elementary properties of kernels; see Propositions 13.1 and 13.2 in Learning with Kernels (Scholkopf & Smola, 2002). The proof does not require a half-page of exposition.
>
> We updated this Lemma accordingly and updated the text.
> >The constant $\frac{\sqrt{6}}{3}$ appears mysteriously throughout section 2. Somewhere at the top, the authors should simply observe that the function $f(\sigma)=exp(-a/\sigma^{2})$ is convex over the interval $(0,\frac{6}{3})$ for all $a$ in some appropriate interval.
>
> This is mentioned as suggested in the method section at the very top.
> >For completeness, the text should precisely and prominently state how the kernel matrix K, the adjacency matrix A, and the Laplacian are related to one another as follows: **The Laplacian matrix $L$ is defined as $L=D-A$ where $D$ is degree matrix. The degree matrix is diagonal, with each diagonal entry ($i.e., D_{ii}$) representing the sum of the weights of all edges connected to node ($i$), ($i.e., D_{ii} = \sum_{j=1}^n A_{ij}$).**
>
> We did a detailed description of the kernel matrix K, the adjacency matrix A, and the Laplacian L and degree matrix D right after equation 4 in section 2.1
>
> >The authors claim in Theorem 1 that the sum of K eigenvalues from the graph Laplacian is a convex function of $\sigma$. It is a well-known result in convex optimization that the sum of the top K eigenvalues of a symmetric matrix is a convex function of its matrix elements; see page 118 in Convex Optimization (Boyd & Vandenberghe, 2003). Can the authors (drastically) simplify the proof of Theorem 1 using this property, along with the observation that $f(\sigma)=exp(-a/\sigma^{2})$ is convex in $\sigma$
>  for sufficiently small $\sigma$ ? Or if not -- if the proof contains other essential ideas -- can the authors more clearly highlight the additional or alternative steps that are required?
>
> Many thanks to the reviewer for this suggestion; however, we still believe that keeping the proofs is necessary as we optimize w.r.t both $f$ and $\sigma$, not just the latter. Hence, it not only makes the paper more complete but also emphasizes the fact that $f$ are updated continuously. Hence, we wrote in the text: **Moreover, as both $\sigma$ and the series of eigenvectors $f_{1, \ldots, K}$ are iteratively updated to minimize the loss function, it can be concluded that the loss function is continuously evolving, increasingly approaching a more flattened landscape.**
>
> >Graph Laplacian embeddings are computed from bottom eigenvectors; the paper should mention this explicitly.
>
> This is a very helpful suggestion and is mentioned above in Equation 6.
> >How do you prevent "over-stepping" in Newton's method (eq. 9) so that $\sigma$ does not obtain negative values?
>
> As discussed previously, given that Newton's method follows tangent curves, it does not overstep outside the convexity interval.
> >Can the authors generalize their method to estimate a positive-definite covariance matrix $\Sigma$
>  rather than a single variance $\sigma^{2}$?
>
> We have been considering the same but have not come to a solid and working objective yet!

---

### Author Response · Authors · 2025-02-18
**Overview of the important amendments in the manuscript**

We wanted to thank the reviewers for the very thorough discussion and suggestions and wanted to provide an outline of the important amendments we provided in the text.

As suggested by the **Reviewer QMhR**.
-  A paragraph was added in the method section to emphasize the convexity of the proposed kernel
-  Definition of the kernel function was added before Lemma 1.
-  Lemma 1 was restructured into a shorter form .
-  A paragraph was added after eqn 9 to emphasize the evolving nature of the GC loss function.

As suggested by the **Reviewer 5tpp**.
-  A notation section was added to the Appendix.
-  Typo in Theorem 2 regarding eqn 18 and the follow-up.
- Another lemma regarding the relative nature of the GC was included in Appendix E.

As suggested by **Reviewer kuMq**
-  We fixed typo in Corollary 1.

We would like to sincerely thank all the reviewers for their valuable feedback and for raising these important suggestions.
We greatly appreciate the time and effort dedicated to improving our work and are always grateful for additional feedback.

---

### Decision · Action_Editor_QWzh · 2025-02-22

**Recommendation:** Reject

**Comment:**

Two out of the three reviewers recommended to reject this paper. The main concern is the scientific soundness and clarity of the paper below, I provide additional concerns raised by the reviewers:

- The paper would benefit from a more coherent presentation. The assumptions, hypotheses, and results are not consistently distinguished, which could result in misunderstandings about the motivations and contributions of the work.

-The optimization process results in a trivial minimum at \(\sigma=0\). The authors appear to recognize that an optimization carried out with infinite precision and an infinite number of iterations would ultimately reach this minimum. This perspective casts a peculiar light on the overall framing of the method.

-During the discussion, the authors made several assertions that they subsequently retracted, such as claims regarding the convexity of the problem.

-Some mathematical assumptions were not sufficiently justified.
The approach was seen as narrow, as it focuses on optimizing a single hyperparameter (which could be generalized to separate values for distinct features).

For these reasons, I believe that the paper requires substantial revisions before it is ready for publication, and I recommend its rejection in its current form. However, I recognize the potential value of the work and encourage the authors to refine their mathematical formulations, improve the clarity of their presentation, and strengthen their theoretical justifications. Addressing these concerns could significantly enhance the rigor and impact of their contributions to spectral clustering. I also recommend sending the paper to a different venue since the paper was rejected twice by TMLR.

**Audience:**

Although the topic of spectral clustering with RBF kernels may be considered somewhat outdated, I believe this type of kernel still offers valuable insight, particularly in neural network-based methods.

**Claims And Evidence:**

The authors tackle the challenge of determining an optimal bandwidth for an RBF-based spectral clustering. Throughout the review and discussion phase, several critical issues were raised by the reviewers regarding the paper. Most notably, the work exhibits concerns that reflect a lack of rigor in its mathematical formulations. Specifically, two reviewers have indicated that the paper is not yet suitable for publication. Among the reasons mentioned are an incoherent description of various mathematical details and a fundamental flaw in the proposed objective, which results in a trivial solution at a bandwidth of zero. Furthermore, based on the discussions, it remains unclear whether this trivial solution is merely a consequence of floating-point operations. If that is the case, it suggests that the entire presentation of the paper requires significant revisions.